# Auditing Sybil: Explaining Deep Lung Cancer Risk Prediction Through Generative Interventional Attributions

**Bartlomiej Sobieski** [* 1 2 3]  **Jakub Grzywaczewski** [* 2 3]  **Karol Dobiczek** [* 4]  **Mateusz Wójcik** [2]  **Tomasz Bartczak** [5]
**Patryk Szatkowski** [6]  **Przemysław Bombiński** [6]  **Matthew Tivnan** [7 8 9]  **Przemyslaw Biecek** [1 2 3]

## Abstract

Lung cancer remains the leading cause of cancer mortality, driving the development of automated screening tools to alleviate radiologist workload. Standing at the frontier of this effort is Sybil, a deep learning model capable of predicting future risk solely from computed tomography (CT) with high precision. However, despite extensive clinical validation, current assessments rely purely on observational metrics. This correlation-based approach overlooks the model's actual reasoning mechanism, necessitating a shift to causal verification to ensure robust decision-making before clinical deployment. We propose S(H)NAP, a model-agnostic auditing framework that constructs generative interventional attributions validated by expert radiologists. By leveraging realistic 3D diffusion bridge modeling to systematically modify anatomical features, our approach isolates object-specific causal contributions to the risk score. Providing the first interventional audit of Sybil, we demonstrate that while the model often exhibits behavior akin to an expert radiologist, differentiating malignant pulmonary nodules from benign ones, it suffers from critical failure modes, including dangerous sensitivity to clinically unjustified artifacts and a distinct radial bias.

## 1. Introduction

Lung cancer remains the leading cause of cancer mortality worldwide (Bray et al., 2024), driving global screening efforts with Low-Dose Computed Tomography (LDCT) (De Koning et al., 2018; Pastorino et al., 2019). To ad-

dress the bottlenecks of radiologist workload and diagnostic variability, automating LDCT interpretation via AI offers a path toward high-throughput, early detection (Duranti et al., 2025; Arshad et al., 2025; Field & Oudkerk, 2025). Standing at the frontier is Sybil (Mikhael et al., 2023), a deep learning model predicting 6-year lung cancer risk solely from a single CT scan. Sybil has achieved significant milestones in clinical translation, demonstrating robustness across diverse populations and settings through extensive retrospective and prospective validation (Simon et al., 2023; 2024; Yang et al., 2025; Kim et al., 2025b; Durney et al., 2025; Aro et al., 2024; Pasquinelli et al., 2025; Krule et al., 2025; Li et al., 2025).

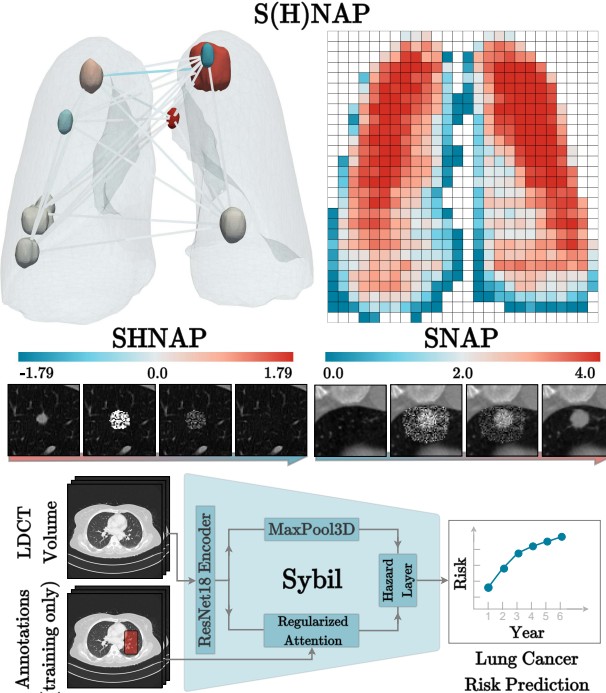

*Figure 1.* Sybil (**bottom**) is a frontier model for lung cancer risk prediction from a single CT scan. We propose S(H)NAP (**top**), a novel framework for auditing such models through diffusion-bridge-based generative interventions on pulmonary nodules (**middle**). SHNAP (**top left**) decomposes the model's prediction into individual nodule contributions and inter-nodule interactions by replacing them with healthy tissue. SNAP (**top right**) probes volumetric sensitivity by systematically inserting nodules of known malignancy, revealing spatial biases in risk estimation.

[1]University of Warsaw [2]Warsaw University of Technology [3]Centre for Credible AI [4]Jagiellonian University [5]Google [6]Medical University of Warsaw [7]Massachusetts General Hospital [8]Harvard Medical School [9]Center for Advanced Medical Computing and Analysis. Correspondence to: Bartlomiej Sobieski <b.sobieski@uw.edu.pl>. * Equal contribution

*Proceedings of the 43$^{rd}$ International Conference on Machine Learning*, Seoul, South Korea. PMLR 306, 2026. Copyright 2026 by the author(s).

However, current assessments rely purely on observational studies. This correlation-based approach confirms *that* the model works, but not *why* or *when* it might fail, creating a critical oversight for high-stakes deployment. To address this, we propose S(H)NAP, a framework shifting the paradigm from observational validation to *interventional* auditing. By synergizing game-theoretic attributions, generative modeling, and radiological expertise, we rigorously audit Sybil using a model-agnostic framework applicable to models in lung cancer screening scenarios. Specifically, **1.** we introduce novel methodologies for synthetic interventions on pulmonary nodules in 3D LDCT using diffusion bridge modeling. **2.** We construct SHNAP and SNAP, intervention-based attribution methods tailored for lung cancer risk prediction. **3.** We empirically verify that Sybil functions as a linear model with pairwise interactions over pulmonary nodules. **4.** We reveal critical misalignments, including a radial sensitivity bias and learned spurious artifacts, highlighting the necessity of interventional auditing.

## 2. Related works

**Sybil.** We define Sybil (Mikhael et al., 2023) as a parameterized distribution $p_\theta(\mathbf{y} \mid \cdot)$ over lung cancer risk $\mathbf{y} = (y_i)_{i=0}^6 \in [0, 1]^7$. The outputs consist of a base risk $y_0$ and cumulative risks $y_i = y_0 + \sum_{j=1}^i O_j$ for $i = 1, \ldots, 6$ ($O_i \geq 0$). We restrict our analysis to the *base hazard* logit $f_\theta(y_0 \mid \cdot)$, where $p_\theta(y_0 \mid \cdot) = \sigma(f_\theta(y_0 \mid \cdot))$, as the base risk correlates almost perfectly with cumulative outputs (see Figure 10) and Sybil's performance decays over time (Mikhael et al., 2023; fig. 2). Structurally (Figure 1), Sybil employs a 3D ResNet18 encoder (He et al., 2016) whose features are processed by separate max-pooling and attention branches (Vaswani et al., 2017) to compute final estimates. The attention mechanism is additionally regularized to match expert annotations in the form of segmentation masks. The model was developed using a large-scale cohort of over 28,000 scans from the NLST (National Lung Screening Trial Research Team, 2011).

**Attribution methods.** Most common explanation methods for deep neural networks assign scalar importance scores to input features. Various approaches explored diverse importance definitions and constraints (Simonyan et al., 2014; Bach et al., 2015; Shrikumar et al., 2017; Sundararajan et al., 2017; Selvaraju et al., 2017). Grounded in game theory (Shapley, 1953), SHapley Additive exPlanations (SHAP; Lundberg & Lee, 2017) are a specific instance of *additive* feature attribution methods. SHAP uniquely satisfies the interpretablity axioms of *local accuracy*, *missingness*, and *consistency*. Recent work emphasizes evaluating models on in-distribution inputs, improving saliency maps by respecting the data manifold (Zaher et al., 2024; Ademi et al., 2025; Salek & Enguehard, 2025). In Section A.4, we provide a broader overview of XAI techniques applied to medical imaging models.

**Counterfactual explanations.** Occupying the highest rung of Pearl's causality ladder (Pearl, 2009), visual counterfactual explanations (VCEs) aim to modify a sample in a minimal and semantically meaningful way to alter a model's prediction. Unlike standard attributions, VCEs enable *cause-and-effect* analysis of *what-if* scenarios. To ensure modifications remain on the data manifold, most approaches leverage generative models to approximate the underlying distribution (Jacob et al., 2022; Jeanneret et al., 2022; Augustin et al., 2022; Jeanneret et al., 2023; Augustin et al., 2024; Jeanneret et al., 2024; Sobieski & Biecek, 2024; Sobieski et al., 2025a). Most relevant to this work are *region-constrained* VCEs (Sobieski et al., 2025a), which employ guided image-to-image diffusion bridges (Liu et al., 2023a; Sobieski et al., 2025b) to localize edits in natural images.

## 3. Background

**Notation.** Let $[d] = \{1, \ldots, d\}$ denote feature indices. For any subset $S \subseteq [d]$, we define $\mathbf{x}_S$ as the feature subvector restricted to indices in $S$. Similarly, $f_S(\mathbf{x}_S)$ denotes a component function depending exclusively on $S$.

**Linear Models with Pairwise Interactions.** Linear models are widely recognized as inherently *interpretable* tools for explaining data mechanisms (Hastie, 2009; Rudin, 2019). To enlarge their scope while retaining this *white-box* nature, they are often extended to include *pairwise interactions*. We term $f : \mathbb{R}^d \to \mathbb{R}$ a *linear model with pairwise interactions* (LMPI) if it decomposes as:

$$f(\mathbf{x}) = \underbrace{\beta_0}_{\text{Intercept}} + \underbrace{\sum_i \beta_i x_i}_{\text{Main effects}} + \underbrace{\sum_{1 \leq i < j \leq d} \beta_{ij} x_i x_j}_{\text{Pairwise interactions}}, \quad (1)$$

where $\beta_0, \beta_i, \beta_{i,j} \in \mathbb{R}$ are learnable coefficients.

**n-Shapley Values.** Shapley-based Interaction Indices (SII, Grabisch & Roubens, 1999) extend additive attributions to capture feature dependencies (Sundararajan et al., 2020; Tsai et al., 2023). For a model $f$ and sample $\mathbf{x}$, let the *value function* $v_\mathbf{x}(T)$ represent $f$'s predictions using only features $T \subseteq [d]$. Its *discrete derivative* $\Delta_S v_\mathbf{x}(T) = \sum_{L \subseteq S}(-1)^{|S|-|L|} v_\mathbf{x}(T \cup L)$ isolates the pure interaction effect of $S$ given $T$.

To explain $f(\mathbf{x})$, *n-Shapley Values* (nSV, Bordt & von Luxburg, 2023) approximate the prediction via:

$$f(\mathbf{x}) \approx \sum_{S \subseteq [d], 0 \leq |S| \leq n} \phi_S(\mathbf{x}), \text{ where} \quad (2)$$

$$\phi_S(\mathbf{x}) = \sum_{T \subseteq [d] \setminus S} \frac{(d - |T| - |S|)! \, |T|!}{(d - |S| + 1)!} \Delta_S v_\mathbf{x}(T). \quad (3)$$

For brevity, we omit the dependence of $\phi$ on $\mathbf{x}$ (denoting $\phi_S$) in the remainder of the text.

In this work, we use $n = 2$ with the *interventional* value function $v_{\mathbf{x}}(S) = \mathbb{E}_{\mathbf{z} \sim \mathcal{D}}[f(\mathbf{x}_S, \mathbf{z}_{[d] \setminus S})]$, which recovers the structure of a pairwise interaction model, where $\phi_{\varnothing}$ corresponds to the baseline value. Notably, nSV constitutes the unique least-squares projection of the set function $v_{\mathbf{x}}$ onto the space of additive games of order $n$, effectively approximating the local decision boundary as an LMPI.

To measure the quality of this approximation, we use the coefficient of determination over the feature lattice:

$$R^2 = 1 - \frac{\sum_{S \subseteq [d]} \left( v_{\mathbf{x}}(S) - \hat{v}_{\text{nSV}}(S) \right)^2}{\sum_{S \subseteq [d]} \left( v_{\mathbf{x}}(S) - \bar{v}_{\mathbf{x}} \right)^2}, \quad (4)$$

where $\bar{v}_{\mathbf{x}} = 2^{-d} \sum_{S \subseteq [d]} v_{\mathbf{x}}(S)$ is the mean value and $\hat{v}_{\text{nSV}}(S) = \sum_{K \subseteq S, |K| \leq n} \phi_K(\mathbf{x})$ is the value predicted by the nSV approximation for subset $S$. In XAI, Equation (4) is termed *unweighted local fidelity* (Garreau & Luxburg, 2020).

**Bridging with diffusion.** Generative modeling in computer vision is currently dominated by *diffusion models* (DMs, Ho et al., 2020; Song et al., 2021b). Formally, DMs define a *forward* process (Equation (5)) mapping data $p(\mathbf{x})$ to a latent state over time $t \in [0, 1]$, and a *reverse* generative process (Equation (6)), governed by the SDEs:

$$d\mathbf{x}_t = \mathbf{F}_t \mathbf{x}_t dt + \mathbf{G}_t d\mathbf{w}_t, \quad (5)$$
$$d\mathbf{x}_t = [\mathbf{F}_t \mathbf{x}_t - \mathbf{G}_t \mathbf{G}_t^\top \nabla_{\mathbf{x}_t} \log p(\mathbf{x}_t)]dt + \mathbf{G}_t d\overline{\mathbf{w}}_t, \quad (6)$$

where $\mathbf{F}_t, \mathbf{G}_t$ are the drift and diffusion coefficients, and the *score* $\nabla_{\mathbf{x}_t} \log p(\mathbf{x}_t)$, approximated by the neural network $\mathbf{s}_{\boldsymbol{\xi}}$, drives generation through numerical integration. While standard DMs map to noise, *System-Embedded Diffusion Bridges* (SDB, Sobieski et al., 2025b) generalize the endpoint to a linear measurement $\mathbf{x}' = \mathbf{A}\mathbf{x} + \boldsymbol{\Sigma}^{\frac{1}{2}}\boldsymbol{\varepsilon}$ at $t = 1$. When $\mathbf{A}$ is a binary mask and $\boldsymbol{\Sigma} = \mathbf{0}$, SDB functions as a specialized inpainting model, initializing from masked input $\mathbf{x}'$ to restore missing content. Crucially, this confines the diffusion strictly to unobserved regions, leaving known content unperturbed.

## 4. Methodology

**Motivation.** Current attribution methods face a trade-off: either highlight individual feature importance while staying on the data manifold (Zaher et al., 2024; Ademi et al., 2025; Salek & Enguehard, 2025) or measure feature interactions while evaluating in ambient space (Sundararajan et al., 2020; Tsai et al., 2023; Bordt & von Luxburg, 2023). Conversely, VCEs provide in-distribution causal explanations but fail to isolate specific feature contributions. To capture both feature importance and mutual influence while adhering to the

data manifold, we propose novel attribution methods based on generative interventions. These leverage domain knowledge to simplify the feature space, making them specifically suited for lung cancer risk prediction.

Grounded in the clinical consensus that pulmonary nodules are the primary predictive biomarkers for lung cancer (Callister et al., 2015; MacMahon et al., 2017; Wood et al., 2025), we formulate the following hypothesis:

*Hypothesis* 1. For a given sample, Sybil's decision function can be effectively approximated by an LMPI consisting of a sample-specific background term and a model of main effects and pairwise interactions over pulmonary nodules.

Formally, let $\mathbf{x}$ represent a given patient's CT scan. Hypothesis 1 states that

$$f_{\boldsymbol{\theta}}(y_0 \mid \mathbf{x}) \approx \mu_{\mathbf{x}} + \sum_{i=1}^{N} \phi_i n_i + \sum_{i=1}^{N} \sum_{j > i} \phi_{ij} n_i n_j, \quad (7)$$

where $n_i \in \{0, 1\}$ indicates the presence of the $i$-th nodule, $\phi$ coefficients capture main and pairwise effects, $N$ is the nodule count, and $\mu_{\mathbf{x}}$ represents the sample-specific baseline (nodule-free scan). Notably, for a given $\mathbf{x}$, Equation (2) constitutes the unique least-squares solution to Equation (1), thereby fully determining the coefficients for Equation (7).

Validity of Equation (7) implies Sybil processes nodules as modular semantic units, independent of the background context. Consequently, introducing a nodule should shift predictions in a structurally predictable manner governed by its main and interaction effects. This modularity facilitates nodule-centric counterfactuals, isolating causal impacts without confounding effects from the anatomical background.

**Methodological Gap.** The primary challenge in estimating Equation (7) is the lack of paired counterfactual data, *i.e.*, scans where nodule coalitions are selectively modified. To bridge this, we leverage SDB as a generative proxy for the LDCT distribution. We design a framework utilizing this prior to perform reliable semantic interventions, enabling both the replacement of nodules with healthy tissue and the synthesis of realistic nodules with controlled properties.

### 4.1. Synthetic perturbations

**Distribution blending.** To unify SDB with nodule removal and insertion, we invoke the theoretical result of Verdú (2009) on how distributions *blend* under diffusion.

**Theorem 4.1** (Mismatched estimation, Verdú, 2009)**.** *Let $p(\mathbf{x}^1)$ and $p(\mathbf{x}^2)$ be two probability distributions with time-parameterized evolutions $p(\mathbf{x}_t^1)$ and $p(\mathbf{x}_t^2)$ under the forward process (Equation (5)). The Kullback-Leibler diver-*

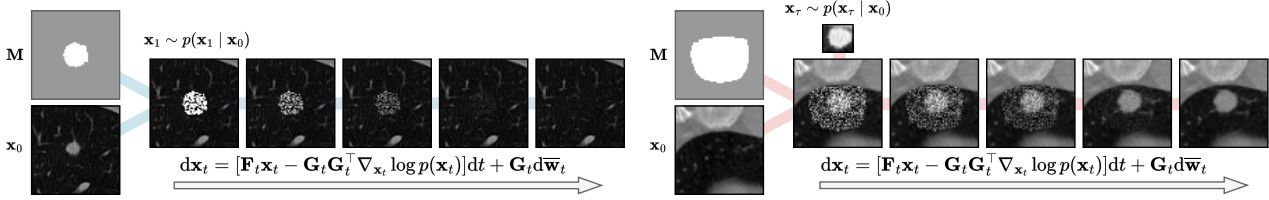

*Figure 2.* 2D visualizations of our nodule removal (**left**) and insertion (**right**) approaches performed on 3D subvolumes of an LDCT scan.

*gence between them decomposes as:*

$$D_{KL}(p(\mathbf{x}_t^1)\|p(\mathbf{x}_t^2)) = D_{KL}(p(\mathbf{x}_0^1)\|p(\mathbf{x}_0^2)) \\ - \frac{1}{2}\int_0^t \mathcal{J}_{\mathbf{D}_\tau}(\tau)\, d\tau, \quad (8)$$

*where $\mathbf{D}_\tau = \mathbf{G}_\tau \mathbf{G}_\tau^T$ and $\mathcal{J}$ is the Relative Fisher Information (RFI):*

$$\mathcal{J}_{\mathbf{D}_\tau}(\tau) = \mathbb{E}_{\mathbf{x}\sim p(\mathbf{x}_\tau^1)}\Big[\big\|\nabla_\mathbf{x}\log p(\mathbf{x}_\tau^1) - \nabla_\mathbf{x}\log p(\mathbf{x}_\tau^2)\big\|_{\mathbf{D}_\tau}^2\Big]. \quad (9)$$

Crucially, Theorem 4.1 states that $p(\mathbf{x}_t^1)$ and $p(\mathbf{x}_t^2)$ become increasingly indistinguishable over time due to the non-negativity of RFI. Consequently, for a score model trained on $p(\mathbf{x}^1)$, diffused samples from a different distribution $p(\mathbf{x}^2)$ become statistically indistinguishable from $p(\mathbf{x}_t^1)$ after some timestep. While implicitly relied upon in image editing (Choi et al., 2021; Meng et al., 2022; Lugmayr et al., 2022; Su et al., 2023; Couairon et al., 2023) and seemingly connected to recent fluctuation theory (Ramachandran et al., 2025), this theoretical justification is rarely explicitly formulated. Here, we leverage the general matrix-valued SDE to link this result directly to SDB.

**Nodule removal.** Let $p(\mathbf{x}^1)$ be the LDCT training distribution for $\mathbf{s}_\xi$ and $p(\mathbf{x}^2)$ be a counterfactual distribution where a specific nodule (within mask $\mathbf{A}$) is replaced with healthy tissue via a *do*-operator (Pearl, 2009). Applying Theorem 4.1 to their forward evolution up to $t = 1$, the distinguishing information within $\mathbf{A}$ vanishes. SDB guarantees this erasure, as its dynamics converge to a state defined entirely by the masked region at $t = 1$. Since pulmonary nodules rarely exceed 0.1% of total lung volume (Horeweg et al., 2014), the score model $\mathbf{s}_\xi$ effectively acts as a healthy tissue prior during reverse sampling, justifying this procedure for in-distribution nodule removal.

**Nodule insertion.** As anatomical anomalies linked to malignancy, nodules are unlikely to be generated by an unconditional model. To insert them, we define $p(\mathbf{x}^2)$ as a copy of $p(\mathbf{x}^1)$ where a nodule from a different patient is transplanted into mask $\mathbf{A}$ and aligned with the new context. Theorem 4.1 implies $p(\mathbf{x}_t^1)$ and $p(\mathbf{x}_t^2)$ become statistically indistinguishable at some specific timestep $t = \tau$. In practice, we simulate $p(\mathbf{x}^2)$ by "copy-pasting" a specific nodule prior to diffusion. The theorem guarantees that at $t = \tau$, the

trained score model treats the diffused state as a valid sample from $p(\mathbf{x}_\tau^1)$, allowing the reverse process to coherently integrate the inserted nodule into the surrounding anatomy.

Examples of both nodule removal and insertion procedures are visualized in Figure 2, while broader literature context is provided in Section A.5.

### 4.2. Constructing the explanations

**Explaining by removing.** Nodule removal enables the LMPI approximation in Equation (7). Let $\mathcal{N} = \{1, \dots, N\}$ denote the set of detected nodules. For any subset $S \subseteq \mathcal{N}$, let $\mathbf{x}_S$ be the scan where nodules in $S$ are preserved and those in $\mathcal{N}\setminus S$ are replaced with healthy tissue. We generate the collection of samples $\mathcal{X} = \{\mathbf{x}_S : S \subseteq \mathcal{N}\}$ covering all possible coalitions of nodules. We pair these with Sybil's responses to form the dataset $D = \{(S, v_\mathbf{x}(S))\}_{\mathbf{x}_S \in \mathcal{X}}$, where $v_\mathbf{x}(S) = f(y_0 \mid \mathbf{x}_S)$. This requires only $N$ SDB removal trajectories to construct the base $\mathbf{x}_\varnothing$ and components for linear recomposition, and $2^N$ evaluations of Sybil, which is computationally inexpensive for a realistic number of lung nodules. We then solve for the coefficients using nSV[1] due to its uniqueness and axiomatic properties (Lundberg & Lee, 2017; Garreau & Luxburg, 2020; Section A). We term this procedure *SHapley Nodule Attribution Profiles* (SHNAP). For pseudocode, see Algorithm 1.

**Explaining by inserting.** Inserting nodules with known properties allows verifying Sybil's reasoning and spatial sensitivity. Given a nodule volume extracted from a patient using a mask $\mathbf{A}$, let $\mathbf{r}$ denote the nodule's volumetric content. We align this content to be centered at a target coordinate $\mathbf{c} = (i, j, k)$ in a different scan $\mathbf{x}$ using our insertion protocol. We assign an attribution score based on the resulting prediction shift:

$$\psi_\mathbf{c} = f(y_0 \mid \mathbf{x}_{\mathbf{c}\leftarrow\mathbf{r}}) - f(y_0 \mid \mathbf{x}), \quad (10)$$

where $\mathbf{x}_{\mathbf{c}\leftarrow\mathbf{r}}$ denotes the scan with the nodule $\mathbf{r}$ inserted at location $\mathbf{c}$. As $f$ represents the logit, Equation (10) corresponds to the log-odds ratio between the intervened and initial states. We term this procedure *Substitutive Nodule Attribution Probing* (SNAP). For pseudocode, see Algorithm 2.

---

[1]Extending the SHAP-IQ package (Muschalik et al., 2024).

# 5. Experiments

**Datasets & Implementation.** We utilize three 3D LDCT datasets: **D1.** NLST (National Lung Screening Trial Research Team, 2011) (approx. 28,000 training and 6,000 test scans) for SDB training; **D2.** LUNA25 (Peeters et al., 2025) (4,069 scans), featuring biopsy-confirmed malignancy or 2-year stability verification for benign cases, for which we construct naive spherical masks from nodule coordinates; and **D3.** iLDCT (243 scans), an internal out-of-distribution testbed with a higher prevalence of severe cases and precise expert radiologist annotations. Lung and lobe segmentations are obtained using the improved TotalSegmentator (Wasserthal et al., 2023) proposed by Chrabaszcz et al. (2025). To manage computational constraints, we train a discrete-time (1000 steps) Schrödinger Bridge (SB) variant of SDB on randomly sampled $64^3$ cubes, generating training masks procedurally via metaballs (Blinn, 1982). We compare SDB reconstruction performance with other baselines for CT synthesis. Both nodule removal and insertion are performed with 100 NFE. For details, see Section C.2.

## 5.1. Validity of synthetic perturbations

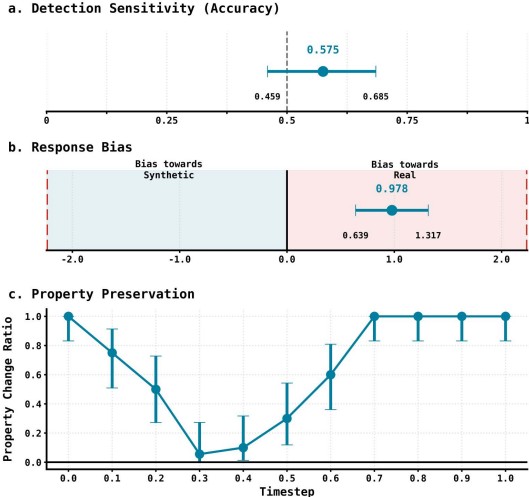

*Figure 3.* Results of an expert study evaluating the realism of nodule removal (**a.**, accuracy; **b.**, response bias) and the preservation of properties during nodule insertion (**c.**).

To ensure our perturbations remain in-distribution, a prerequisite for valid explanations, we conducted a study with two board-certified radiologists.

**Nodule removal.** We performed a blinded evaluation where radiologists assessed 40 3D cubes from NLST (20 real healthy tissue, 20 synthetic removal) in a binary classification task distinguishing real tissue from synthetic. Figure 3 (**a**) shows that their performance is statistically indistinguishable from random guessing (exact binomial test, point estimate 0.575). Furthermore, evaluation of response bias $c$ (Macmillan & Creelman, 2004) reveals a significant ten-

dency to label samples as "real" (Figure 3 (**b**), single-sample Z-test), confirming the high fidelity of the generations.

**Nodule insertion.** The parameter $\tau$ controls the trade-off between preserving source content and aligning with the new context. Radiologists evaluated 110 pairs of scans (original vs. inserted) across $\tau \in \{0, 0.1, \ldots, 1.0\}$. For each pair, they assessed whether the synthetic nodule exhibited perceptible deviations from the reference regarding structural properties, malignancy characteristics, or background alignment. Figure 3 (**c**) presents 95% CIs from an exact binomial test for each $\tau$, illustrating the transition from artifact-heavy naive insertion ($\tau \to 0$) to excessive deviation ($\tau \to 1$). The optimal balance is achieved at $\tau = 0.3$, where nodules consistently preserve source properties without visual artifacts. We use this value for all subsequent experiments.

## 5.2. Validity of the explanations

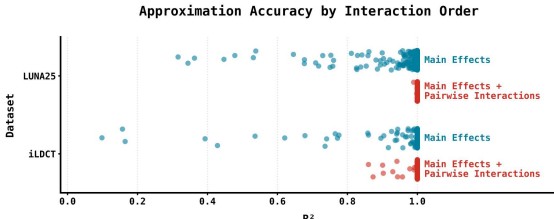

*Figure 4.* Accuracy of approximating Sybil as an LMPI over pulmonary nodules across two datasets, including both first- and second-order effects.

**Approximation accuracy.** Establishing that nodule removal results in indistinguishable healthy tissue confirms that SHNAP explanations stem from in-distribution perturbations. This allows verifying Hypothesis 1: that Sybil is effectively an LMPI. We compute SHNAP on the entire LUNA25 test split and iLDCT, evaluating the fit via $R^2$ (Equation (4)). Figure 4 shows results for main effects and pairwise interactions. Main effects alone suffice for a perfect fit in the majority of cases, evidenced by the collapsed interquartile range at $R^2 \approx 1$. The outlier tail is almost entirely eliminated by adding pairwise interactions. Remaining failures typically correspond to rare, anomalously large nodules where SDB struggles with reconstruction; a limitation addressable by training on larger volumes. Overall, these results strongly support Hypothesis 1.

**Counterfactual tractability.** While Figure 4 validates SHNAP, it also reinforces the reliability of SNAP. Confirming that Sybil functions as an LMPI implies it processes nodules additively. Consequently, synthetic insertion adheres to this same mechanism, ensuring mathematically consistent counterfactuals and transparent attributions.

**Stability.** To evaluate robustness, Figure 13 displays the density of standard deviations for SHNAP attributions across 5

independent runs, where stochasticity arises from the random initial state of the reverse diffusion process. The values concentrate heavily around zero, indicating a systematic response to healthy tissue imputation and minimizing the risk of adversarial artifacts. We also compare SHNAP to naive perturbations (Section C.6); the latter reveals that out-of-distribution inputs result in unstable attributions, highlighting the necessity of generative interventions.

### 5.3. Opening Sybil's black box with SHNAP

Building on the foundation that Sybil can be effectively represented as a LMPI over pulmonary nodules, we proceed to use SHNAP to dissect its decision-making mechanisms.

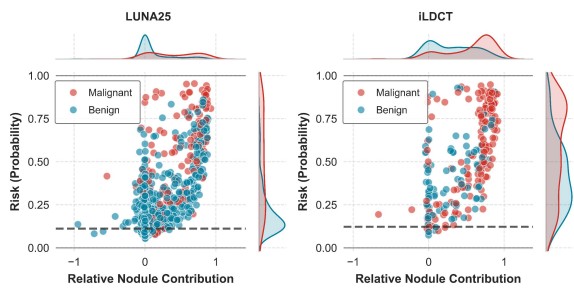

*Figure 5.* Comparison of risk predicted by Sybil and relative nodule contributions across two datasets.

**When pulmonary nodules matter most.** A unique advantage of our decomposition (Equation (7)) is the separation of nodule-specific effects from the anatomical background $\mu_\mathbf{x}$. We leverage this to move beyond manual auditing, defining the *Relative Nodule Contribution* (RNC) as $\text{RNC}(\mathbf{x}) = \frac{\sigma(f(y_0|\mathbf{x})) - \sigma(\mu_\mathbf{x})}{\sigma(f(y_0|\mathbf{x}))}$ to detect cases where model reasoning may be flawed. Here, $\sigma(\cdot)$ converts the logits $f$ to probabilities. To interpret risk levels, we set a decision threshold on the test split enforcing $\geq 95\%$ sensitivity, reflecting the clinical priority of minimizing false negatives.

Figure 5 shows RNC densities for the LUNA25 test split (**a.**) and iLDCT (**b.**). On LUNA25, the distribution for benign cases is highly right-skewed. Intuitively, low nodule contribution aligns with clinical reasoning: absent pathology, risk estimates should rely on background markers like emphysema. The long positive tail thus signals potential flaws where benign nodules erroneously drive high risk. Conversely, confirmed cancer cases in this dataset exhibit a bimodal distribution with a significant mode near 0, revealing that Sybil frequently ignores known lesions in favor of background context. In iLDCT (**b.**), characterized by higher severity, Sybil's focus shifts even more toward nodules, highlighting its capacity to identify ominous lesions. Yet, this increased sensitivity coincides with a larger tail of erroneous high attributions for benign findings.

**Understanding false positives.** Figure 6 (**a.**) visualizes Sybil's reasoning on a false positive where nodules alone

drive 60% of the risk. Among 5 nodules, Sybil focuses almost entirely on a large, dense lesion in the upper right lung (**R3**), ignoring two smaller ones in the left lung (**R1**, **R2**). This behavior mimics a radiologist's caution: **R3** is highly suspicious, while the global spread of smaller nodules suggests prior infection rather than malignancy. Sybil's caution overshoots the ground truth (which benefits from multi-year stability data unavailable to the model), but the decision logic is understandable.

Figure 6 (**b.**) reveals another false positive where 78% of the risk stems from nodules, dominated by a single lesion (**R2**) with a dense center and ground-glass opacity. While this morphology can indicate invasive adenocarcinoma, it also typifies pneumonia. Sybil's high risk estimate reflects justifiable caution. However, inspection of three-year longitudinal follow-up (Figures 16 and 17) reveals dangerous instability: the model's focus shifts to a different nodule (**R1**), neglecting **R2**. While **R1** is suspicious, the reasoning *shift* exposes inconsistency, rendering the model untrustworthy despite the initial defensible prediction.

**False negatives reveal flawed reasoning.** Figure 6 (**c.**) presents a malignant case where Sybil severely underestimates risk. Although it correctly ranks the spiculated nodule (**R1**) above the pleural one (**R2**), the total nodule contribution is merely 11%. Crucially, both are subpleural, a common site for adenocarcinoma, yet the model fails to assign them significant weight, indicating a potential suppression mechanism. Figures 18 and 19 further illustrate this systematic failure, where Sybil repeatedly ignores a malignant pleural nodule across consecutive yearly scans.

**Right for wrong reasons.** Figure 6 (**d.**) demonstrates that correct predictions do not imply correct reasoning. In this malignant case, Sybil assigns negative attribution to the actual nodules, effectively treating them as evidence *against* cancer. It is only their interaction that negates this effect, rendering their total contribution as effectively zero. Consequently, the correct high-risk prediction is driven almost entirely by background features, while the primary evidence, distinct malignant patterns, is ignored.

**Beyond pulmonary nodules.** SHNAP provides rigorous insights into reasoning about nodules, but model developers must also uncover reliance on out-of-distribution patterns like artifacts and spurious correlations (Biecek & Samek, 2024). To shift perspective, we propose *generalized* SHNAP (gSHNAP), which views Sybil as an LMPI over arbitrary image regions by replacing nodule indicators with region indicators $z_i$. This leverages SDB to replace specific areas with their "most likely" synthesized counterparts, effectively scrubbing rare artifacts. A key challenge is choosing these regions. While prior work on counterfactuals employs post-hoc attributions to detect important areas (Sobieski et al., 2025a), we leverage Sybil's own attention mechanism. By

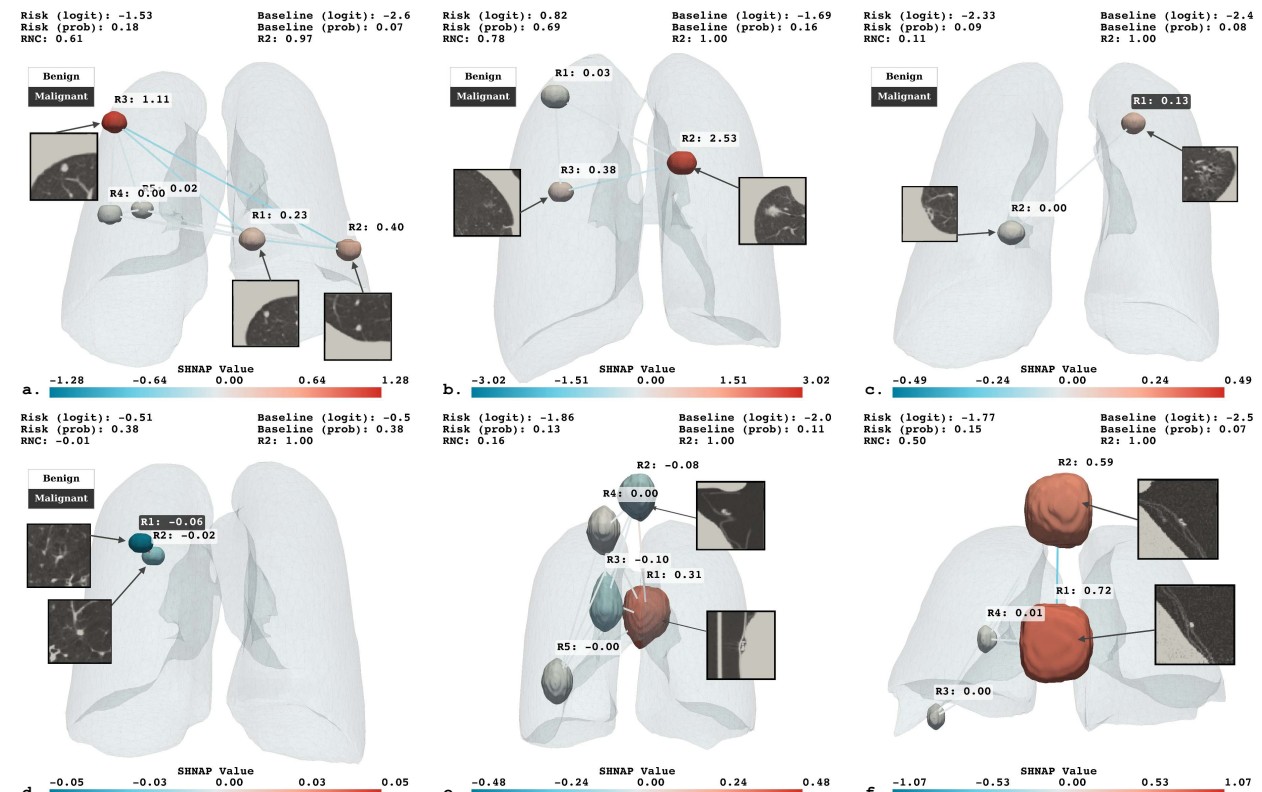

*Figure 6.* SHNAP explanations for Sybil predictions across various patients. Each subplot indicates the initial prediction, the baseline term, local fidelity ($R^2$), and RNC. 3D renders display attribution magnitudes, with labels for benign and malignant nodules.

binarizing attention maps, we directly probe the role of salient sub-volumes in the decision process.

**Discovering clinically unjustified artifacts.** We explore LUNA25 and iLDCT using the relative contribution of attention-based regions (see Section C.8). Our primary findings reveal that attention often points to regions with non-zero gSHNAP attributions that do not overlap with pulmonary nodules. Figure 6 (**e.**) displays 5 distinct attention-based regions, 2 of which are located *outside* the patient's body. Dominant **R1** points to an artifact resembling metal snaps used to close a hospital gown, pressed between the patient's skin and the scanner table. Moreover, while contributing negatively to the prediction, **R2** points to a cross-section of the patient's chin (mandible), suggesting that Sybil treats it as a large, solitary mass resembling a benign nodule.

Figure 6 (**f.**) presents a critical failure in a benign case where 50% of the predicted risk stems from the joint influence of two symmetric objects outside the patient's body (**R1** and **R2**). Upon closer inspection, these regions reveal ECG electrodes attached to the chest skin. While likely used for cardiac synchronization to avoid motion blur, Sybil appears to correlate these leads with critical conditions, erroneously inflating the risk. Such spurious correlations, akin to "hospital tag" shortcuts in other domains (Baniecki et al., 2025;

Mishra & Celebi, 2016), make deployment unacceptable. Further examples in Section C.7 show reliance on other clinically unjustified artifacts, such as thyroid goiters or metallic objects outside the body.

**Influential regions are sparse.** The richness of findings based on SHNAP and gSHNAP raises the question of whether Sybil is sensitive to perturbations in *any* arbitrary region. To ablate this, we generate gSHNAP explanations for random regions within the lung volume that are disjoint from both nodules and attention-based regions. Figure 23 shows that their importance is highly concentrated around zero, confirming that influential regions are sparse and specific, rather than the model simply reacting to random changes.

### 5.4. Opening Sybil's black box with SNAP

**Revealing local failures.** Figure 7 visualizes a high-resolution SNAP map of over 5,000 insertions of a malignant nodule in a single patient. Three specific sites highlight Sybil's variable sensitivity: **N1** (lung base) triggers a strong response, indicating correct identification; **N2** (near the pleura) shows a weaker response, suggesting distraction; and **N3** reveals a complete failure to detect the nodule, likely lost by Sybil in surrounding tissue. These examples demonstrate that while Sybil's risk estimates are generally smooth, sensitivity is not uniform and can fail locally.

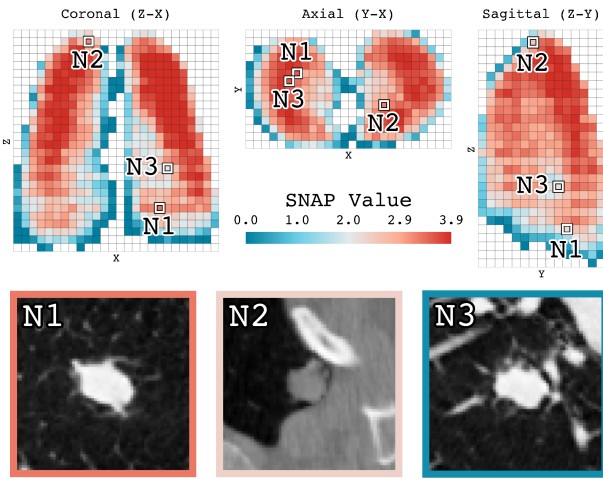

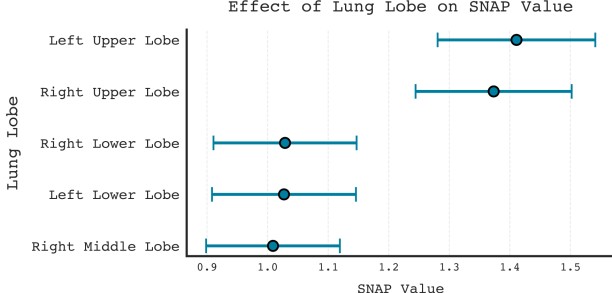

*Figure 7.* SNAP attribution map across three anatomical views, featuring example nodule insertions. Each nodule's value range is optimized for visual clarity.

**Revealing global anatomical biases.** To explore this at scale, we generated 240 patient-nodule combinations (20 nodules × 12 scans), performing ≈ 900 insertions per combination (total ≈ 200, 000 samples), see Section C.10. We aggregated attributions by lung lobe and performed a two-way ANOVA. Results show significant main effects for patient identity ($p < 0.001$) and lobe class ($p < 0.001$), confirming that some patients naturally trigger higher risks and that anatomical sensitivity varies distinctively. Crucially, the interaction between patient and lobe was insignificant ($p \approx 1.0$), indicating that lobar bias is a *global characteristic* of Sybil, independent of patient-specific variation. To determine directionality, we performed a post-hoc Tukey's HSD test (Tukey, 1949, Figure 8). Attribution in the left and right upper lobes was significantly higher than in the middle/lower lobes ($p \leq 0.009$). This aligns perfectly with clinical gold standards (*e.g.*, PanCan, Mayo models (Swensen et al., 1997; McWilliams et al., 2013)), which identify upper lobe location as a significant predictor of malignancy. Furthermore, the right middle lobe was indistinguishable from the lower lobes ($p - \text{value} \approx 1.0$), forming a single inferior low-attribution zone, also consistent with literature. Finally, Sybil correctly ignores laterality (left vs. right), treating them as equivalent.

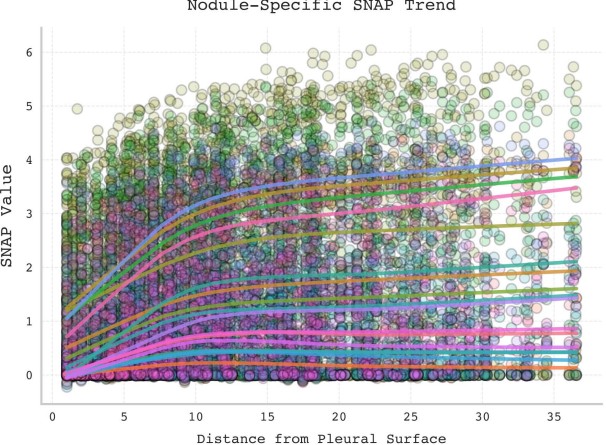

*Figure 8.* Average SNAP across 240 patient–nodule combinations, stratified by anatomical lung lobe.

**Radial sensitivity bias.** While lobar biases align with medical knowledge, SNAP also reveals misalignment. Building on our SHNAP findings of false negatives near the pleura (Figure 6), we identified a systematic failure to detect peripheral nodules. A linear regression predicting attribution from the distance-to-pleura yielded a significant positive coefficient ($p < 0.001$), indicating that sensitivity drops near the lung boundary. While distance alone explained little variance ($R^2 = 0.071$), adding interactions with nodule identity raised this to $R^2 = 0.455$, confirming a heterogeneous effect where malignant nodules are progressively attenuated near the boundary while benign ones remain robust. For an example trend, see Figure 9 and its high-resolution version in Figure 30.

*Figure 9.* SNAP attribution values and corresponding trends across several nodules inserted into the same patient as a function of their distance from the pleural surface.

We hypothesize this *radial sensitivity bias* stems from zero-padding in 3D convolutions, a known cause of activation attenuation at boundaries (Alsallakh et al., 2021). This structural vulnerability is clinically concerning: adenocarcinoma, the most common lung cancer subtype, predominantly arises in the periphery (Travis et al., 2011), making this blind spot a critical failure mode.

### 5.5. Analyzing the background effect

The decomposition in Equation (7) enables granular analysis by isolating the background effect (the nodule-removed scan). While not our primary focus, analyzing this baseline offers complementary insights. Preliminary mixed-effects regression links the baseline term $\mu_{\mathbf{x}}$ to patient age, showing a positive trend on LUNA25 ($p = 0.05$) and iLDCT ($p = 0.027$). This implies Sybil likely infers age from global cues like bone density, embedding it into the baseline risk. These findings suggest that global features hold strong predictive power, justifying future work to disentangle them from localized pathologies.

## 5.6. Ablation studies and further details

Due to space constraints, extensive methodological details, further related works, and comprehensive experimental results are relegated to the Appendix. This includes an expanded description of the generative inpainting architecture and its training process, exhaustive robustness and stability ablations for the proposed framework, quantitative comparisons against classical attribution methods, and additional qualitative case studies alongside expert validation results. We also provide a general implementation of S(H)NAP within a unified codebase [2].

## 6. Discussion and limitations

We audited Sybil, a deep-learning lung cancer risk model, through the S(H)NAP framework. While checking for correct predictions is standard, our principled analysis of decision mechanisms corroborated Sybil's discriminative power and exposed critical reasoning misalignments. Our approach answers recent calls to prioritize strict model verification over purely observational studies (Biecek & Samek, 2025).

A primary limitation is the reliance on partially synthetic data, introducing the risk of generative artifacts. We mitigated this via a blinded expert study, although the pursuit of provably robust counterfactuals remains an active frontier (Zaher et al., 2026). Crucially, S(H)NAP is model-agnostic, relying exclusively on input-output pairs. This allows the framework to be applied to arbitrary systems, including proprietary commercial models like Optellum (Massion et al., 2020), highlighting the immense value of domain-specific, object-level explanations.

## Impact Statement

This paper advances the field of model explainability within the high-stakes domain of lung cancer risk prediction. While machine learning models demonstrate significant potential for automating lung cancer screening, our work highlights the necessity of rigorous auditing prior to clinical deployment. We demonstrate that reliance on clinically unjustified artifacts or spatial biases can lead to unreliable predictions. By providing a framework for generative interventions, this work contributes to the ethical development of AI in medicine, promoting transparency and safety. The societal implications include fostering trust in automated systems that patients and professionals rely on, ensuring that diagnostic assessments are both interpretable and accountable.

[2]https://github.com/sobieskibj/auditing_sybil

## Acknowledgments

Work on this project is financially supported by the Polish National Science Centre PRELUDIUM BIS grant No. 2023/50/O/ST6/00301 and the Foundation for Polish Science (FNP) grant 'Centre for Credible AI' No. FENG.02.01-IP.05-0058/24.

The computational resources for this work were provided by the Laboratory of Bioinformatics and Computational Genomics and the High Performance Computing Center of the Faculty of Mathematics and Information Science, Warsaw University of Technology. We also gratefully acknowledge Poland's High-performance Infrastructure PLGrid ACC Cyfronet AGH for providing computer facilities and support within computational grant no. PLG/2025/018330.

Finally, we express our gratitude to Hanna Piotrowska for her valuable assistance in preparing the visualizations and to Zuzanna Matuszewska for the detailed annotations of initial results.

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

## A. Extended related works

### A.1. SHAP and nSV.

While various methods exist to estimate the coefficients of an additive model, SHAP (Lundberg & Lee, 2017) provides a unique solution that satisfies three distinct axiomatic properties. **P1.** *Local accuracy* guarantees that the sum of the feature attributions matches the original model output when all features are present. **P2.** *Missingness* requires that if a feature is absent from the input, its assigned attribution must be zero. **P3.** *Consistency* ensures that if a model changes such that the marginal contribution of a feature increases or remains the same across all possible contexts, its attribution should not decrease. These properties ensure that the resulting explanations are faithful and allow for reliable comparison between

different models.

Building on this additive framework, we extend univariate attribution to higher-order interactions through nSV (Bordt & von Luxburg, 2023). While traditional SHAP decomposes the model output into individual feature contributions, nSV captures interaction effects between groups of features explicitly. This effectively represents the model as a Generalized Additive Model with interactions, where the output is the sum of main effects and their combinations. This decomposition is uniquely defined by a set of generalized axiomatic properties. **P1.** *Efficiency* ensures that the total sum of all interaction effects, including a constant baseline value, perfectly recovers the full model output. **P2.** *Symmetry* dictates that if two features contribute identically to every possible combination of other features, their respective interaction attributions must be equal. **P3.** *Null interaction* requires that if a feature adds no value to any possible grouping, all interaction terms involving that feature are assigned an attribution of zero. Together, these axioms provide a theoretically grounded method for auditing complex models, ensuring that interaction effects are neither arbitrarily assigned nor omit critical dependencies.

### A.2. Generalized Additive Models.

Linear models stand at the core of statistics, offering a plethora of practical tools for explaining underlying data mechanisms, though often at the cost of predictive capacity (Hastie, 2009). Their simplicity makes them widely recognized as inherently *interpretable* (Rudin, 2019). Generalized Additive Models (GAMs, Hastie & Tibshirani, 1986) combine their white-box nature with the flexibility of non-linear models. Formally, a GAM of order $n$ is defined as a function $f : \mathbb{R}^d \to \mathbb{R}$ that can be written in the form

$$f(\mathbf{x}) = \sum_{S \subseteq [d], 0 \leq |S| \leq n} f_S(\mathbf{x}_S), \tag{11}$$

where $[d] = \{1, \ldots, d\}$ and $S = \{s_1, \ldots, s_k\} \subseteq [d], |S| = k$ is used to index both the data features $\mathbf{x}_S = (x_{s_1}, \ldots, x_{s_k})$ and the component functions acting solely on their subsets $f_S(\mathbf{x}_S) = f_{s_1, \ldots, s_k}(x_{s_1}, \ldots, x_{s_k})$.

### A.3. Counterfactual explanations.

Standing at the top of Pearl's causality ladder (Pearl, 2009), visual counterfactual explanations (VCEs) aim to modify a given sample in a minimal and semantically meaningful way that also changes the prediction of a decision model to a target one, allowing to explore its behavior under *what-if* scenarios. To guide the image modification along the data manifold, most approaches make use of a separate generative model as an approximation to the data distribution. Commonly, finding VCEs is formulated as solving the optimization problem

$$\arg \min_{\hat{\mathbf{x}}} s(\mathbf{x}, \hat{\mathbf{x}}) - \lambda \cdot p_{\boldsymbol{\theta}}(y' \mid \hat{\mathbf{x}}), \tag{12}$$

where $s(\mathbf{x}, \hat{\mathbf{x}})$ is the *semantic* distance between the true $\mathbf{x}$ and modified $\hat{\mathbf{x}}$, $p_{\boldsymbol{\theta}}$ is the $\boldsymbol{\theta}$-parameterized decision model, $y'$ is the target decision and $\lambda > 0$ controls the trade-off between the decision change and semantic distance.

Approaches for solving Equation (12) may be categorized as either white- or black-box. The former differentiate through Equation (12) directly, utilizing access to weights and computational graph of the decision model (Jacob et al., 2022; Jeanneret et al., 2022; Augustin et al., 2022; Jeanneret et al., 2023; Augustin et al., 2024; Sobieski et al., 2025a). The latter assume more general scenario of access limited to only input-output interaction, making the optimization more difficult. This can be performed either explicitly by approximating Equation (12) (Sobieski & Biecek, 2024; Jeanneret et al., 2024) or implicitly by observing its behavior under modification of various semantic attributes (Kazimi et al., 2024).

### A.4. XAI in medical imaging.

Prior approaches for understanding the decision-making mechanisms of predictive models trained on medical images (e.g., CT, MRI, X-ray) were largely limited to the direct application of standard XAI techniques (Ahmed et al., 2026). In the context of Grad-CAM (Selvaraju et al., 2017), Panwar et al. (2020), Moujahid et al. (2022), Musthafa et al. (2024), and Hammad et al. (2025) apply it to localize pathologies in lung CT, chest X-rays, and brain MRI, while Panboonyuen (2026) critically analyzes its reliability. Nahiduzzaman et al. (2024) and Rai et al. (2025) use SHAP to construct lung cancer detection frameworks. Eitel et al. (2019) apply Layer-Wise Relevance Propagation (LRP, Bach et al., 2015) to MRI-based multiple sclerosis diagnosis. Sigut et al. (2023) approximate CNN predictions using surrogate models for glaucoma diagnosis from fundus images. Elbouknify et al. (2023), Islam et al. (2025), and Navaratnarajah et al. (2025) combine multiple XAI methods to obtain a more comprehensive view of the diagnosis across different modalities. In a more

sophisticated fashion, Mertes et al. (2022), Arora & Lee (2024), and Navaratnarajah et al. (2025) analyze counterfactual scenarios in mammography and neuroimaging contexts.

Crucially, while prior approaches reuse or adapt standard XAI techniques, they do not consider explanations tailored specifically to the 3D medical imaging domain, which presents its own set of challenges and characteristic predictive features, a gap that is the focus of our work. Moreover, our work goes beyond standard approaches by leveraging recent advances in generative modeling to reinterpret the decision-making process within a simplified feature space composed of pulmonary nodules, upon which we construct explanations using state-of-the-art methods for Shapley-based Interaction Indices (Grabisch & Roubens, 1999).

### A.5. Synthesis, nodule removal and insertion.

3D medical imaging poses significant computational challenges for synthesis due to large image sizes, complex anatomical structures, and the importance of high-frequency details. The focus of recent approaches has shifted almost entirely to diffusion models (DMs) due to their natural advantages in image generation quality (Dhariwal & Nichol, 2021). MedicalDiffusion (Khader et al., 2023) was one of the first approaches to successfully apply the DM framework to 3D medical image synthesis. Subsequent models, such as Lung-DDPM (Jiang et al., 2025a), Lung-DDPM+ (Jiang et al., 2025b), and Med-DDPM (Dorjsembe et al., 2024), extended the standard framework with semantic-layout guidance, allowing for controlled generation with faster inference times. For the same purpose, GEM-3D (Zhu et al., 2024) utilized conditional diffusion models (Batzolis et al., 2021). LAND (Oliveras et al., 2025) applied latent DMs (Rombach et al., 2022) to generate high-quality 3D chest CT scans from anatomical masks. In the domain of image forensics, M3DSYNTH (Zingarini et al., 2024) introduced a DM capable of shrinking and enlarging existing pulmonary nodules. Kim et al. (2025a) combined Generative Adversarial Networks and Brownian Bridge DMs to allow for the removal, synthesis, and modification of nodule shape and texture based on radiomics features. MAISI (Guo et al., 2025) combined latent DMs with a volume compression network to generate high-resolution CT images using various conditioning mechanisms, while MAISIv2 (Zhao et al., 2026) extended this framework to rectified flows (Liu et al., 2023b).

In contrast to our work, prior approaches either focused on the pure synthesis of entire volumes without granular control over pulmonary nodules (Jiang et al., 2025a;b; Dorjsembe et al., 2024; Zhu et al., 2024) or required annotated data during training to enable nodule removal and synthesis (Oliveras et al., 2025; Zingarini et al., 2024; Kim et al., 2025a; Guo et al., 2025; Zhao et al., 2026). Crucially, none of these works allows for both nodule removal *and* the insertion of an existing, extracted nodule that remains aligned with the new context and preserves its original properties, all while relying solely on unannotated CT scans for training.

### A.6. Diffusion models.

In recent years, generative modeling of visual data has been largely dominated by diffusion models (Ho et al., 2020; Dhariwal & Nichol, 2021), with a particular focus on their continuous-time formulation (Song & Ermon, 2019; Song et al., 2021b). Inspired by the physics of non-equilibrium thermodynamics (Sohl-Dickstein et al., 2015), these models learn to reverse a noising process through iterative denoising. Formally, let $p(\mathbf{x}_0)$ represent the data and $p(\mathbf{x}_1)$ the prior distribution respectively. Diffusion models connect the two by considering $p(\mathbf{x}_t)$ for $t \in [0, 1]$ as the distribution induced by the *forward process* evolving according to a stochastic differential equation (SDE)

$$\mathrm{d}\mathbf{x}_t = \mathbf{F}_t \mathbf{x}_t \mathrm{d}t + \mathbf{G}_t \mathrm{d}\mathbf{w}_t, \tag{13}$$

with a *linear* drift term $\mathbf{F}_t \mathbf{x}_t$, a time-dependent matrix $\mathbf{F}_t \in \mathbb{R}^{d \times d}$ and matrix-valued diffusion coefficient $\mathbf{G}_t$. Equation (5) allows for mapping the data distribution to an easy-to-sample prior, *e.g.*, an isotropic zero-mean Gaussian. To invert this mapping, a *reverse process* is considered

$$\mathrm{d}\mathbf{x}_t = [\mathbf{F}_t \mathbf{x}_t - \mathbf{G}_t \mathbf{G}_t^\top \nabla_{\mathbf{x}_t} \log p(\mathbf{x}_t)]\mathrm{d}t + \mathbf{G}_t \mathrm{d}\overline{\mathbf{w}}_t, \tag{14}$$

where $\nabla_{\mathbf{x}_t} \log p(\mathbf{x}_t)$ is the *score function*. The inversion is understood as the marginals $p(\mathbf{x}_t)$ of Equation (5) and Equation (6) being equal. As $\nabla_{\mathbf{x}_t} \log p(\mathbf{x}_t)$ is unavailable in most practical scenarios, a neural network $\mathbf{s}_{\boldsymbol{\xi}}$ is trained to regress $\nabla_{\mathbf{x}_t} \log p(\mathbf{x}_t \mid \mathbf{x}_0)$ marginalized over samples $\mathbf{x}_0$, leading to an asymptotically unbiased estimate of $\nabla_{\mathbf{x}_t} \log p(\mathbf{x}_t)$ (Hyvärinen, 2005). The choice of the drift and diffusion coefficients $\mathbf{F}_t, \mathbf{G}_t$ determines the dynamics of the mapping between the Gaussian and data distribution. A large majority of works restricted them to simple scalar functions, *i.e.*, $\mathbf{F}_t = f_t \mathbf{I}, \mathbf{G}_t = g_t \mathbf{I}$ for some time-dependent $f_t, g_t \in \mathcal{C}([0, 1], \mathbb{R})$ (Ho et al., 2020; Song et al., 2021b; Dhariwal & Nichol, 2021; Song et al., 2021a).

## A.7. Diffusion bridges.

An important extension of diffusion models are the so-called *diffusion bridges*, which generalize to mappings between arbitrary distributions for $p(\mathbf{x}_0)$ and $p(\mathbf{x}_1)$. Initial works arrived at such mappings by considering more general SDEs with additional dependence on the "endpoints" from $p(\mathbf{x}_1)$ (Luo et al., 2023; Liu et al., 2023a; Yue et al., 2024; Zhou et al., 2024; Li et al., 2023; Liu et al., 2023a; Kim et al., 2024; De Bortoli et al., 2024). Diffusion bridges offer a principled data-based approach to solving *inverse problems*, *i.e.*, tasks of the form

$$\mathbf{y} = \mathcal{A}(\mathbf{x}) \tag{15}$$

with $\mathbf{x}$ being the true signal, $\mathcal{A}$ a (possibly stochastic) measurement operator, $\mathbf{y}$ the resulting measurement, with the goal of retrieving the true $\mathbf{x}$ from the observed $\mathbf{y}$. Attempting to solve an inverse problem with a diffusion bridge can be generally described as setting $p(\mathbf{x}_0) = p(\mathbf{x}), p(\mathbf{x}_1) = p(\mathbf{y})$ and learning its corresponding SDE using pairs $(\mathbf{x}, \mathbf{y})$ sampled from the joint $p(\mathbf{x}, \mathbf{y})$ induced by Equation (15). An important special case of Equation (15), covering many practical applications, is the *linear Gaussian* form

$$\mathbf{y} = \mathbf{A}\mathbf{x} + \boldsymbol{\Sigma}^{\frac{1}{2}}\boldsymbol{\varepsilon}, \tag{16}$$

where $\mathbf{A} \in \mathbb{R}^{m \times n}$ for $m, n \in \mathbb{N}$, $\boldsymbol{\varepsilon} \sim \mathcal{N}(\mathbf{0}_m, \mathbf{I}_{m \times m})$ and $\boldsymbol{\Sigma} \in \mathbb{R}^{m \times m}_{\geq 0}$ is a positive semi-definite covariance matrix. Notably, (Sobieski et al., 2025b) showed that by utilizing the matrix-valued formulation of Equations (5) and (6), where the drift and diffusion coefficients do not reduce to scalar functions, one may embed the system from Equation (16) directly into the SDE, leading to a Markovian diffusion bridge between $p(\mathbf{x})$ and $p(\mathbf{y})$ termed SDB.

For the purpose of this work, consider the specific form of Equation (16) for the no-noise inpainting problem with $\mathbf{y} = \mathbf{A}\mathbf{x}$, where $\mathbf{A}$ is the masking operator. Let $\mathbf{F}_t = \frac{\mathrm{d}}{\mathrm{d}t} \log \alpha_t (\mathbf{I} - \mathbf{A}^+\mathbf{A})$, $\mathbf{G}_t \mathbf{G}_t^\top = \left( \frac{\mathrm{d}\beta_t}{\mathrm{d}t} - 2\beta_t \frac{\mathrm{d}}{\mathrm{d}t} \log \alpha_t \right) (\mathbf{I} - \mathbf{A}^+\mathbf{A})$ with $\mathbf{A}^+$ denoting the pseudoinverse of $\mathbf{A}$ and scalar functions $\alpha_t, \beta_t$ such that $\alpha_0 = 1, \alpha_1 = 0$ and $\beta_t \to 0$ uniformly for all $t$. These lead to SDB (SB), the Schrödinger Bridge (SB) variant of SDB, which solves the optimal transport plan (Mikami, 2004) in the null space of $\mathbf{M}$, following the main theoretical result from (Liu et al., 2023a), while entirely preserving the range space component.

# B. Extended methodology

## B.1. Pseudocode

We include the pseudocode for both SHNAP and SNAP in Algorithm 1 and Algorithm 2 respectively.

---

**Algorithm 1** SHNAP: SHapley Nodule Attribution Profiles

---

1: **Input:** Original scan $\mathbf{x}$, logit function $f_{\boldsymbol{\theta}}$, score model $s_{\boldsymbol{\xi}}$, nodule set $\mathbf{N}$, nodule masks $\{\mathbf{A}_i\}_{i=1}^{|\mathbf{N}|}$
2: **Output:** Baseline $\mu_{\mathbf{x}}$, main effects $\phi_i$, pairwise interactions $\phi_{ij}$
3: **for** each $i \in \mathbf{N}$ **do**
4:     $\mathbf{x}_{1,i} \leftarrow$ ForwardDiffusion$(\mathbf{x}, t = 1, \text{mask} = \mathbf{A}_i)$            # Remove nodule $i$
5:     $\tilde{\mathbf{x}}_i \leftarrow$ ReverseSampling$(\mathbf{x}_{1,i}, s_{\boldsymbol{\xi}}, t = 1 \to 0)$        # Inpaint healthy tissue for $i$
6: **end for**
7: $\mathbf{x}_\emptyset \leftarrow$ Recompose$(\mathbf{x}, \{\tilde{\mathbf{x}}_i\}_{i \in \mathbf{N}})$         # Construct nodule-free baseline
8: $\mathcal{D} \leftarrow (\emptyset, f_{\boldsymbol{\theta}}(y_0 \mid \mathbf{x}_\emptyset))$         # Initialize dataset
9: **for** each subset $S \subseteq \mathbf{N}$ **do**
10:     $\mathbf{x}_S \leftarrow$ Recompose$(\mathbf{x}_\emptyset, \{\mathbf{x}_j\}_{j \in S})$         # Reinsert original nodules from $S$
11:     $\mathcal{D} \leftarrow \mathcal{D} \cup \{(S, f_{\boldsymbol{\theta}}(y_0 \mid \mathbf{x}_S))\}$         # Form dataset for nSV
12: **end for**
13: Solve for $\mu_{\mathbf{x}}, \phi_i, \phi_{ij}$ with nSV         # Least-squares projection
14: **return** $\mu_{\mathbf{x}}, \phi_i, \phi_{ij}$

---

## B.2. Pathways to Bias Mitigation

While the primary scope of this work focuses on rigorously auditing and diagnosing reasoning misalignments, the generative nature of the S(H)NAP framework naturally introduces actionable strategies for downstream bias mitigation. Although full implementation remains for future work, we outline two primary pathways:

---

**Algorithm 2** SNAP: Substitutive Nodule Attribution Probing

---

 1: **Input:** Target scan $\mathbf{x}$, source nodule content $\mathbf{r}$, mask $\mathbf{A}$, target coordinate $\mathbf{c}$, score model $s_{\boldsymbol{\xi}}$, logit function $f_{\boldsymbol{\theta}}$
 2: **Output:** Attribution score $\psi_{\mathbf{c}}$
 3: **Parameter:** Blending timestep $\tau = 0.3$
 4: Translate $\mathbf{r}$ and $\mathbf{A}$ to coordinate $\mathbf{c}$        # Spatial alignment
 5: $\mathbf{x}_{paste} \leftarrow \text{CopyPaste}(\mathbf{x}, \mathbf{r}, \mathbf{A}, \mathbf{c})$      # Simulate distribution $p(\mathbf{x}^2)$
 6: $\mathbf{x}_{\tau} \leftarrow \text{ForwardDiffusion}(\mathbf{x}_{paste}, t = \tau)$      # Diffuse up to step $\tau$
 7: $\mathbf{x}_{\mathbf{c} \leftarrow \mathbf{r}} \leftarrow \text{ReverseSampling}(\mathbf{x}_{\tau}, s_{\boldsymbol{\xi}}, t = \tau \rightarrow 0)$      # Align with the background
 8: $\psi_{\mathbf{c}} = f_{\boldsymbol{\theta}}(y_0 \mid \mathbf{x}_{\mathbf{c} \leftarrow \mathbf{r}}) - f_{\boldsymbol{\theta}}(y_0 \mid \mathbf{x})$      # Compute log-odds ratio
 9: **return** $\psi_{\mathbf{c}}$

---

**Synthetic Data Augmentation.** Because the synthetic inpaints produced by the System-Embedded Diffusion Bridge (SDB) are validated to remain strictly in-distribution, they serve as high-fidelity data augmentations. By systematically inserting malignant nodules into samples known to exhibit spurious correlations, practitioners can actively decouple the predictive target from clinically unjustified artifacts during retraining.

**Attribution-Guided Regularization.** The exact nodule-level attributions derived via SHNAP offer a dense supervisory signal. Procedurally generated or expert-validated SHNAP maps can be directly integrated into the model's training objective as regularization targets, explicitly penalizing the network when it assigns high importance to background regions or known clinical artifacts rather than established pulmonary biomarkers.

## C. Extended experiments

### C.1. Risk correlation

Figure 10 presents the correlation matrix between Sybil's base hazard and its per-year risk outputs as computed on iLDCT. The lowest observed correlation is $\approx 0.9$ between the first-year risk and the base hazard; this high degree of alignment suggests that the base risk serves as a robust and universal representation of the model's overall diagnostic output.

### C.2. Experimental Setup

#### C.2.1. DATASETS

We base our analysis on three 3D Low-Dose Computed Tomography (LDCT) datasets, each serving a distinct role in auditing the model.

**D1. NLST.** We utilize the National Lung Screening Trial (NLST, National Lung Screening Trial Research Team, 2011) using the official splits from Sybil. This large-scale cohort comprises over 28,000 training and 6,000 test scans. The data provides a high-variance backdrop for training, with each scan containing between 120 and 400 slices of $512^2$ pixel resolution and a slice thickness of $\leq 2.5$ mm.

**D2. LUNA25.** The LUng Nodule Analysis dataset (LUNA25, Peeters et al., 2025) is a curated subset of the NLST consisting of 4,069 scans from 2,120 patients. It is particularly valuable for our nodule-centric analysis as it includes precise coordinates for 555 malignant and 5,608 benign nodules. Crucially, the ground truth labels are highly reliable: malignancy was confirmed via biopsy (providing pathology data unavailable to visual inspection), while benignity was verified by stability over a 2-year follow-up period. We use the provided centroid coordinates to construct naive spherical segmentation masks for our interventions.

**D3. iLDCT.** To test robustness on out-of-distribution data, we employ an internal screening dataset (iLDCT) of 243 scans from 145 patients. Unlike the screening-focused NLST, this dataset features a higher prevalence of severe cases. It has been annotated by two expert radiologists who provided detailed segmentation masks and malignant/benign classifications, serving as a rigorous testbed for model reasoning under more pathological conditions.

We include histograms for nodule counts for both LUNA25 and iLDCT in Figures 11 and 12.

## Risk Correlation Matrix

| | Base Hazard | Year 1 Risk | Year 2 Risk | Year 3 Risk | Year 4 Risk | Year 5 Risk | Year 6 Risk |
|---|---|---|---|---|---|---|---|
| **Base Hazard** | 1.000 | 0.898 | 0.935 | 0.941 | 0.949 | 0.953 | 0.970 |
| **Year 1 Risk** | 0.898 | 1.000 | 0.992 | 0.988 | 0.985 | 0.983 | 0.968 |
| **Year 2 Risk** | 0.935 | 0.992 | 1.000 | 0.998 | 0.998 | 0.997 | 0.988 |
| **Year 3 Risk** | 0.941 | 0.988 | 0.998 | 1.000 | 0.999 | 0.998 | 0.992 |
| **Year 4 Risk** | 0.949 | 0.985 | 0.998 | 0.999 | 1.000 | 0.999 | 0.994 |
| **Year 5 Risk** | 0.953 | 0.983 | 0.997 | 0.998 | 0.999 | 1.000 | 0.996 |
| **Year 6 Risk** | 0.970 | 0.968 | 0.988 | 0.992 | 0.994 | 0.996 | 1.000 |

*Figure 10.* Correlation matrix of Sybil's base hazard and per-year risk outputs computed on iLDCT.

### C.2.2. SDB TRAINING AND IMPLEMENTATION

**Computational Strategy.** Processing full 3D LDCT volumes is computationally prohibitive. We mitigate this by training SDB on randomly sampled $64^3$ cubes from volumes which were resampled to a target spacing of (1mm, 1mm, 2mm), utilizing the fact that nodule structure depends primarily on local high-frequency details rather than global context.

**Architecture and Training.** We employ a discrete-time (1000 steps) Schrödinger Bridge (SB) variant of SDB. Our implementation adapts the code from Liu et al. (2023a) by replacing 2D convolutions with 3D counterparts and implementing FlashAttention (Dao et al., 2022) in BF16 to improve efficiency, while the remaining computations are performed in FP32. The resulting model (76M parameters) was trained for approximately $1,600$ A100-hours on the NLST training split.

The architecture operates on $64^3$ volumes with a base width of 32 channels and multipliers of $\{1, 2, 3, 4\}$ across levels. Each resolution level contains 3 residual blocks, with single-head self-attention (32 channels per head) applied at the $16^3$ and $8^3$ resolutions. Training utilized an effective batch size of 128, a dropout rate of 0.1, exponential moving average (EMA) of weights ($\rho = 0.99$), and a noise schedule with $\beta_{\max} = 1.0$.

**Procedural Masking.** To avoid the bottleneck of costly manual labeling for training data, we generate subvolume masks procedurally using the metaballs algorithm (Blinn, 1982; Zhao et al., 2021). To further augment the variety of generated

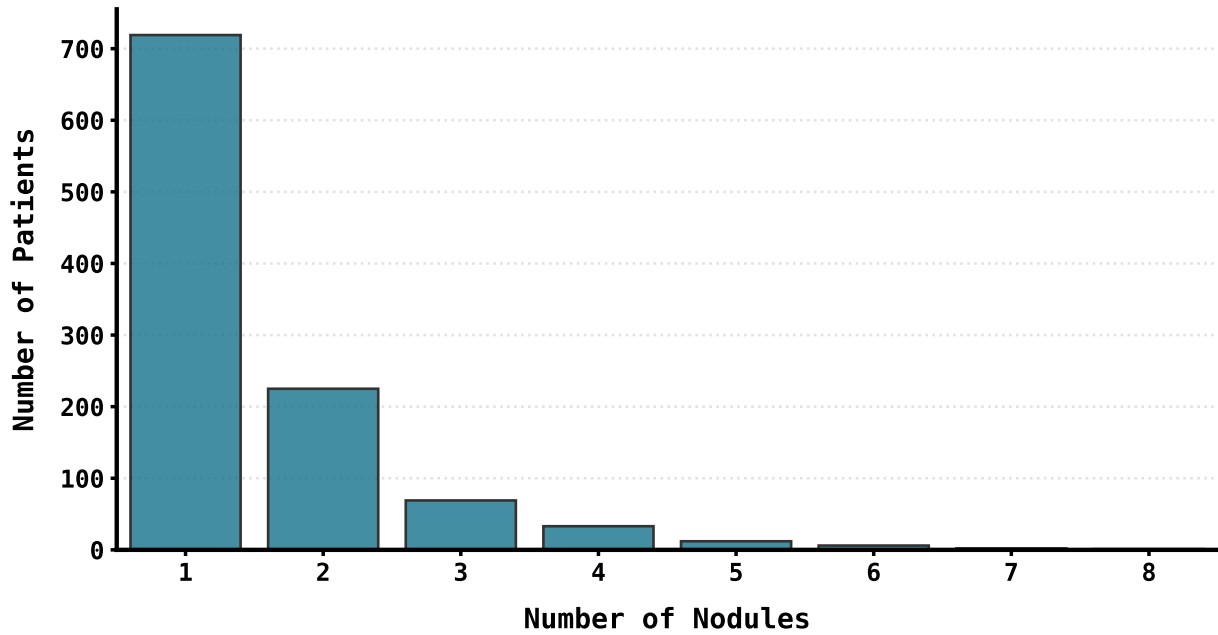

*Figure 11.* Distribution of nodule counts on LUNA25.

masks we randomly sample from 4 to 7 core-points, than for all points in the volume we compute the sum of cartesian distances to the core-points and binarize the mask using a randomly selected quintile (from near 0 to 0.4) as a threshold. Which results in random smooth regions with at most 40% of the cube's volume covered. This strategy yields a model capable of high-fidelity nodule removal and insertion without assuming training-time access to annotated nodule locations. This advantage enables large-scale training on the full NLST cohort, unlike recent methods that are limited to small, fully annotated cohorts (Zingarini et al., 2024; Kim et al., 2025a).

**Inference.** During the auditing phase, both nodule removal and insertion are performed using 100 Number of Function Evaluations (NFE) of the trained SDB model. To ensure that nodule insertions are precisely limited to the pulmonary volume, we utilize lung and lobe segmentation masks obtained from the TotalSegmentator (Wasserthal et al., 2023) adaptation proposed by Chrabaszcz et al. (2025).

### C.3. Baseline comparison

We compare SDB reconstruction capabilities against two pretrained models, Lung-DDPM (Jiang et al., 2025a) and MAISIv2 (Guo et al., 2025), adapted for inpainting via RePaint (Lugmayr et al., 2022), as well as pure noise and mean-value baselines. Quantitative results (see Table 1) were computed on 3,226 samples with metaball masks. To accommodate the pretrained baselines, inputs were padded to $128^3$ without mask resizing. Fréchet Inception Distance (FID) (Heusel et al., 2017) metrics were calculated using RadImageNet (Mei et al., 2022) embeddings on slices containing masked regions.

### C.4. Expert validation

We performed two distinct experiments to evaluate the realism of the synthetic samples resulting from nodule removal and insertion. For this purpose, two expert radiologists evaluated the images using both consumer-grade and medical-grade displays, and domain-standard medical imaging software (Kikinis et al., 2014). Because our overall approach involves preserving the majority of the original image exactly, with modifications restricted to select cubic regions, each experiment presented the potentially modified region at the center of a cropped 3D CT scan. Every crop included a surrounding large enough offset to ensure that genuine local content served as a visual reference. This design ensures that experts do not need to search the entire scan for localized changes, allowing them to concentrate solely on the region of interest.

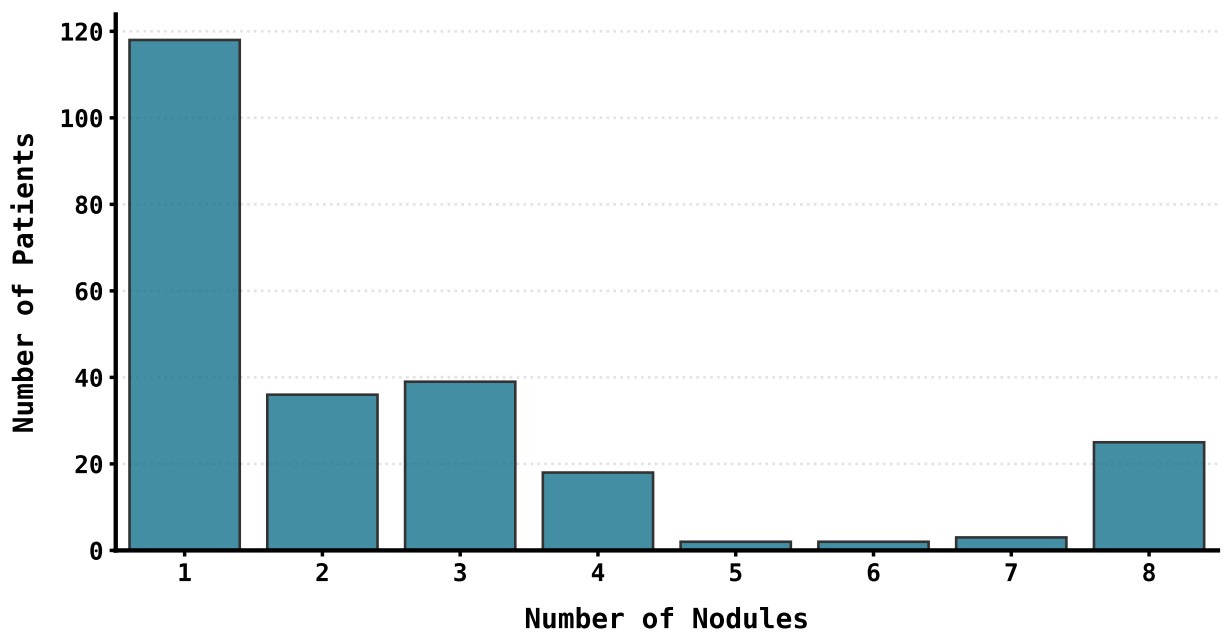

*Figure 12.* Distribution of nodule counts on iLDCT.

### C.5. SHNAP attribution stability

To assess the stability of the explanations, Figure 13 presents the distribution of standard deviations for all SHNAP attributions across the iLDCT dataset. By computing these attributions in probability space, we provide a direct measure of local explanation variance. The tight clustering of low standard deviation values indicates that the SHNAP estimates are highly stable, ensuring that the reported nodule contributions are not artifacts of sampling noise or local model instabilities. Figure 14 presents an analogous plot in logit space.

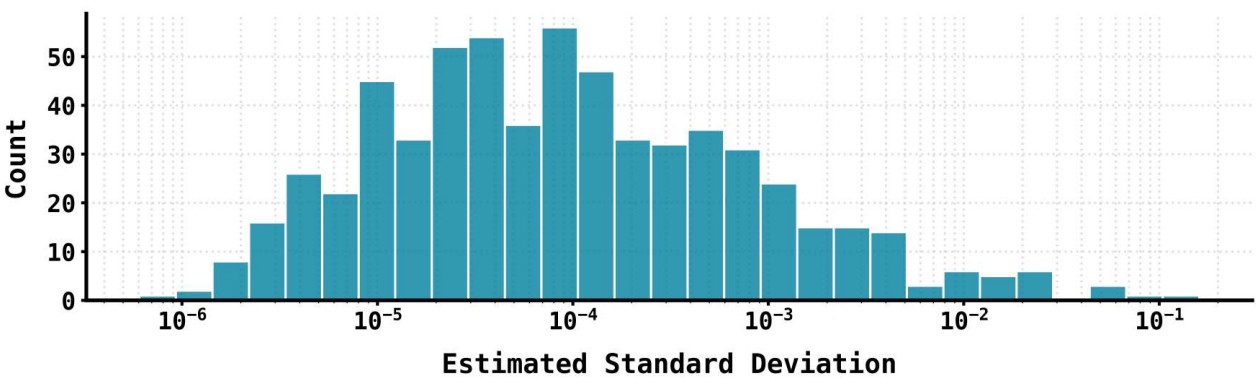

*Figure 13.* Histogram of standard deviations for all SHNAP attributions from iLDCT. Attributions were computed in probability space to facilitate the interpretation of deviation magnitudes.

*Table 1.* **Quantitative evaluation of inpainting methods.** The table reports Fréchet Inception Distance (FID) across three planes (XY, XZ and YZ) and metrics computed on masked regions. RMSE and MAE are reported in Hounsfield Units (HU). The proposed method is highlighted using a gray background.

| | FID Metrics ($\downarrow$) | | | | Masked Region Metrics | | |
|---|---|---|---|---|---|---|---|
| **Method** | XY | XZ | YZ | Mean | RMSE (HU) | MAE (HU) | SSIM ($\uparrow$) |
| Gaussian Noise $\mathcal{N}(\mu, 200.0)$ | 19.3888 | 16.8558 | 16.5952 | 17.6133 | 519.6376 | 424.0856 | 0.0395 |
| Sample mean $\mu$ | 4.8369 | 5.1570 | 5.1722 | 5.0554 | 445.61 | 379.27 | 0.2935 |
| Lungs-DDPM | 15.0303 | 13.9117 | 14.1776 | 14.3732 | 439.53 | 235.60 | 0.6608 |
| MAISIv2 | 0.4369 | 0.5297 | 0.4996 | 0.4887 | 278.71 | 147.25 | 0.6777 |
| Our Method | 0.0439 | 0.0400 | 0.0410 | 0.0417 | 124.51 | 65.80 | 0.5972 |

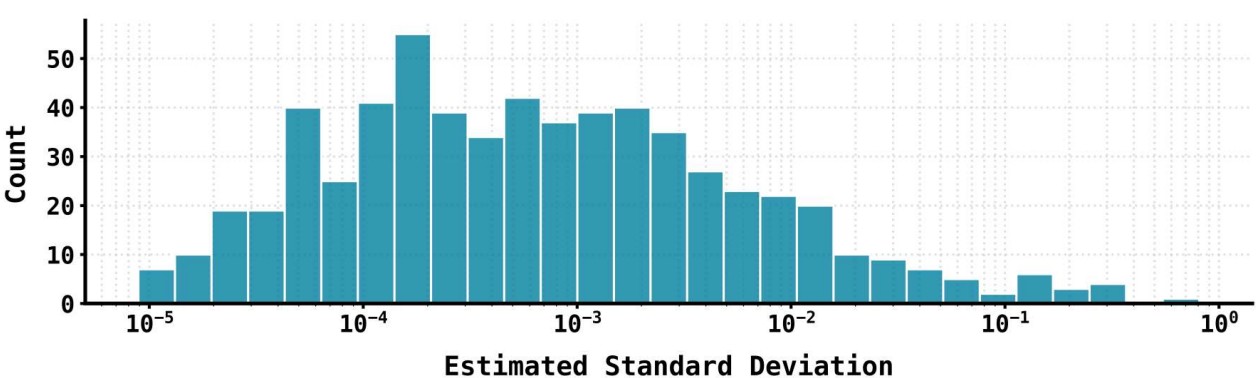

*Figure 14.* Histogram of standard deviations for all SHNAP attributions from iLDCT. Attributions were computed in logit space.

## C.6. SHNAP attribution stability across different baselines

Selecting appropriate baselines for nodule removal is non-trivial. Simple strategies often yield out-of-distribution samples, leading to inconsistent model interpretations. We quantify this inconsistency in Figure 15 by reporting the standard deviation of SHNAP attributions computed across four distinct naive baselines for each image. Specifically, for every masked sample, we generated four separate variations using: global mean replacement, unmasked-region mean replacement, median replacement, and fixed lung tissue intensity. The resulting standard deviations of SHNAP attributions are, on average, orders of magnitude higher than those observed with our method (Figure 14). This result indicates that naive baselines are inconsistent while our approach behaves stability and evaluates the model on the in-manifold data.

## C.7. Additional examples of SHNAP

We include additional SHNAP explanations that extend the exploration from Figure 6. Figures 16 and 17 show false positives characterized by multi-year instability. Figures 18 and 19 visualize a pleural nodule missed by Sybil in year-after-year follow-up scans. Figure 20 presents Sybil's reliance on a thyroid goiter, while Figure 21 illustrates how the model focuses on a metallic object behind the patient's back.

## C.8. Contribution of attention-based regions

Figure 22 compares Sybil's base risk predictions against the relative contributions of attention-based regions, computed analogously to RNC. A visible divergence in attribution patterns exists when compared to Figure 5, as the attention-based approach focuses more heavily on negative contributions.

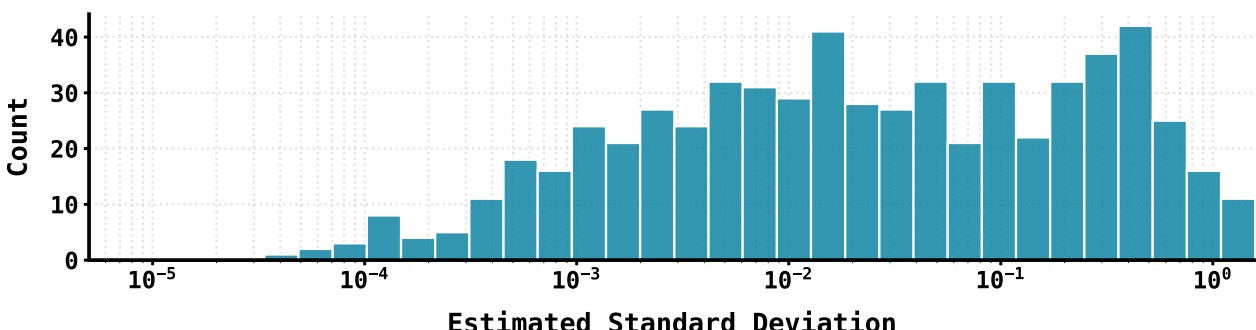

*Figure 15.* Histogram of standard deviations for all SHNAP attributions from iLDCT on simple baselines. Attributions were computed in logit space.

### C.9. Influential regions are sparse

Figure 23 visualizes the relative contributions (on a symlog scale) of lung regions distinct from both annotated nodules and attention-based regions. These results indicate that the attribution for these background areas is effectively zero.

### C.10. SNAP attribution maps

Figures 24 to 26 depict SNAP attribution maps across 12 patient scans and 10 malignant nodules in coronal, axial, and sagittal planes, respectively.

Figures 27 to 29 depict SNAP attribution maps across 12 patient scans and 10 benign nodules in coronal, axial, and sagittal planes, respectively.

### C.11. Radial sensitivity bias

Figure 30 visualizes nonlinear trends for SNAP values across multiple nodules inserted into a single patient, plotted against each insertion's distance from the pleural surface.

### C.12. Extended Baseline Comparisons and Efficiency

#### C.12.1. COMPARISON WITH CLASSICAL ATTRIBUTION METHODS

To establish the efficacy of our approach relative to existing post-hoc attribution methods, we evaluate SHNAP against several standard perturbation-based and gradient-based techniques. As detailed in Table 2, our approach consistently outperforms classical methods across standard metrics.

#### C.12.2. SHNAP GOODNESS-OF-FIT ON CLASSICAL REGIONS

We further evaluate the goodness-of-fit of the SHNAP approximation on binarized regions derived from classical attribution methods. The results, shown in Figure 31, demonstrate the framework's adaptability and high fidelity across different region definitions.

#### C.12.3. COMPUTATIONAL OVERHEAD

A fundamental limitation of post-hoc attribution methods in 3D medical imaging is their prohibitive execution time. We compare the execution time of SHNAP against standard perturbation-based methods. As detailed in Table 3, SHNAP evaluates the underlying model drastically faster than black-box alternatives, completing in 08:08 minutes for 8 nodules compared to over an hour for Kernel SHAP. By operating at the modular semantic level of nodules rather than dense volumetric grids, SHNAP provides an exact n-Shapley decomposition with extreme efficiency.

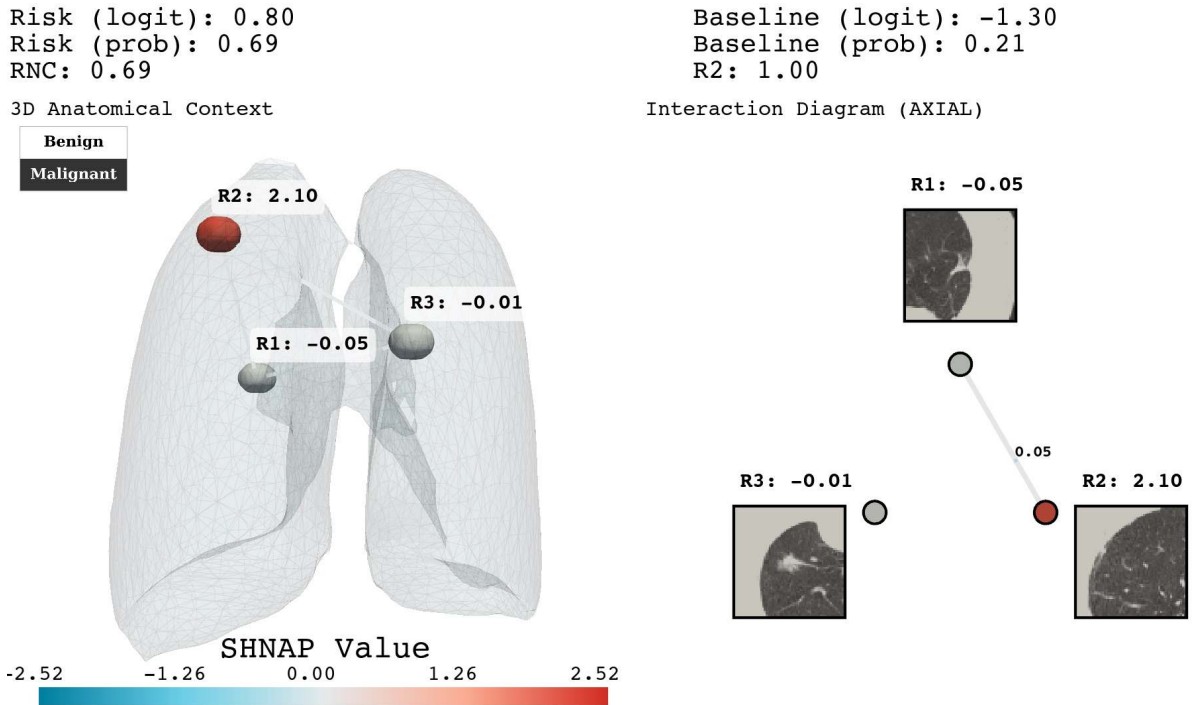

```
Risk (logit): 0.80          Baseline (logit): -1.30
Risk (prob): 0.69           Baseline (prob): 0.21
RNC: 0.69                   R2: 1.00
```

*Figure 16.* SHNAP explanation visualizing 3D anatomical context and nodule interaction effects. This case demonstrates a follow-up scan of Figure 6 (**b**) featuring a false positive prediction where Sybil shifts focus to a different nodule than the one identified in Figure 6 (**b**).

Conversely, SNAP requires dense, high-resolution evaluations (approximately 1,000 insertions per image). Consequently, generating a single SNAP attribution map requires $\approx 100$ minutes on a single GPU. However, because each insertion evaluates independently, SNAP is highly parallelizable and scales linearly across available computing devices.

### C.13. Robustness to Upstream Detection and Scene Complexity

#### C.13.1. IMPACT OF NODULE COUNT ON SHNAP FIT

To assess the influence of complexity on explanation quality, we stratify the SHNAP goodness-of-fit by nodule count. As depicted in Figure 32, the explanation accuracy remains highly robust even as the number of evaluated nodules increases.

#### C.13.2. SENSITIVITY TO AUTOMATED SEGMENTATION MASKS

We replace expert-annotated masks with those generated by a fully automated SwinUNETR (Hatamizadeh et al., 2021) model to test sensitivity to upstream segmentation methods. Figure 33 displays the resulting goodness-of-fit. Furthermore, Figure 34 illustrates the resulting RNC distribution, indicating that the framework successfully recovers the attribution patterns observed when using expert annotations.

#### C.13.3. IMPACT OF MISSED NODULE DETECTIONS

We provide an assessment of nodule detection dependence by simulating missed nodules using the analytically tractable formulation of nSV. We first select patients with at least 3 nodules and compute their mean goodness-of-fit by averaging over cases with a simulated, randomly chosen missed nodule. As indicated by Figure 35, even when nodules are missed, goodness-of-fit remains extremely high.

#### C.13.4. SENSITIVITY TO MASK BOUNDARY NOISE

We perform a sweep over mask volumes to evaluate the method's sensitivity to boundary noise. Table 4 reveals that the attribution values stabilize around a mask volume of 10 mm$^3$, indicating high robustness against imprecise segmentation boundaries.

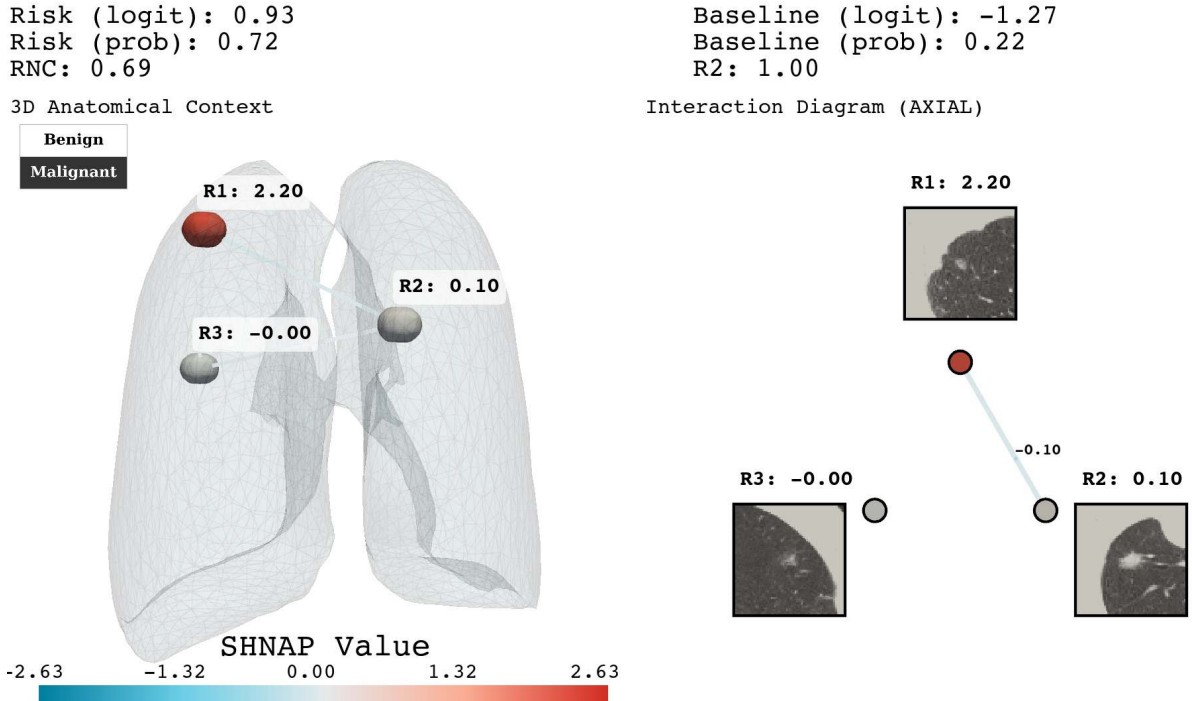

Risk (logit): 0.93
Risk (prob): 0.72
RNC: 0.69

Baseline (logit): −1.27
Baseline (prob): 0.22
R2: 1.00

*Figure 17.* SHNAP explanation visualizing 3D anatomical context and nodule interaction effects. This example extends Figure 16, illustrating that Sybil does not retain its original focus from Figure 6 (**b**).

## C.14. Fidelity and Stability of Generative Interventions

### C.14.1. CROSS-GENERATOR CONSISTENCY AND CHECKPOINT STABILITY

To verify cross-generator consistency and benchmark against existing generative methods, we evaluate the stability of our method across multiple SDB training checkpoints alongside standard baselines. Table 5 confirms stable quantitative inpainting metrics across iterations while demonstrating significantly lower FID and reconstruction errors compared to baseline methods. Furthermore, Figure 36 shows that the generative attribution patterns remain strictly consistent irrespective of minor network weight fluctuations between checkpoints.

### C.14.2. SNAP STABILITY ACROSS INITIAL RANDOM SEEDS

We evaluate the impact of initial generative noise on SNAP explanations by computing attributions across multiple random seeds. Figure 37 demonstrates that the resulting explanations are highly insensitive to initial noise variations.

### C.14.3. INPAINTING IMPACT ON ORIGINAL MODEL PREDICTIONS

To verify that the generative inpainting process does not introduce artifacts that skew model behavior, we evaluated the difference in predictions between original scans and those where nodules were removed and immediately regenerated using identical masks. On the iLDCT dataset, this procedure yielded a negligible probability difference of $−0.02 \pm 0.08$ (mean $\pm$ standard deviation). This minor variance, stemming entirely from the inherent stochasticity of SDB, confirms the high stability of the generative interventions and ensures they do not artificially distort the model's decision-making.

## C.15. Generalizability and Ensemble Diagnostics

### C.15.1. INTRA-ENSEMBLE CONSISTENCY ACROSS SYBIL BACKBONES

We analyze the intra-ensemble consistency by separating SNAP attributions across Sybil's 5 constituent backbones (Figure 38). The pairwise Kolmogorov-Smirnov test (Table 6) yields statistically significant differences ($p < 0.001$) between all backbones. These distinct distributions highlight that backbones behave diversely due to training stochasticity, possibly improving model robustness when their predictions are aggregated within the ensemble.

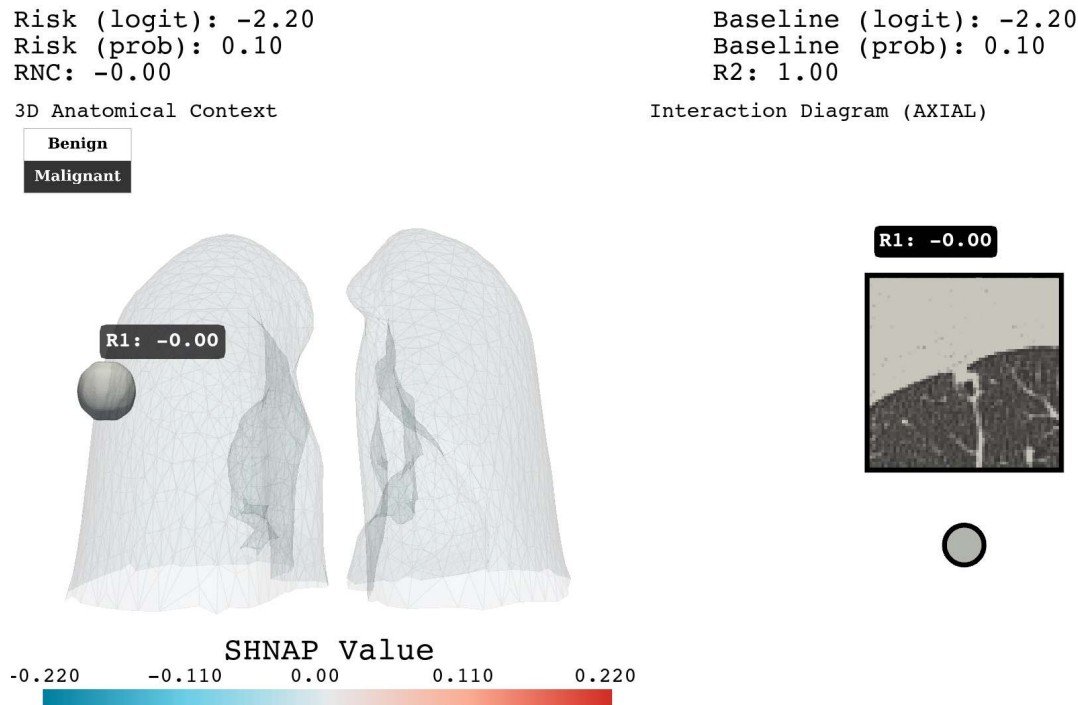

```
Risk (logit): -2.20          Baseline (logit): -2.20
Risk (prob): 0.10            Baseline (prob): 0.10
RNC: -0.00                   R2: 1.00
```

*Figure 18.* SHNAP explanation visualizing 3D anatomical context and nodule interaction effects. The map reveals a pleural nodule receiving negligible attribution, which contributes to its omission from the model's risk assessment.

### C.15.2. GENERALIZABILITY TO ALTERNATIVE MODELS (CXR-LC)

To test the generalizability of the auditing framework to alternative modalities, we apply SHNAP to the 2D X-ray-based CXR-LC lung cancer risk prediction model (Lu et al., 2020). SHNAP is applied in an equivalent manner, but the resulting 3D LDCTs are first mapped to 2D X-rays using DiffDRR (Gopalakrishnan & Golland, 2022) before being passed to the model. Figure 39 presents the goodness-of-fit distribution, confirming the framework's adaptability. Furthermore, Figure 40 establishes that this attribution stability holds regardless of scan complexity.

### C.16. Qualitative examples

We include qualitative examples of nodule removal and nodule insertion in Figures 41 and 42 respectively.

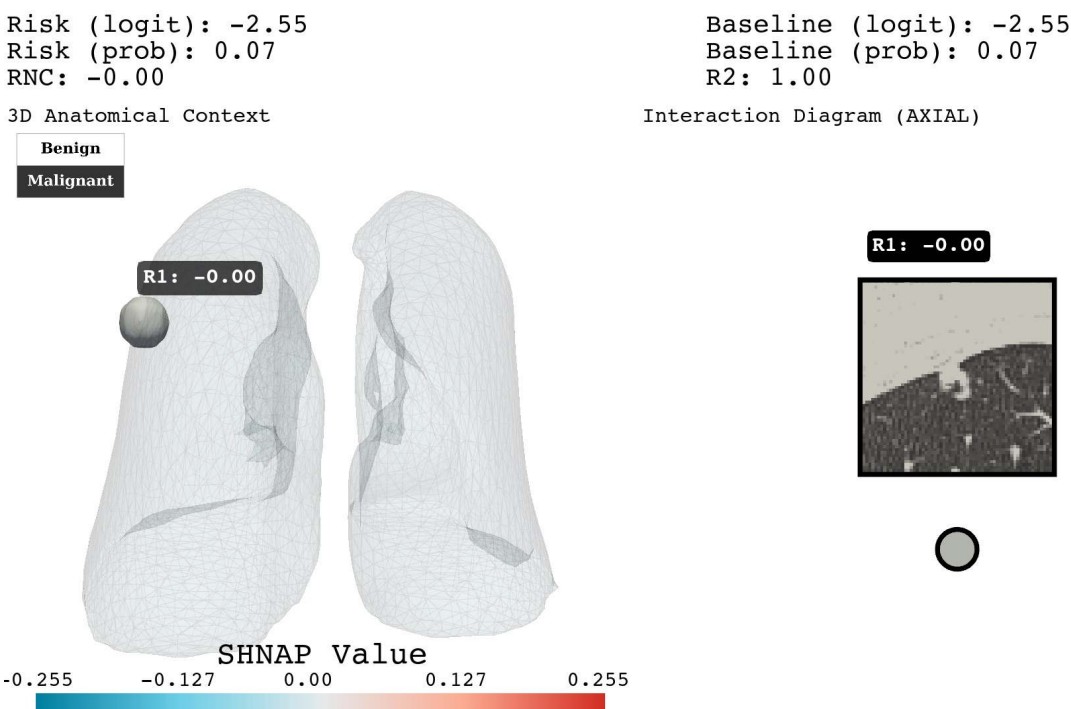

Risk (logit): -2.55
Risk (prob): 0.07
RNC: -0.00

3D Anatomical Context

Baseline (logit): -2.55
Baseline (prob): 0.07
R2: 1.00

Interaction Diagram (AXIAL)

*Figure 19.* SHNAP explanation visualizing 3D anatomical context and nodule interaction effects. Longitudinal follow-up of Figure 18 confirms a persistent failure to detect the peripheral pleural nodule across consecutive screening rounds.

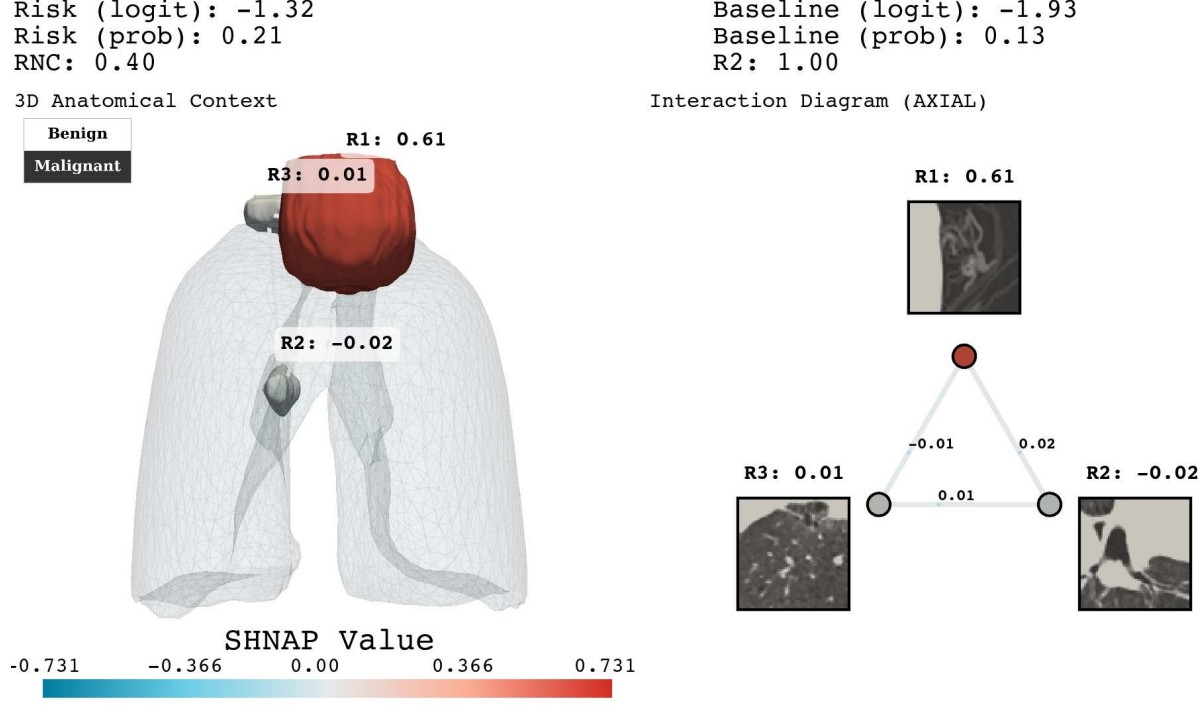

Risk (logit): -1.32
Risk (prob): 0.21
RNC: 0.40

3D Anatomical Context

Baseline (logit): -1.93
Baseline (prob): 0.13
R2: 1.00

Interaction Diagram (AXIAL)

*Figure 20.* SHNAP explanation visualizing 3D anatomical context and nodule interaction effects. The model erroneously relies on a thyroid goiter as a primary indicator of risk—a clinically unjustified diagnostic artifact.

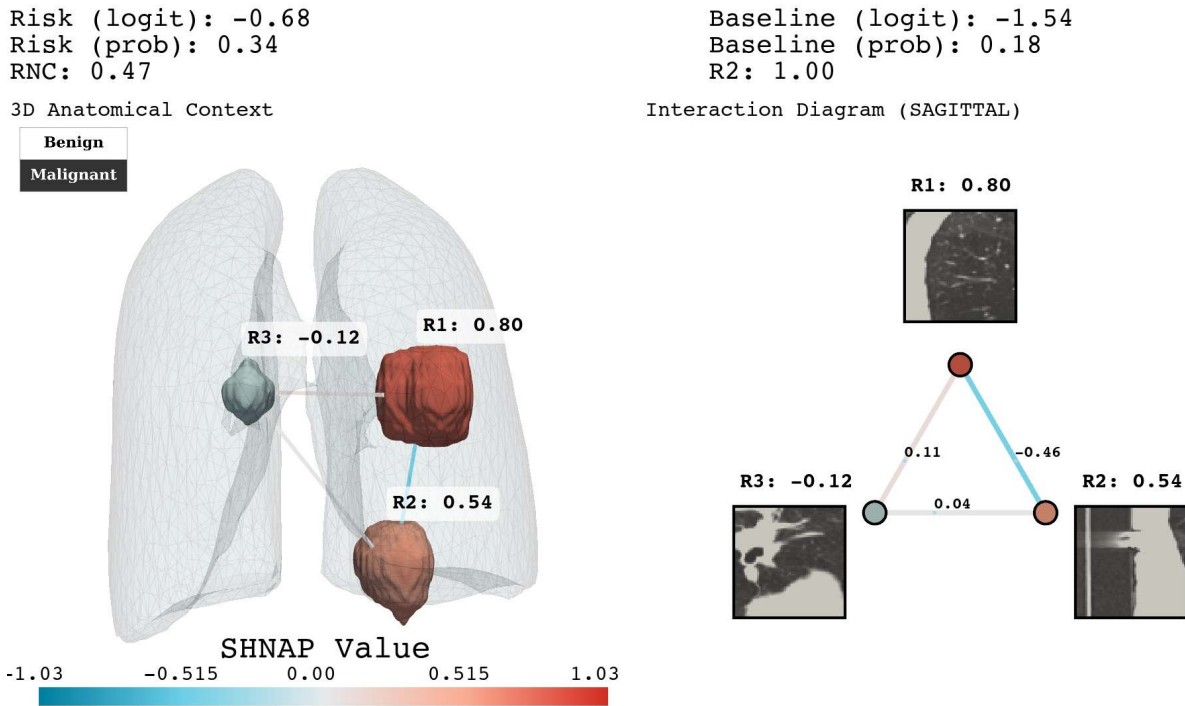

Figure 21. SHNAP explanation visualizing 3D anatomical context and nodule interaction effects. Significant attribution is incorrectly placed on a metallic object positioned behind the patient's back, outside of the pulmonary region.

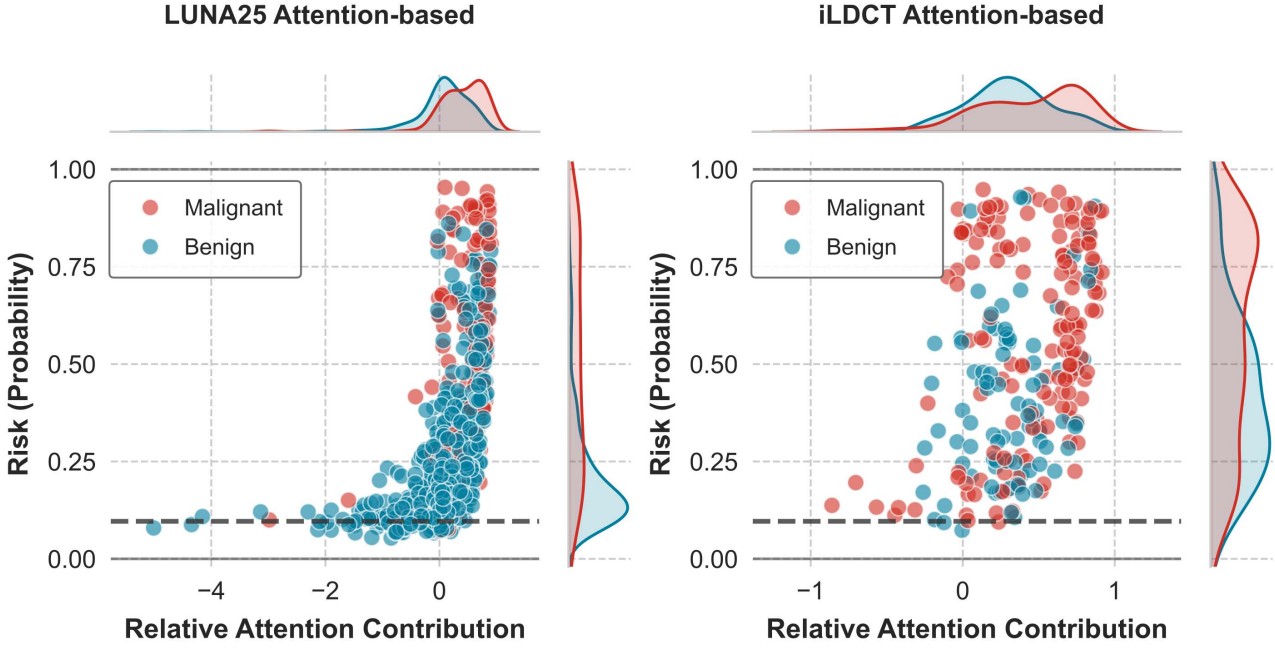

Figure 22. Comparison of Sybil's risk predictions against attention-based region contributions. The plots illustrate a shift in the importance distribution, highlighting a greater emphasis on the negative contributions within attention-based regions.

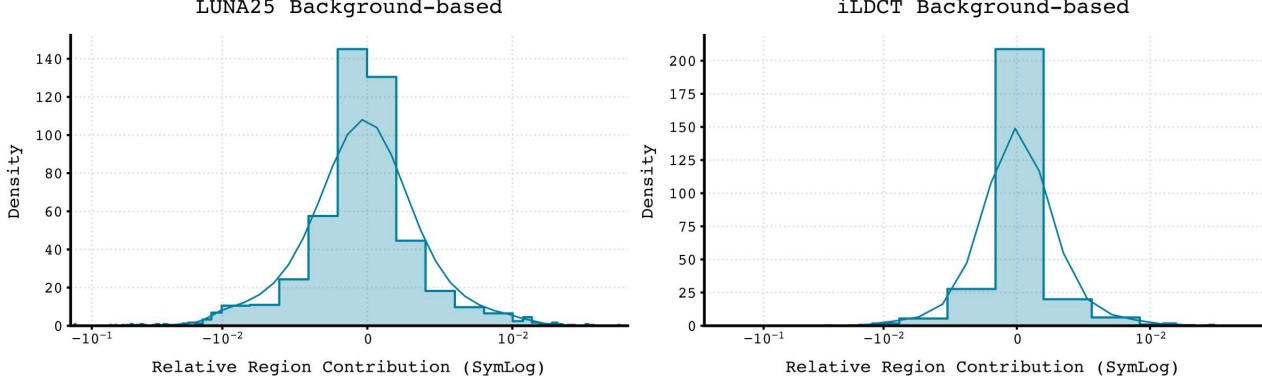

*Figure 23.* SHNAP attribution for background lung regions. The distribution, plotted on a symlog scale, shows that regions excluding nodules and attention-based areas contribute negligibly to the overall risk prediction.

*Table 2.* Quantitative evaluation of SHNAP vs standard post-hoc attribution methods. The evaluation is divided into Localization and Fidelity metrics (top) and Complexity and Exactness metrics (bottom).

| Method | RRA(↑) | RRA abs.(↑) | Insertion AUC(↑) | Deletion AUC(↓) |
|---|---|---|---|---|
| Feature Ablation | 0.0740 | 0.0746 | 0.2469 | 0.4132 |
| Guided Grad-CAM | 0.0969 | 0.1215 | 0.2737 | 0.4610 |
| Input × Gradient | 0.1261 | 0.1798 | 0.3117 | 0.2933 |
| Integrated Gradients | 0.1054 | 0.1464 | 0.2796 | 0.3090 |
| Kernel SHAP | 0.0064 | 0.0064 | 0.1677 | 0.5614 |
| Occlusion 3D | 0.1091 | 0.1057 | 0.2945 | 0.3104 |
| Saliency | 0.2569 | 0.2569 | 0.3440 | 0.2691 |
| SHNAP | 0.8050 | 0.9283 | 0.3450 | 0.2490 |

| Method | Perturbation AUC(↑) | Mean Perturbation(↑) | Sparseness(↑) | Completeness (our baseline)(↓) | Completeness (zero baseline)(↓) |
|---|---|---|---|---|---|
| Feature Ablation | 0.1368 | 0.1342 | 0.7850 | 7415.97 | 7415.88 |
| Guided Grad-CAM | 0.1232 | 0.1220 | 0.9600 | 0.2752 | 0.3677 |
| Input × Gradient | 0.2277 | 0.2252 | 0.8518 | 0.2208 | 0.3133 |
| Integrated Gradients | 0.1941 | 0.1921 | 0.8688 | 0.0763 | 0.0162 |
| Kernel SHAP | 0.0108 | 0.0108 | 0.1288 | 1605.94 | 1605.85 |
| Occlusion 3D | 0.1627 | 0.1603 | 0.7404 | 33644.30 | 33644.30 |
| Saliency | 0.2195 | 0.2181 | 0.7381 | 1105.40 | 1105.31 |
| SHNAP | 0.2530 | 0.2515 | 0.9893 | 0.0157 | 0.0768 |

## Mean SNAP over Coronal (Y) axis
## (Malignant nodules)

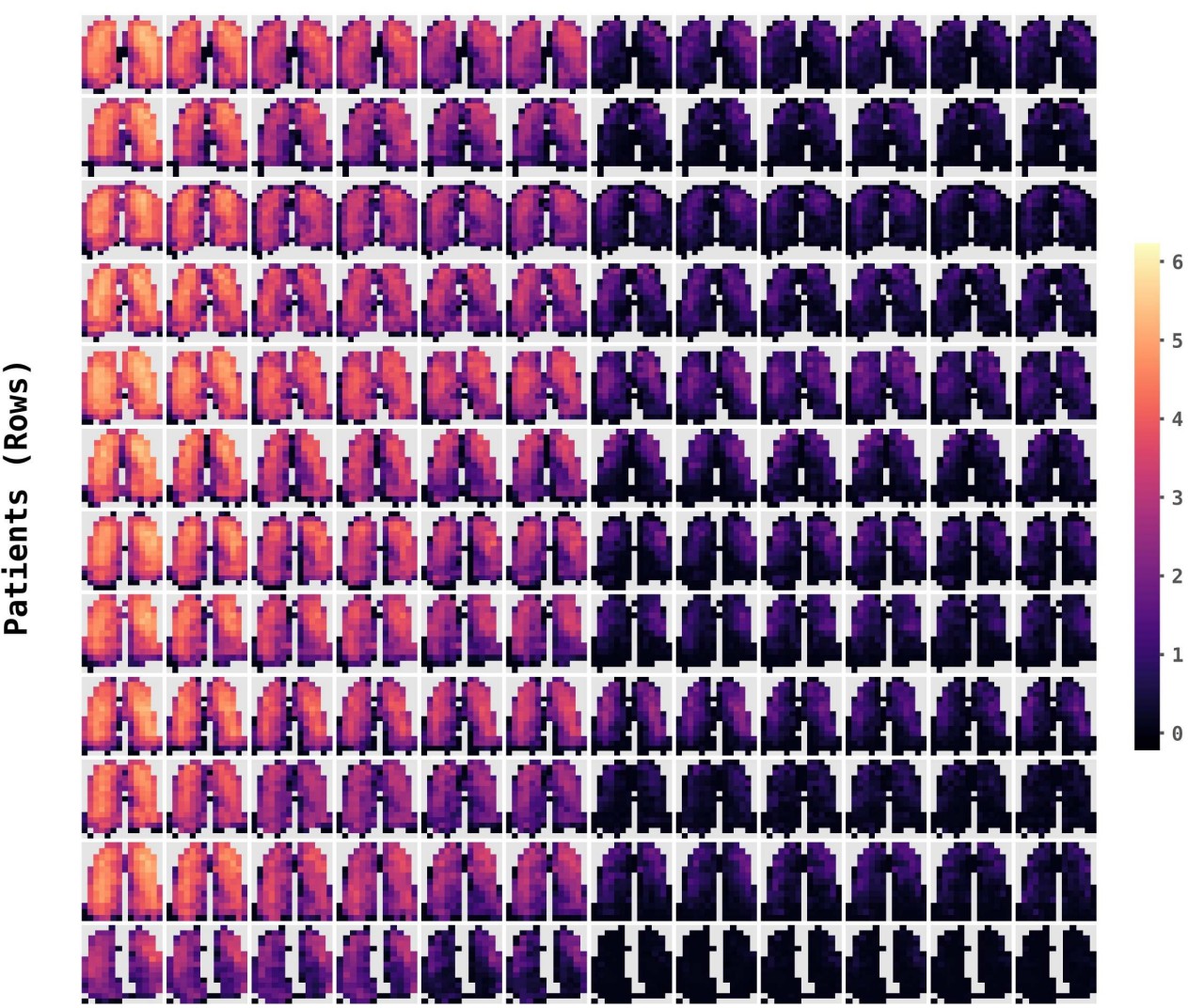

*Figure 24.* SNAP attribution maps for malignant nodules across patients in the coronal plane.

*Table 3.* **Attribution Method Execution Time.** Execution times [mm:ss] for perturbation-based methods evaluated on a standard 3D LDCT volume.

| Method | Execution Time [mm:ss] |
| --- | --- |
| Feature Ablation | 26:05 |
| Occlusion | 56:20 |
| Kernel SHAP (2048 evaluations) | 70:20 |
| SHNAP (8 nodules) | 08:08 |

## Mean SNAP over Axial (Z) axis
## (Malignant nodules)

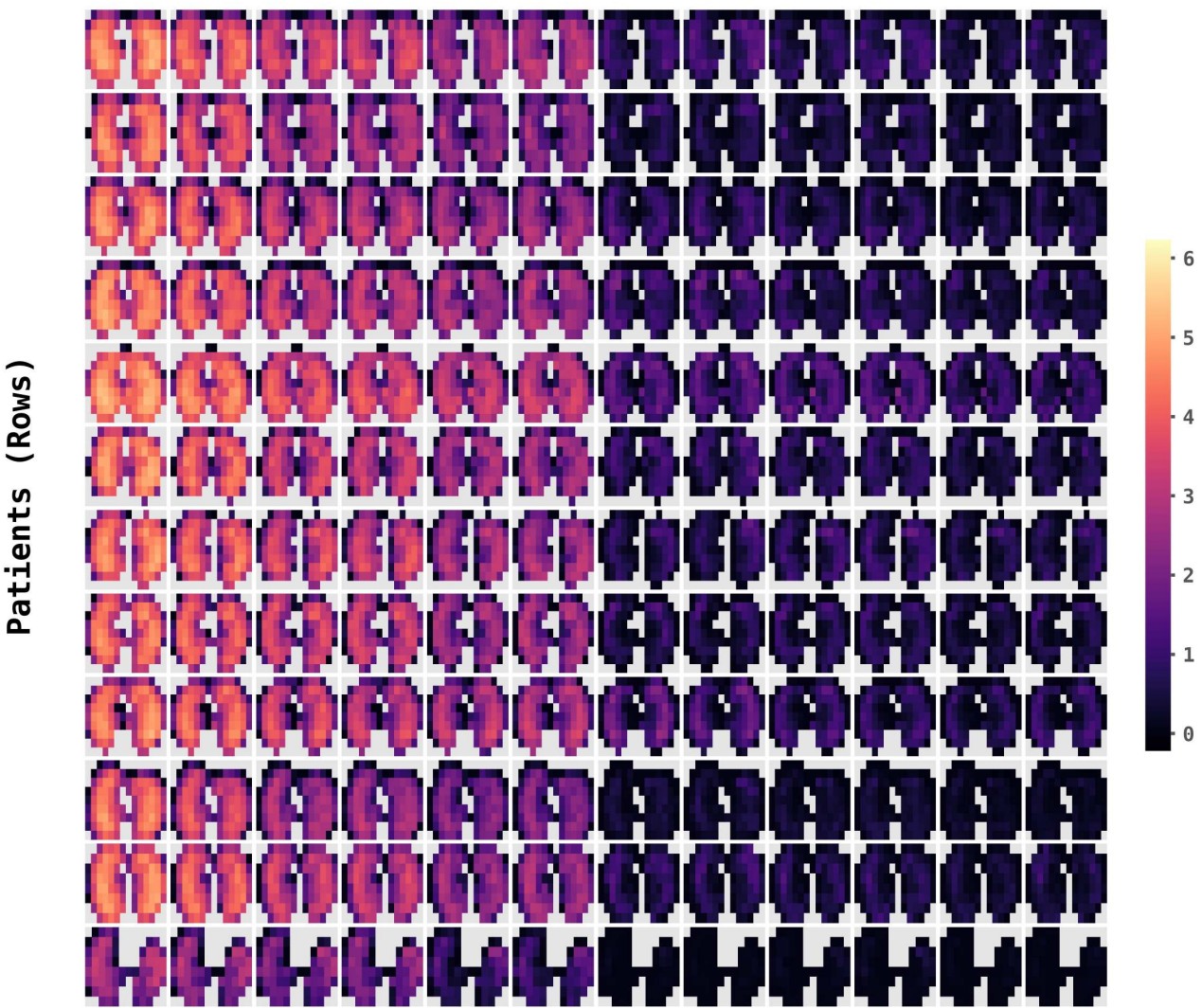

*Figure 25.* SNAP attribution maps for malignant nodules across patients in the axial plane.

*Table 4.* SHNAP Tolerance to Mask Boundary Noise. The table reports the percentage of stabilized samples at various cumulative tolerance thresholds (maximum $\Delta$ from the 10 mm$^3$ mask baseline).

| Max. $\Delta$ from 10 mm$^3$ | 0.001 | 0.005 | 0.010 | 0.015 | 0.020 | 0.030 | 0.050 | 0.100 |
|---|---|---|---|---|---|---|---|---|
| Stabilized Samples (%) | 28.85 | 64.52 | 74.23 | 78.85 | 83.46 | 88.08 | 90.38 | 92.69 |

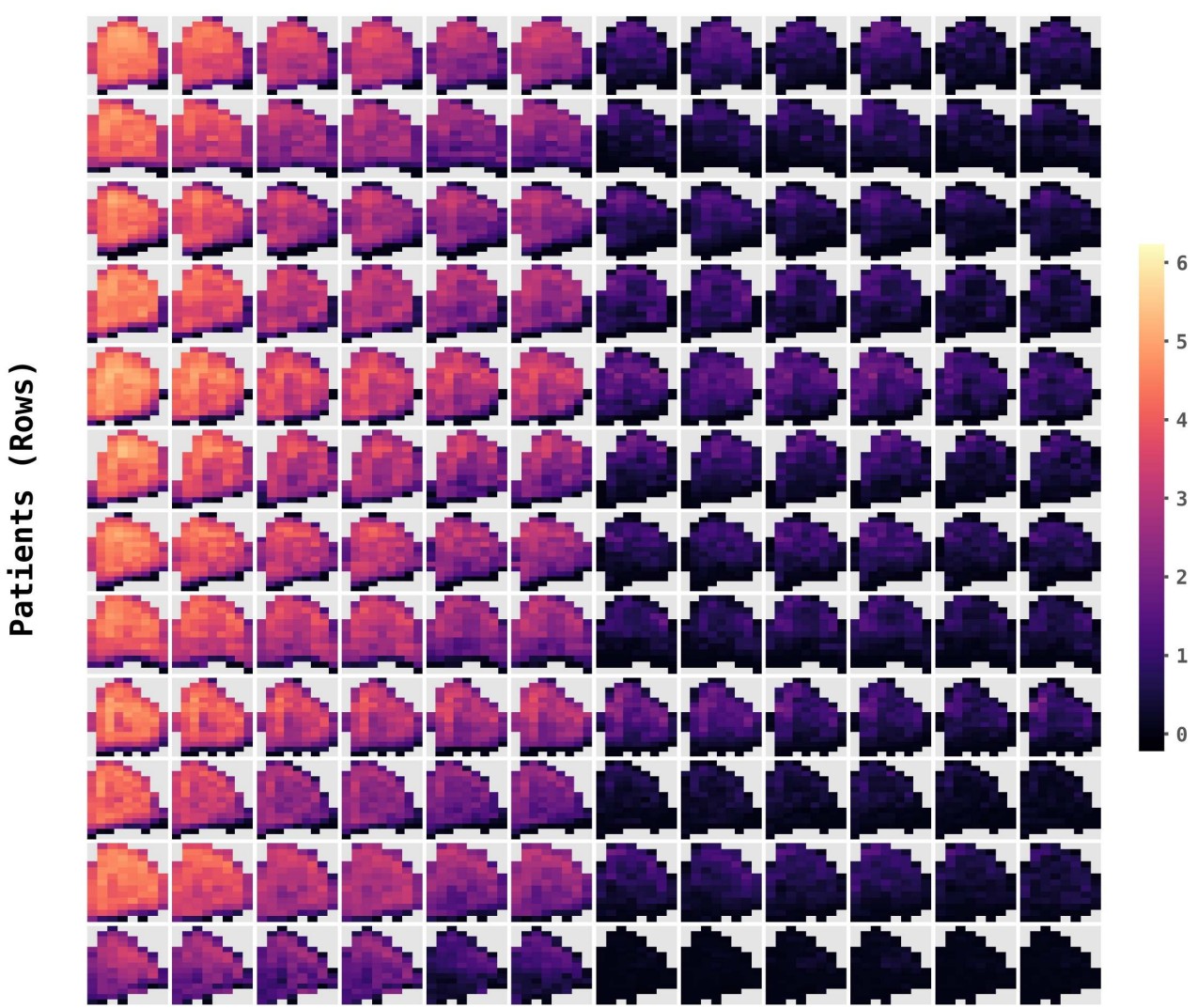

**Mean SNAP over Sagittal (X) axis (Malignant nodules)**

*Figure 26.* SNAP attribution maps for malignant nodules across patients in the sagittal plane.

*Table 5.* Quantitative evaluation of inpainting stability across SDB checkpoints and baseline methods. The table reports FID, KID, and LPIPS metrics, alongside reconstruction metrics computed on masked regions.

| | FID (↓) | | | | Perceptual (↓) | | Masked Region | | |
|---|---|---|---|---|---|---|---|---|---|
| Method | XY | XZ | YZ | Mean | KID | LPIPS | RMSE (HU) (↓) | MAE (HU) (↓) | SSIM (↑) |
| Gaussian Noise $\mathcal{N}(\mu, 200.0)$ | 19.4382 | 16.8807 | 16.6886 | 17.6691 | 0.012603 | 0.3018 | 520.69 | 424.98 | 0.0395 |
| Sample mean $\mu$ | 4.8345 | 5.2016 | 5.1683 | 5.0682 | 0.002411 | 0.2808 | 447.05 | 380.52 | 0.2939 |
| Lungs-DDPM | 20.4442 | 10.4830 | 11.1370 | 14.0214 | 0.003493 | 0.0401 | 439.53 | 235.60 | 0.6614 |
| MAISIv2 | 1.5379 | 0.6365 | 0.6001 | 0.9248 | 0.000193 | 0.0159 | 278.71 | 147.25 | 0.6751 |
| SDB (34k) | 0.0320 | 0.0300 | 0.0318 | 0.0313 | 0.000007 | 0.0772 | 128.02 | 68.25 | 0.5838 |
| SDB (36k) | 0.0314 | 0.0293 | 0.0309 | 0.0306 | 0.000008 | 0.0765 | 127.09 | 67.71 | 0.5869 |
| SDB (main paper) | 0.0290 | 0.0301 | 0.0302 | 0.0298 | 0.000005 | 0.0760 | 126.60 | 67.29 | 0.5816 |

## Mean SNAP over Coronal (Y) axis
## (Benign nodules)

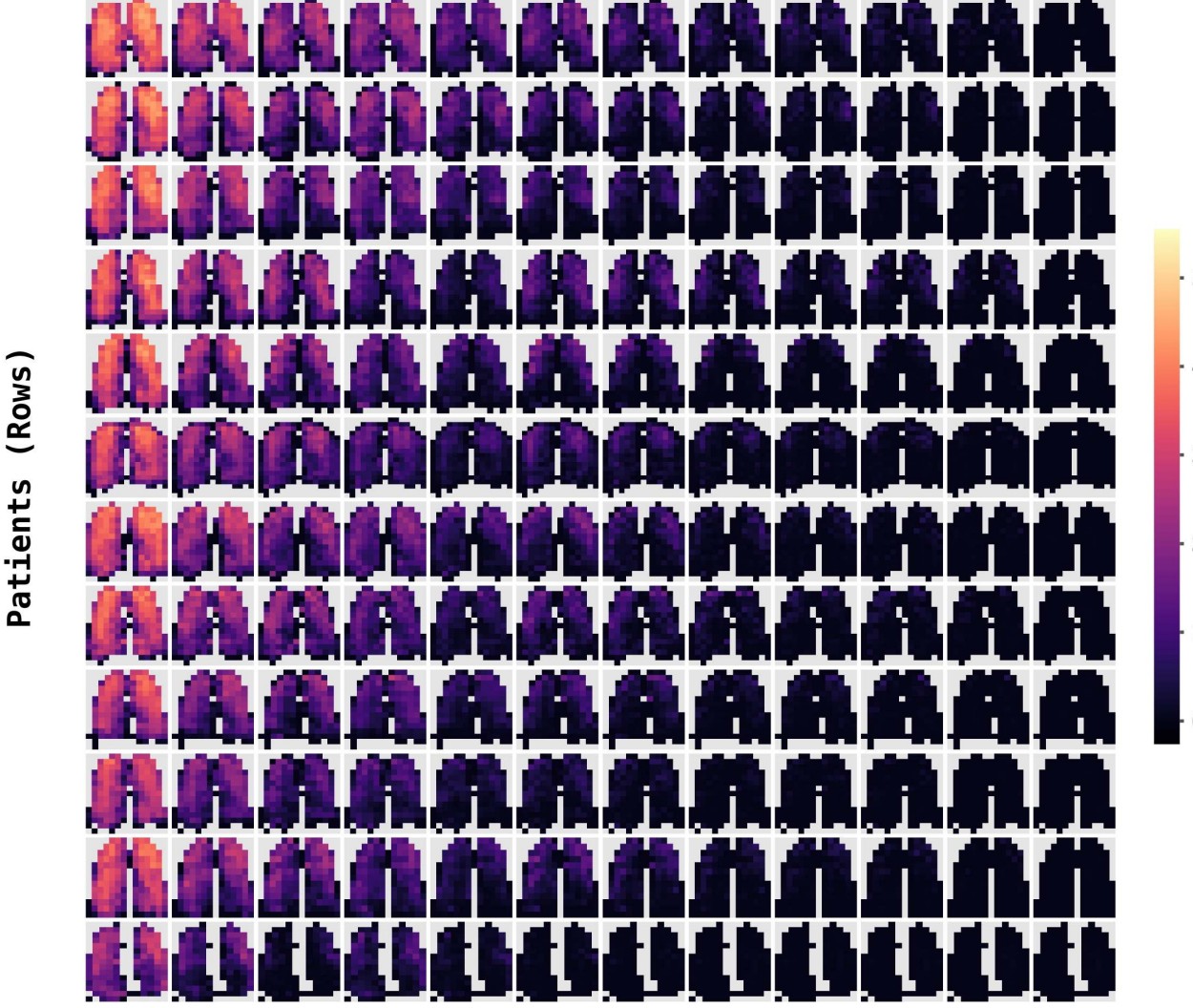

**Nodules (Columns)**

*Figure 27.* SNAP attribution maps for benign nodules across patients in the coronal plane.

*Table 6.* Pairwise Kolmogorov-Smirnov test p-values comparing attribution distributions across the 5 Sybil backbones. The tests reveal that attributions within each pair differ with statistical significance.

| Backbone | 1 | 2 | 3 | 4 | 5 |
|---|---|---|---|---|---|
| 1 | – | $< 0.001$ | $< 0.001$ | $< 0.001$ | $< 0.001$ |
| 2 | $< 0.001$ | – | $< 0.001$ | $< 0.001$ | $< 0.001$ |
| 3 | $< 0.001$ | $< 0.001$ | – | $< 0.001$ | $< 0.001$ |
| 4 | $< 0.001$ | $< 0.001$ | $< 0.001$ | – | $< 0.001$ |
| 5 | $< 0.001$ | $< 0.001$ | $< 0.001$ | $< 0.001$ | – |

# Mean SNAP over Axial (Z) axis
## (Benign nodules)

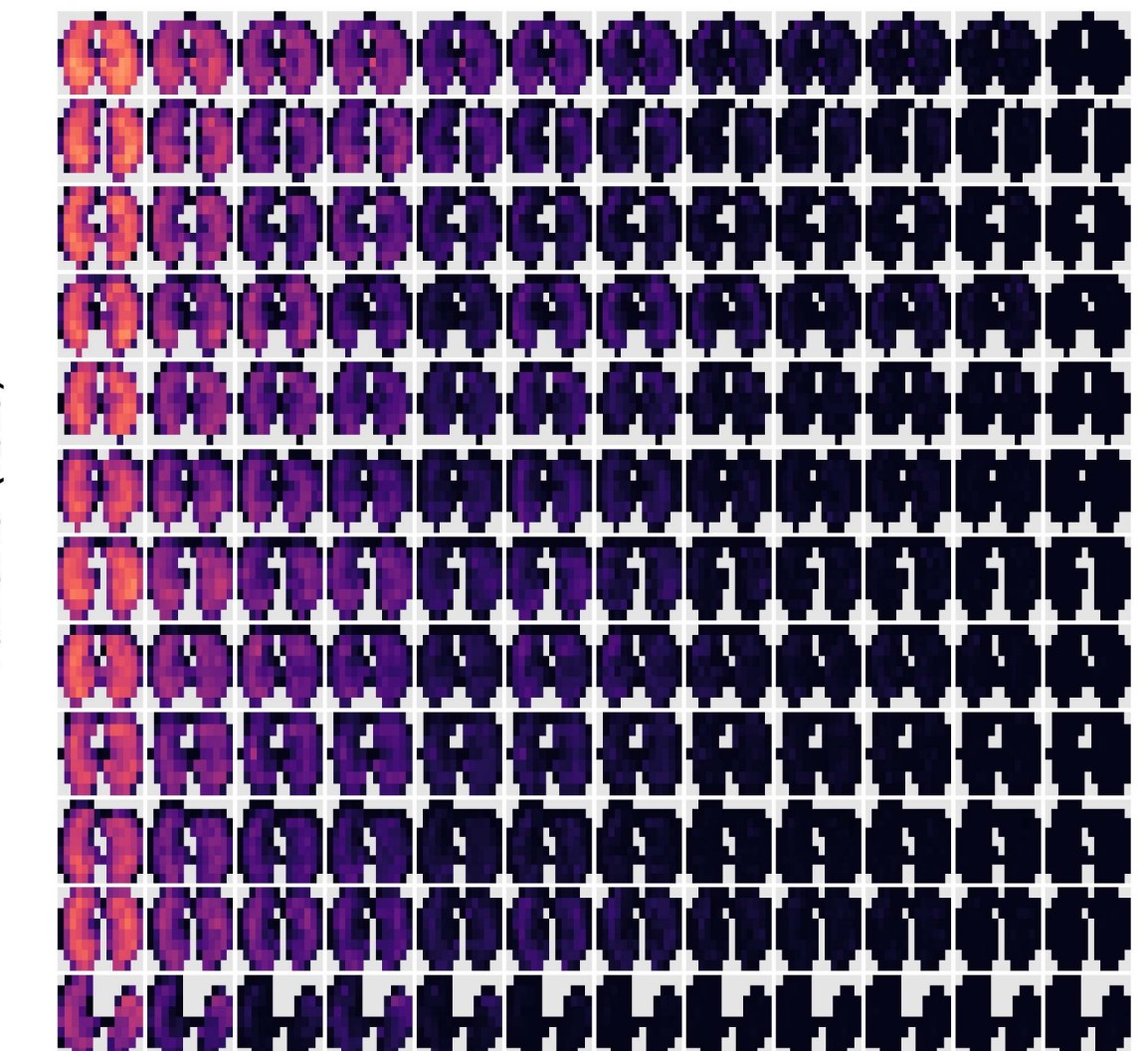

*Figure 28.* SNAP attribution maps for benign nodules across patients in the axial plane.

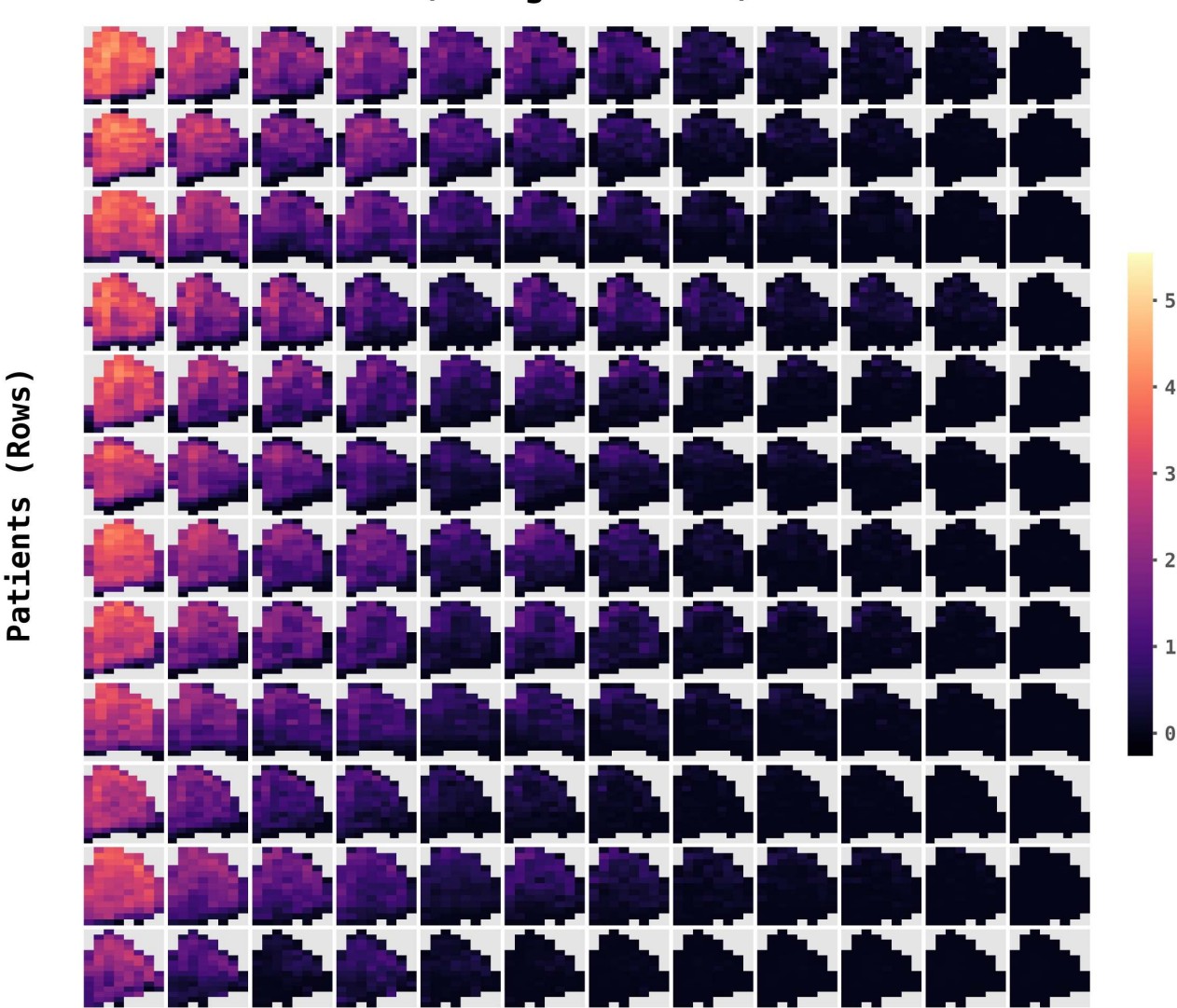

*Figure 29.* SNAP attribution maps for benign nodules across patients in the sagittal plane.

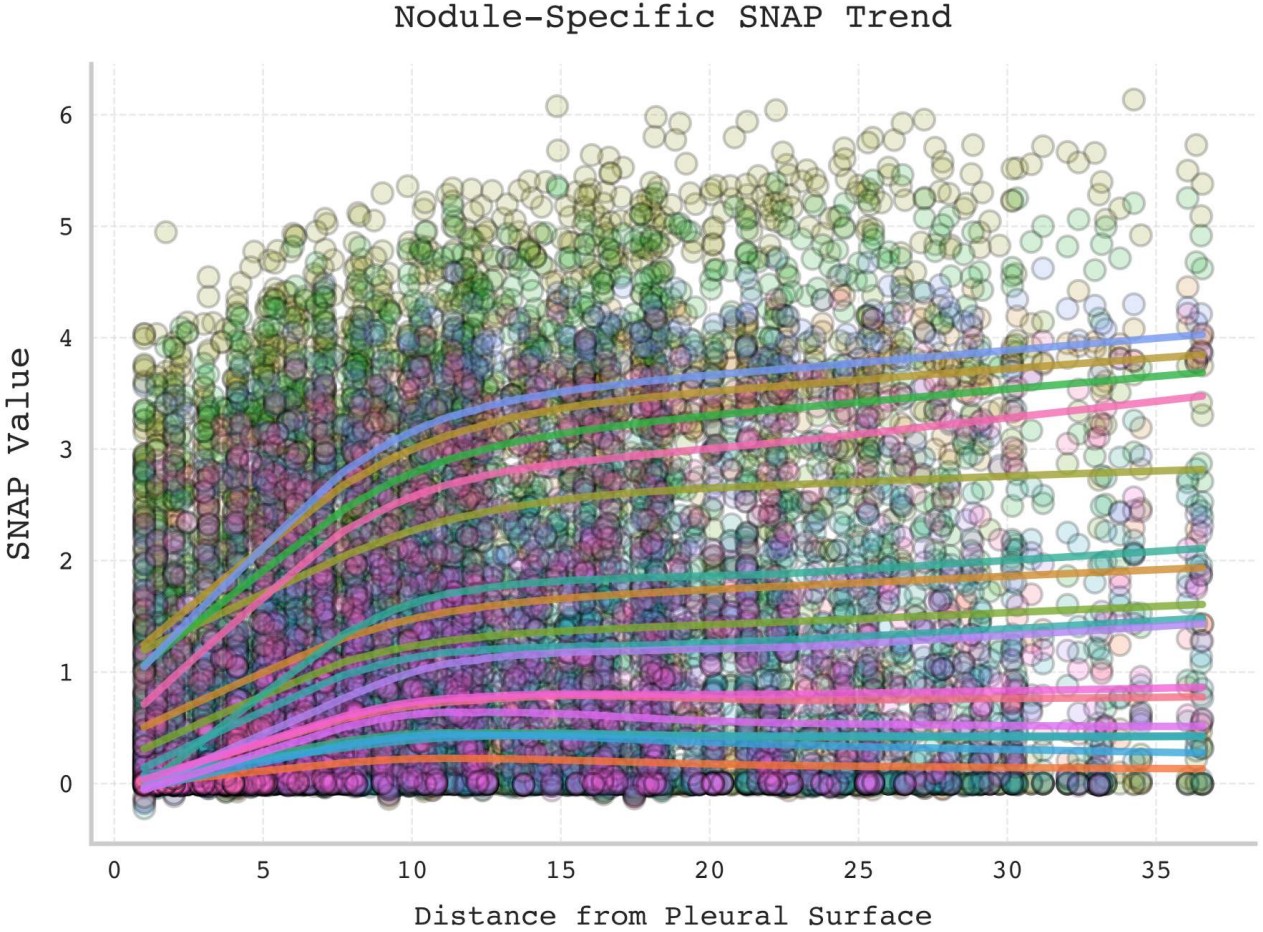

*Figure 30.* SNAP attribution values and corresponding trends across several nodules inserted into the same patient as a function of their distance from the pleural surface.

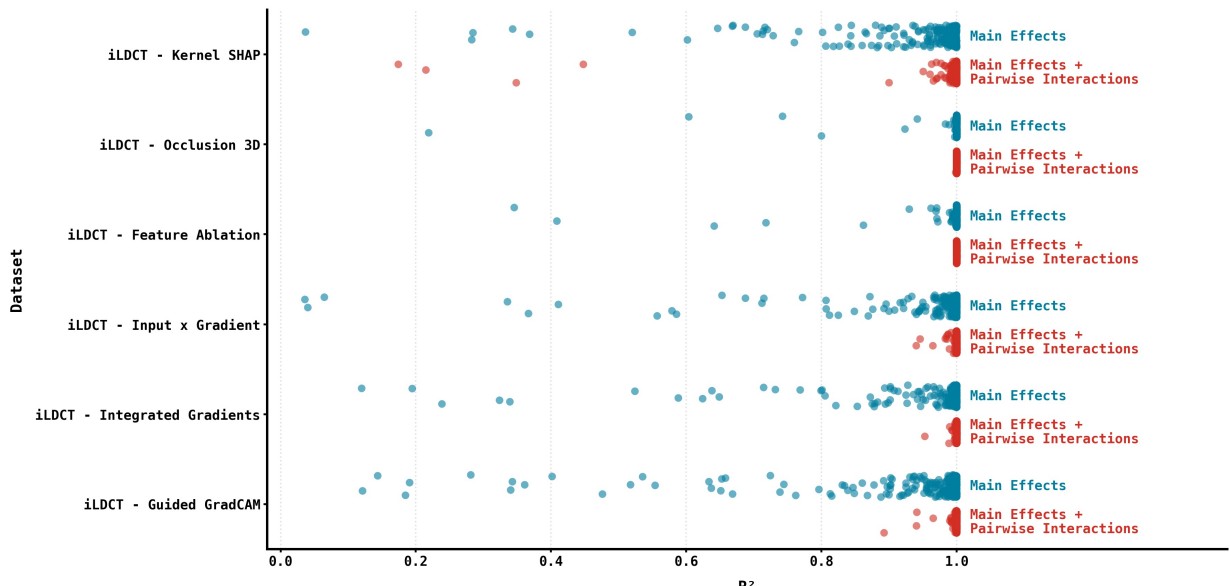

*Figure 31.* Goodness-of-fit for the SHNAP approximation on binarized regions derived from classical attribution methods.

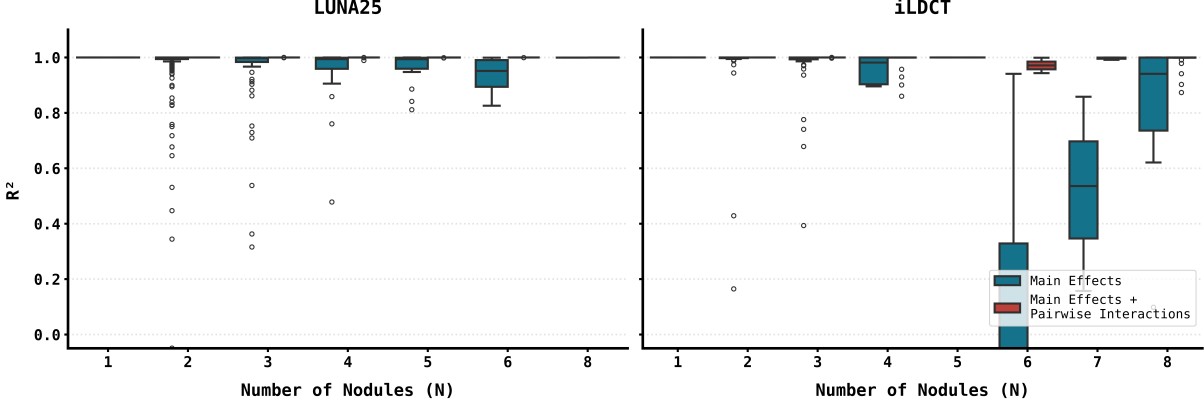

*Figure 32.* SHNAP $R^2$ goodness-of-fit stratified by the number of evaluated nodules across the LUNA25 and iLDCT datasets.

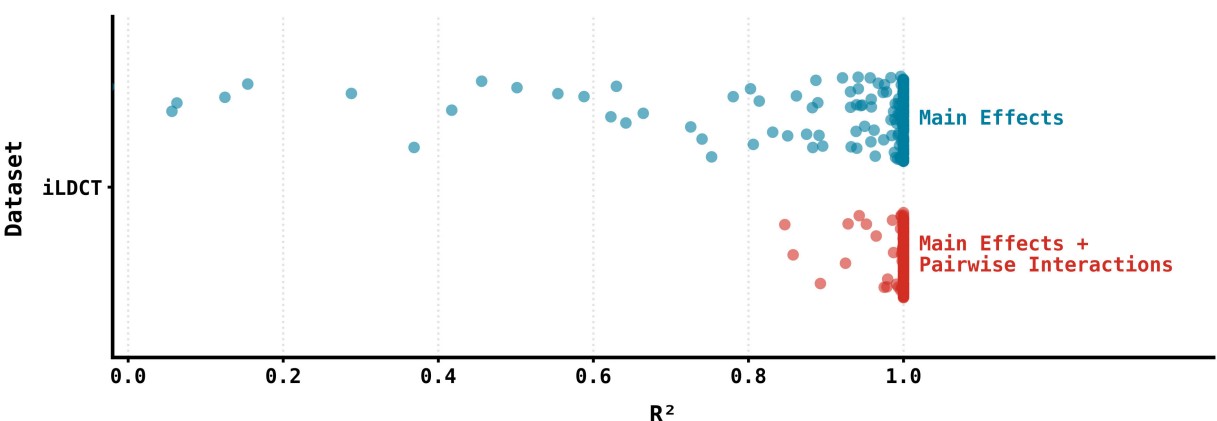

*Figure 33.* Goodness-of-fit comparing Sybil's base hazard predictions with the SHNAP approximation utilizing fully automated SwinUNETR segmentation masks.

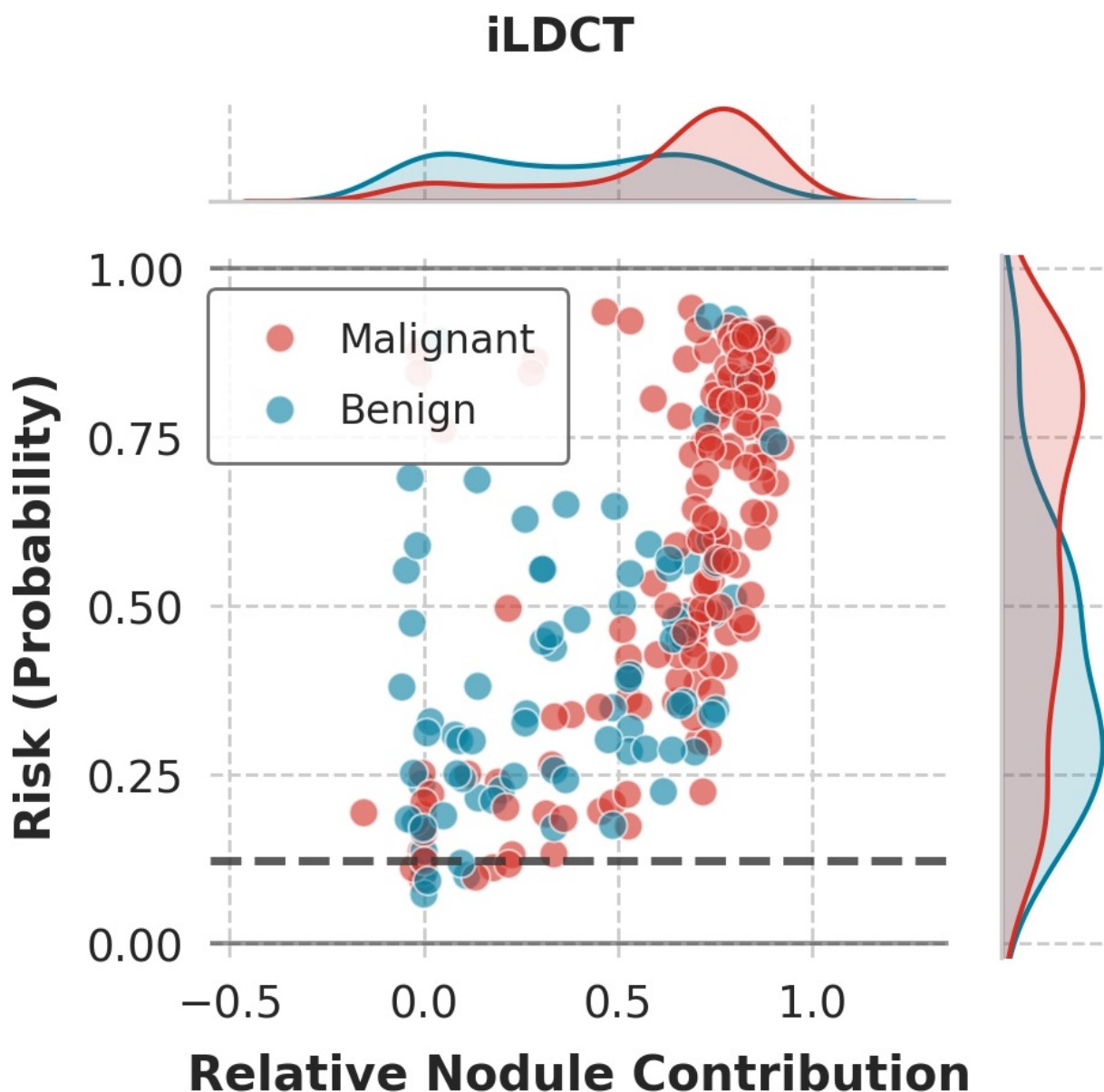

*Figure 34.* Distribution of the Relative Nodule Contribution (RNC) against predicted risk (in probability space) computed using automated SwinUNETR masks.

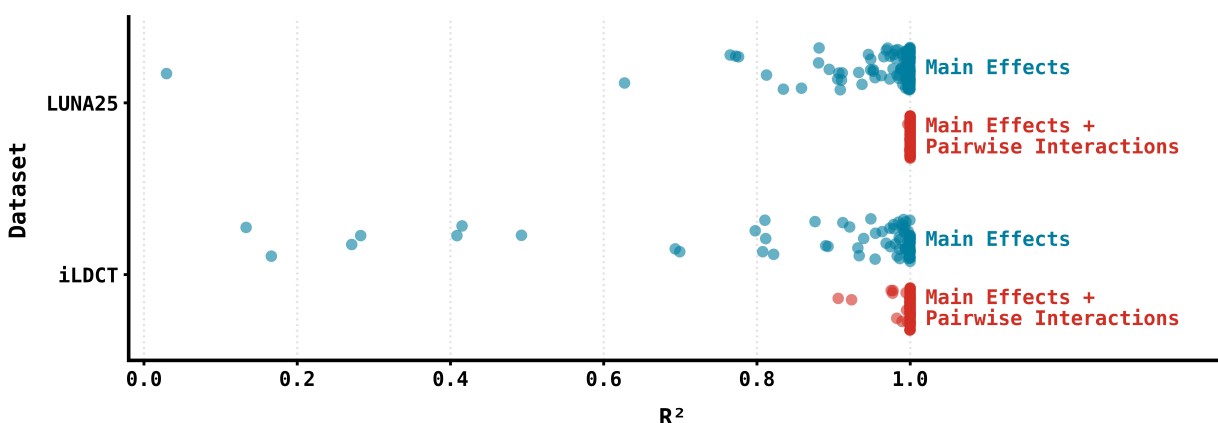

*Figure 35.* Goodness-of-fit of the SHNAP approximation in the scenario of simulated missing nodules.

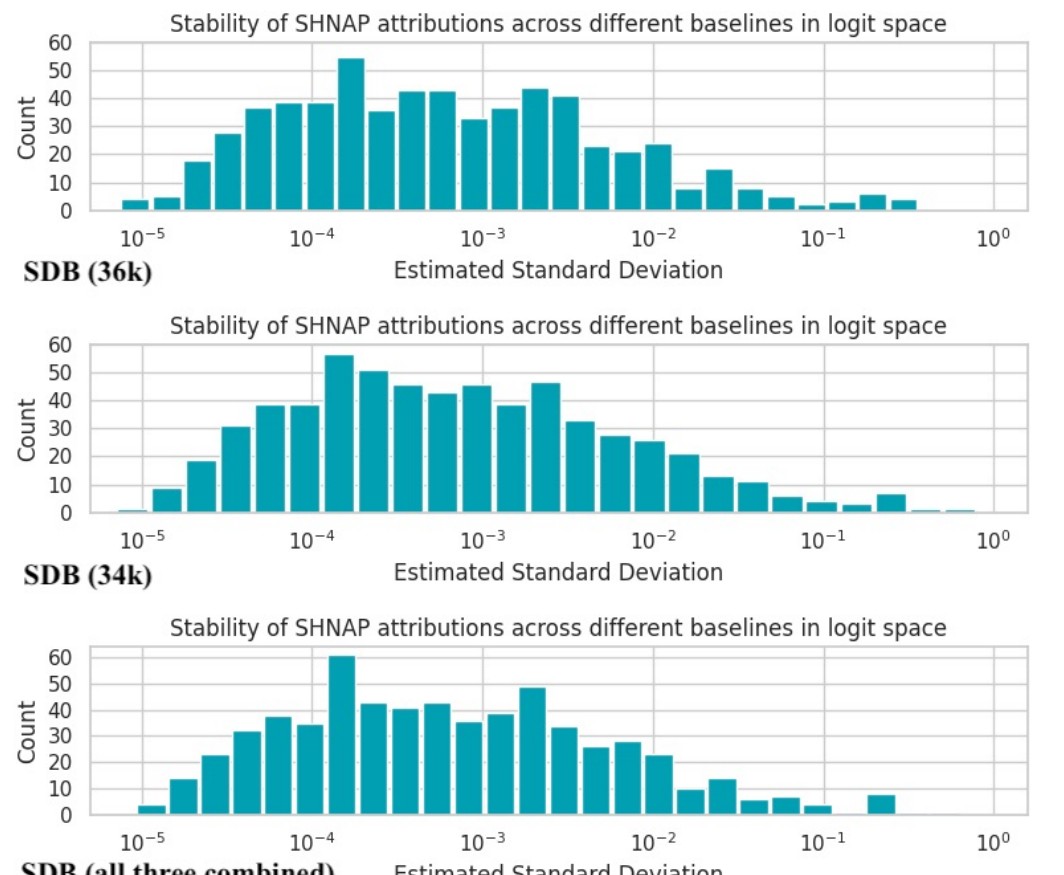

*Figure 36.* SHNAP attributions computed using different SDB training checkpoints (34k, 36k, and Main).

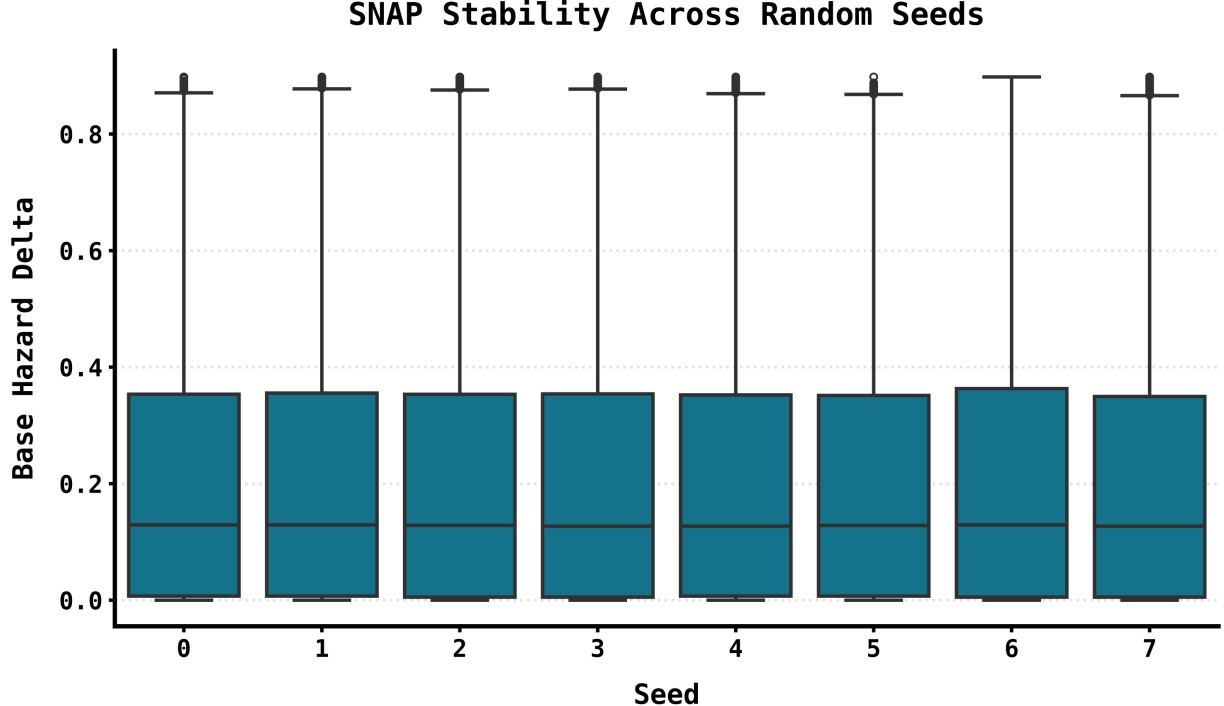

*Figure 37.* SNAP standard deviation versus mean absolute attribution evaluated across 8 different random seeds.

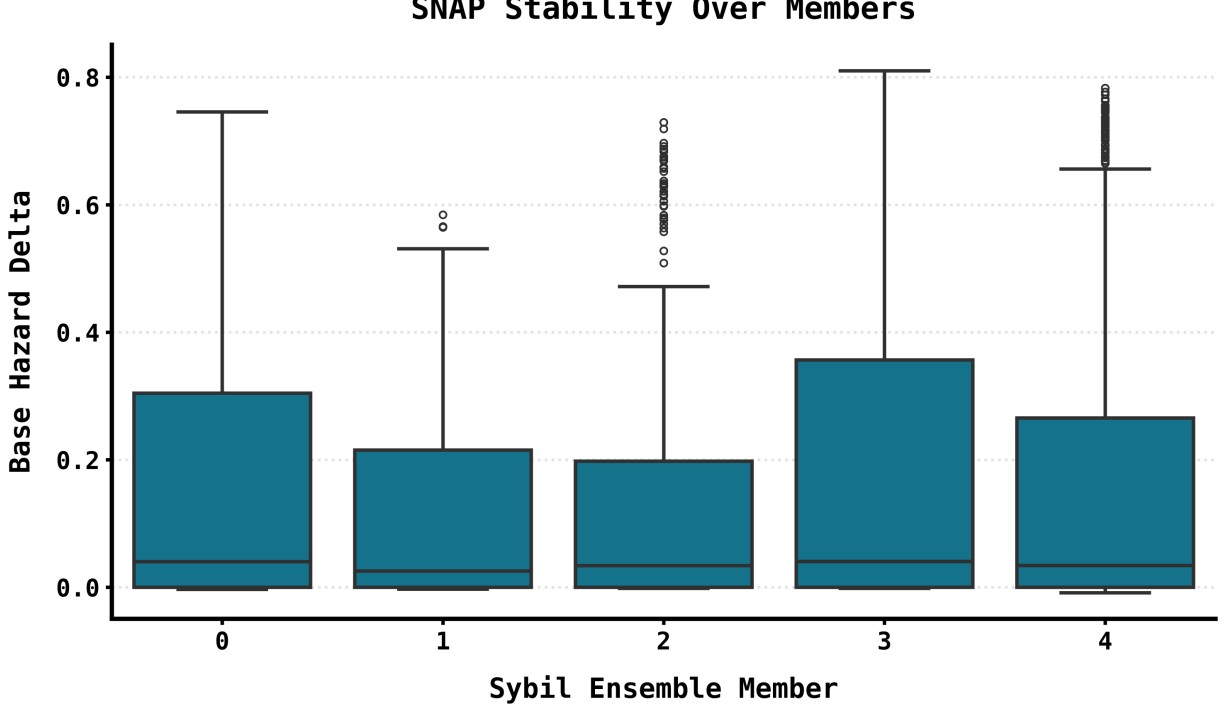

*Figure 38.* SNAP attribution distributions separated across Sybil's 5 constituent backbones.

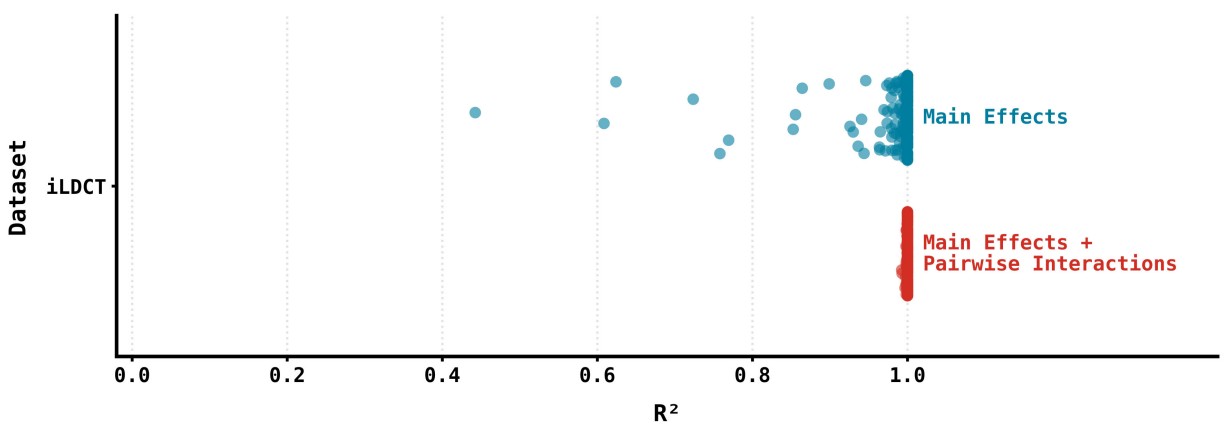

*Figure 39.* Goodness-of-fit for the CXR-LC risk prediction model against the SHNAP approximation.

# Approximation Accuracy Scaled by Nodule Count

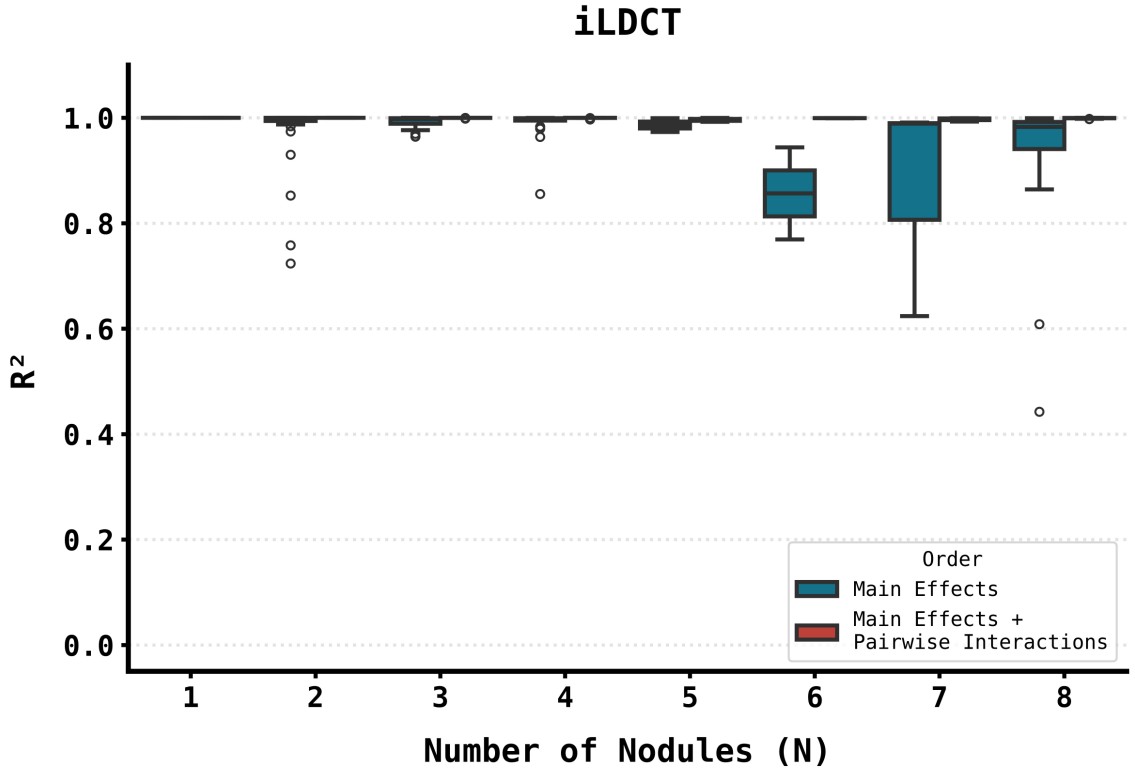

*Figure 40.* SHNAP $R^2$ goodness-of-fit stratified by nodule count for the CXR-LC model.

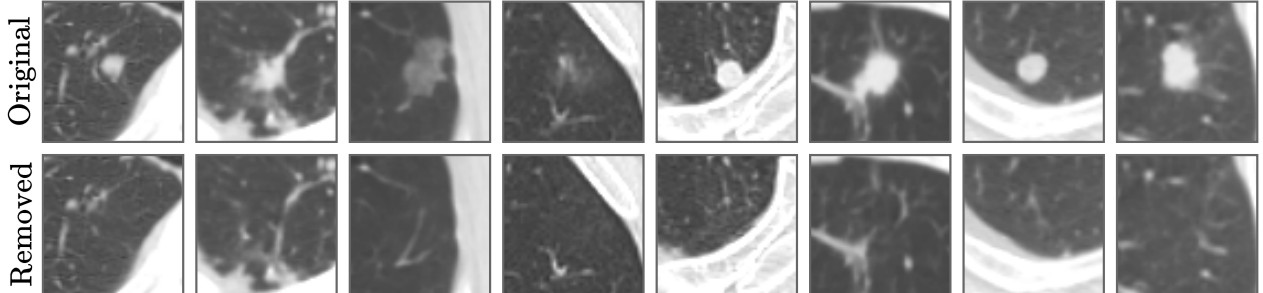

*Figure 41.* Qualitative examples of nodule removal.

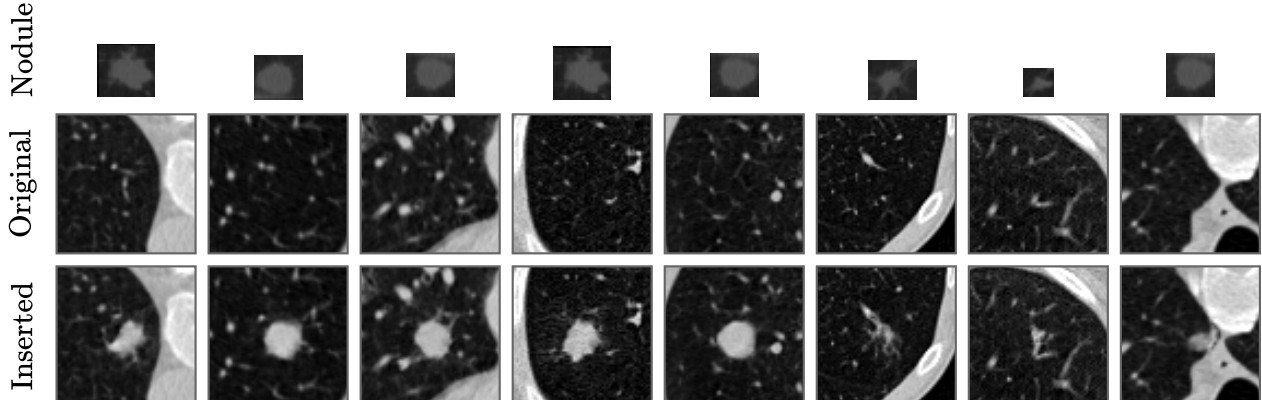

*Figure 42.* Qualitative examples of nodule insertion.

