# OpenReview forum: "Auditing Sybil: Explaining Deep Lung Cancer Risk Prediction Through Generative Interventional Attributions"
_ICML.cc/2026/Conference — ICML 2026 regular_

### Official Review · Reviewer_bWEn · 2026-03-05

**Soundness:** 4
**Presentation:** 4
**Significance:** 4
**Originality:** 4
**Overall Recommendation:** 4
**Confidence:** 4

**Summary:**

This paper argues that current validation of high-stakes medical risk models (exemplified by Sybil for lung cancer risk prediction from LDCT) remains largely observational and therefore insufficient to audit why a model works or when it fails. The authors propose S(H)NAP, a model-agnostic auditing framework that constructs generative interventional attributions via diffusion-bridge-based semantic interventions on 3D LDCT volumes. The framework instantiates two complementary attribution procedures: SHNAP (“explaining by removing”), which replaces selected pulmonary nodules with in-distribution healthy tissue to estimate n-Shapley-style main and pairwise interaction effects over nodules, and SNAP (“explaining by inserting”), which inserts nodules with controlled properties at specified spatial locations to probe sensitivity and anatomical bias. The paper formalizes a hypothesis that Sybil can be well-approximated as a linear model with pairwise interactions over a discrete set of nodules, empirically validates the approximation via local fidelity, and uses the resulting decomposition to surface clinically meaningful behaviors as well as critical failure modes (e.g., spurious reliance on clinically unjustified artifacts and radial sensitivity bias near lung boundaries). The study includes realism checks for interventions with board-certified radiologist evaluation and extensive analyses across multiple datasets and settings.

**Compliance With Llm Reviewing Policy:**

Affirmed.

**Key Questions For Authors:**

1. Intervention fidelity beyond expert realism: Can the authors provide additional quantitative evidence that the edited volumes remain in-distribution (e.g., cross-model agreement, detector/segmenter consistency, feature-space distances, or comparison across multiple generative backbones)?

2. When does the pairwise LMPI approximation fail? What patient/nodule characteristics (nodule count, size, location near pleura, emphysema burden, motion artifacts) correlate with lower local fidelity? Can the authors summarize the failure regimes and their prevalence?

3. Sensitivity to nodule detection and masking quality: How robust are SHNAP/SNAP conclusions to upstream detection errors (missed nodules, spurious nodules) and mask boundary noise? Have the authors tested perturbations of masks or alternative detectors?

**Limitations:**

The paper appropriately acknowledges that the auditing pipeline relies on partially synthetic (generatively edited) data and that generative artifacts remain a risk. I would encourage expanding this discussion along three concrete axes: (i) generator dependence (how conclusions might change with alternative diffusion bridges or training sets), (ii) upstream detection dependence (how errors in nodule proposals impact audits), and (iii) operational deployment (how audit outcomes translate into go/no-go criteria, monitoring, and remediation steps). Additionally, the paper should emphasize that interventional audits are not substitutes for prospective clinical validation, but complementary tools for mechanism verification and safety analysis.

**Strengths And Weaknesses:**

Strengths：

1. Compelling problem framing and high practical relevance. The paper squarely addresses a gap in evaluating clinically deployed (or deployment-ready) models: observational performance does not certify that the model’s mechanism is acceptable for high-stakes use. The motivation for interventional auditing is clear and timely.

2. Methodological clarity: interventions are explicitly tied to causal questions. The work cleanly distinguishes correlation-based attributions from interventional reasoning and uses generative modeling to keep edits on (or close to) the data manifold, which is essential for counterfactual-style analyses in medical imaging.

3. Well-structured decomposition with a testable hypothesis. Framing Sybil as an approximate linear model with pairwise interactions over nodules yields a concrete and falsifiable claim, rather than a purely qualitative “explainability” narrative. The local fidelity analysis provides evidence for when the approximation is adequate and when it breaks down.

Weakness:

1. Interventional validity hinges on diffusion-bridge fidelity, which remains a potential single point of failure. Although the paper conducts expert studies and several sanity checks, the core guarantee is still approximate: diffusion-based edits can introduce subtle artifacts or distribution shifts that may affect the audited model in ways not attributable to the intended semantic intervention. The paper would be strengthened by additional quantitative measures of distributional validity beyond task-level realism assessments (e.g., multi-metric detector consistency, feature-space distances, or cross-generator agreement).

2. Scope and limits of the LMPI approximation need sharper characterization. The claim that Sybil behaves like a linear model with pairwise interactions over nodules is powerful, but it risks over-generalization. It is not fully clear when the approximation is expected to hold (e.g., dependence on nodule detector quality, nodule count, nodule size regimes, or background emphysema severity). A more explicit taxonomy of failure cases—and how often they occur in practice—would increase confidence in downstream conclusions derived from the decomposition.

3. Nodule set construction and detection errors may confound both SHNAP and SNAP. The framework treats nodules as modular semantic units, but the extraction of the nodule set (and masks/coordinates) is itself a nontrivial upstream step. False positives/negatives or imperfect masks can alter coalition interventions and attribution estimates. The paper should more directly quantify sensitivity to detection quality and segmentation noise, as these are realistic deployment-time failure The paper appropriately acknowledges that the auditing pipeline relies on partially synthetic (generatively edited) data and that generative artifacts remain a risk. I would encourage expanding this discussion along three concrete axes: (i) generator dependence (how conclusions might change with alternative
diffusion bridges or training sets), (ii) upstream detection dependence (how errors in nodule proposals impact audits), and (iii) operational deployment (how audit outcomes translate into go/no-go criteria, monitoring, and remediation steps). Additionally, the paper should emphasize that interventional auditsmodes.

---

> ### Author Rebuttal · Authors · 2026-03-30
>
> We thank the reviewer for their valuable feedback. We hope these clarifications resolve all issues and suggest a score increase if satisfied with the outcome.
>
> # W1. and Q1.
>
> Our submission already contains a quantitative comparison to state-of-the-art generative modeling in medical imaging (Table 1). Below, we provide an extended comparison with additional metrics (feature-space distances) to highlight that our approach significantly outperforms prior methods, minimizing the risk of generative artifacts through better performance and expert-based validation. We also include results of two SDB training checkpoints (34k and 36k iterations) to highlight cross-generator consistency; performance remains largely identical despite weight differences.
>
> Performance across checkpoints: https://www.dropbox.com/scl/fi/o0unp45q3r6u3382yc7i5/2_sdb_quantitative_evaluation.png?rlkey=32oql44xdyv5uzn4h2mezzcgp&st=p8uqmv1y&dl=0
>
> SHNAP stability over checkpoints: https://www.dropbox.com/scl/fi/earkrmc76vwtra1v58072/2_shnap_stability_over_different_checkpoints.png?rlkey=cq4f3g4omsqz1pf13w6gz0axn&st=lwyrrgy3&dl=0
>
> We further inspect cross-generator consistency by recomputing all SHNAP explanations with these checkpoints for a 3-way comparison. We achieve a mean 0.0086 and median 0.0005 difference (probability space), highlighting extreme stability. Analyzing the results with a linear mixed-effects model, we observed near-zero coefficients for the selected checkpoints relative to the main one (0.001 ± 0.001 and 0.000 ± 0.001). Through a Repeated Measures ANOVA, we also confirm that the difference between inpainters is not statistically significant (F-statistic=1.61, GG-corrected p-value > 0.05), indicating robustness.
>
> # W2. and Q2.
>
> Based on our submission, the pairwise LMPI approximation holds with high accuracy in all considered cases, with only extremely rare (<2%) samples achieving imperfect R² that remains very high (>0.8). As mentioned (lines 258-262), these cases correspond to large nodules where SDB struggles with reconstruction, with volumes exceeding training observations due to a fixed input cube size. This is easily fixable by considering larger training cubes and does not affect the paper's conclusions in any way.
>
> To address the nodule detector dependency, we replace the expert-annotated segmentation masks for nodules with the ones from a fully automated nnUNetr model. The goodness-of-fit results below, based on these segmentations on the iLDCT dataset, almost fully recover the results observed in the original figure 4, highlighting the robustness of our framework.
>
> SHNAP fit with nnUNetr masks: https://www.dropbox.com/scl/fi/gxnf9hofdjnf2awg28rqf/1a_shnap_nnunetr_fit.png?rlkey=ao7v61sjbbgo4p0vtwjfqhlk7&st=760o9xar&dl=0
>
> SHNAP RNC with nnUNetr masks: https://www.dropbox.com/scl/fi/yvhyxxblets5yagvi8rkg/1b_shnap_nnunetr_rnc.png?rlkey=dg80oxn4uj87uibfg7iqjqczt&st=2nb4sjkh&dl=0
>
> To assess the influence of nodule count, we stratify the original goodness-of-fit from figure 4. based on it on both datasets. The results below highlights that our framework is insensitive to the number of evaluated nodules, showcasing unique robustness and ability to explain even the most complex cases.
>
> SHNAP fit by nodule count: https://www.dropbox.com/scl/fi/cwsd6bbnbv1bi3o257yg5/1_shnap_fit_by_nodule_count.png?rlkey=wa1nu5dt6jscezw78uko364wx&st=wtojoa9x&dl=0
>
> # W3. and Q3.
>
> We begin with an assessment of nodule detection dependence by simulating missed nodules using the analytically tractable formulation of n-Shapley Values. We first select patients with at least 3 nodules and compute their mean goodness-of-fit by averaging over cases with a simulated, randomly chosen missed nodule. As indicated by the table below, even when nodules are missed, goodness-of-fit remains extremely high, following or even exceeding the results from the main paper (figure 4.).
>
> SHNAP fit with simulated missing nodules: https://www.dropbox.com/scl/fi/83u5q1iplya1pw9nhfqns/2_shnap_fit_simulation.png?rlkey=lwnap6xfk3baoizy8ceo4d6c4&st=x4m94dzn&dl=0
>
> We assess generator dependence by the cross-generator consistency in our response to W1. and Q1. To assess the impact of mask boundary noise, we sweep over the volumes of the mask from 1 mm³ up to 35 mm³, and for each value recompute the SHNAP explanations based on such masks. Crucially, the results reveal a plateau of the resulting attributions around 10 mm³, which corresponds to the typical volume of masks used with the SDB model for the original results. Specifically, we observed that for over 83% of samples, the resulting attributions do not deviate from the value attained at 10 mm³ when increasing the volume up to 35 mm³ by more than 0.02 (probability space), indicating extreme robustness of SHNAP to any instabilities in nodule masks.
>
> Cumulative tolerance and stabilized samples: https://www.dropbox.com/scl/fi/cwsd6bbnbv1bi3o257yg5/1_shnap_fit_by_nodule_count.png?rlkey=wa1nu5dt6jscezw78uko364wx&st=4phix8al&dl=0

---

> > ### Author Rebuttal · Reviewer_bWEn · 2026-04-03
> >
> > Thanks for the rebuttal. All my concerns are sloved, I will keep my original rating.

---

> > > ### Author Response · Authors · 2026-04-03
> > >
> > > We sincerely thank the reviewer for engaging in the rebuttal and confirming that all concerns are fully resolved.
> > >
> > > Given that a "Weak Accept" evaluation reflects remaining unresolved weaknesses, we respectfully ask if the reviewer would consider raising the score to a standard Accept to reflect this fully resolved status. If there are any remaining unmentioned weaknesses, we remain eager to address them.

---

### Official Review · Reviewer_FPMx · 2026-03-08

**Soundness:** 3
**Presentation:** 3
**Significance:** 3
**Originality:** 2
**Overall Recommendation:** 4
**Confidence:** 4

**Summary:**

The paper proposes to analyse the behavior of the model Sybil (a famous model designed for cancer risk prediction based on lung CT scan). First, the authors want an attribution value for the background, for each nodule and for each pair of nodules. The authors formulate the risk prediction produced by Sybil as a linear model with pairwise interaction and use the n-Shapley values to compute the attribution coefficients. To obtain this attribution for each nodules (and pairs of nudules), it is necessary for each image containing nodules (segmented) to have images containing only some of the nodules and so to remove them. To perform this inpainting, a diffusion model is used. Second, to analyse the impact of the localization of the nodules for the model prediction, the authors propose to generate nodules (with the same diffusion model) and compute the attributions as previously described.

**Compliance With Llm Reviewing Policy:**

Affirmed.

**Final Justification:**

The authors have addressed all my comments. Despite the paper’s limitations (it is mainly application-oriented work, bringing together state-of-the-art methods for a very specific application), I believe it may be of interest to the community, and I have increased my rating from Weak reject to Weak accept.

**Key Questions For Authors:**

- A comparison to classical attributions methods

**Limitations:**

/

**Strengths And Weaknesses:**

Strengths:
- The performed analysis is of great interest for clinical use of the model.
- Formulating the problem as the contribution of the background, each nodule and each pair of nodules, as well as the method for solving the problem with the n-Shapley values, is intelligent.
- The analysis include clinicians that have validated the inpainting method visually.
- The proposed inpainting method outperforms the state-of-art methods.
- The code is provided.
- The paper is well written.

Weaknesses:
- The contribution is more application-oriented than methodological.
- The method requires the segmentation of the nodules to compute the attributions.
- To assess whether the images generated by the diffusion model are not out of distribution, the authors involoved clinicians. However, just because clinicians cannot see anything visually does not mean that the model is not affected (counterfactual examples have shown that  perturbations invisible to the eye can drastically influence a model's decision). One way to assess the impact of inpainting and model generation could be to look at the difference in prediction for an image and the same image where the nodule has been removed and then regenerated with the same mask by the diffusion model.
- A comparison to classical attributions methods would be a plus. The authors mentioned the attention based region but this is not really explained or developed. We would like to see attribution maps.
- An analysis of the impact of the number of nodules would be interesting.
- In Figure 6, the legend and notations in the images are unclear: what is the baseline? which nodules are malignant/benign?

---

> ### Author Rebuttal · Authors · 2026-03-30
>
> We thank the reviewer for their valuable feedback. We hope these clarifications resolve all issues and suggest a score increase if satisfied with the outcome.
>
> # W3. In-distribution assessment
>
> Our submission already assesses in-distribution generation against state-of-the-art techniques (Table 1). Updated metrics below confirm our approach significantly outperforms prior methods, minimizing out-of-distribution risks. Extending these results with two SDB checkpoints (34k and 36k iterations) further strengthens this claim.
>
> Performance across SDB checkpoints: https://www.dropbox.com/scl/fi/o0unp45q3r6u3382yc7i5/2_sdb_quantitative_evaluation.png?rlkey=32oql44xdyv5uzn4h2mezzcgp&st=p8uqmv1y&dl=0
>
> Per the Reviewer's suggestion, assessing inpainting impact by comparing original predictions to those from removing and regenerating the nodule (using identical masks on the iLDCT dataset) yields a probability difference of -0.02 ± 0.08 (mean ± std). This negligible impact demonstrates inpainting stability, with minor variances stemming from SDB's stochasticity.
>
> # W4 and Q1. Comparison to classical attribution methods
>
> We include a large-scale comparison to white-box gradient-based (Saliency, Guided Grad-CAM, Input X Gradient, Integrated Gradients) and black-box perturbation-based (Feature Ablation, KernelSHAP, Occlusion 3D) methods. We evaluate localization (Relative Regional Attribution (RRA) using nodules as ground truth), faithfulness (insertion/deletion curves, AUC Pert), complexity (Sparseness), and exactness (Completeness).
>
> Comparison: https://www.dropbox.com/scl/fi/1xsl9nfd2iqal4wlxb7ux/1_quantitative_evaluation.png?rlkey=rye21fq04llkwt0v2lcb2cl2x&st=0tsbnm9f&dl=0
>
> Our method outperforms standard benchmarks in explanation quality. Typical metrics often rely on unrealistic assumptions (e.g., OOD perturbations), whereas true value lies in exact model correspondence. SHNAP uniquely enables analytically tractable, exact faithfulness evaluation by computing R² across all coalitions—impossible for Kernel SHAP. Furthermore, SHNAP provides actionable semantic insights, avoiding the interpretative ambiguity of post-hoc explanations.
>
> Additionally, classical attributions can be binarized to indicate high-importance regions, akin to our attention-based regions. SHNAP goodness-of-fit results on such regions (below) emphasize our framework's adaptability, perfectly approximating nearly all samples via main effects and pairwise interactions.
>
> SHNAP with classical attribution methods: https://www.dropbox.com/scl/fi/xiy03xn67rufxskn3l8dy/2_faithfulness.png?rlkey=e833wh67l1m766j7e3gb2w0cg&st=boimutbw&dl=0
>
> # W5. Impact of the number of nodules
>
> Stratifying the goodness-of-fit of SHNAP by nodule count on both datasets demonstrates that our approach is essentially uninfluenced by how many nodules are evaluated, proving that even the most complex scenarios are effectively explained.
>
> SHNAP fit by nodule count: https://www.dropbox.com/scl/fi/cwsd6bbnbv1bi3o257yg5/1_shnap_fit_by_nodule_count.png?rlkey=wa1nu5dt6jscezw78uko364wx&st=wtojoa9x&dl=0
>
> # W2. Requirement of segmentation masks
>
> To prove that our method is not limited to manual segmentation, we computed masks using the nnUNetr model on the entire iLDCT dataset and repeated the SHNAP explanations. The goodness-of-fit statistics for the resulting LMPI (akin to Figure 4) show that performance remains essentially identical to expert-annotated masks, empirically confirming our framework does not require human-in-the-loop approaches.
>
> SHNAP fit with nnUNetr masks: https://www.dropbox.com/scl/fi/gxnf9hofdjnf2awg28rqf/1a_shnap_nnunetr_fit.png?rlkey=ao7v61sjbbgo4p0vtwjfqhlk7&st=760o9xar&dl=0
>
> SHNAP RNC with nnUNetr masks: https://www.dropbox.com/scl/fi/yvhyxxblets5yagvi8rkg/1b_shnap_nnunetr_rnc.png?rlkey=dg80oxn4uj87uibfg7iqjqczt&st=2nb4sjkh&dl=0
>
> # W1. Contribution type
>
> We maintain that the contributions of our paper are largely methodological, with practical experiments highlighting their empirical effectiveness:
> * We introduce the first generative framework for pulmonary nodule manipulation bypassing the need for nodule labels during training, enabling the use of tens of thousands of unlabeled 3D volumes.
> * Our approach is the first to support simultaneous nodule removal and insertion with known malignancy, creating new opportunities for synthetic data augmentation and targeted fine-tuning.
> * This work represents the first application in medical imaging to evaluate attributions at the level of modular semantic units (pulmonary nodules). This conceptual transition allows us to employ n-Shapley Values as an analytically tractable and exactly computable solution to model explainability.
>
> # W6. Figure 6
>
> As indicated (line 133), the baseline represents Sybil's prediction on a 'nodule-free' LDCT scan, where all detected nodules are synthetically replaced with healthy tissue. Legend labels: white squares indicate a benign case, while black squares indicate that the nodule is malignant.

---

> > ### Author Rebuttal · Reviewer_FPMx · 2026-04-03
> >
> > The authors have responded to all my comments. The comparison with state-of-the-art methods and the demonstration of the feasibility of using automatic segmentation strengthen the paper.

---

> > > ### Author Response · Authors · 2026-04-05
> > >
> > > We appreciate the Reviewer’s constructive participation in the rebuttal and their positive reception of our work. If any outstanding concerns remain, we would be happy to provide additional clarification.

---

### Official Review · Reviewer_yrMN · 2026-03-13

**Soundness:** 2
**Presentation:** 3
**Significance:** 2
**Originality:** 3
**Overall Recommendation:** 3
**Confidence:** 4

**Summary:**

The authors propose S(H)NAP  for auditing Sybil, a frontier model for lung cancer risk prediction from a single CT scan. To achieve this, the authors approximate Sybil with a Linear Model with Pairwise Interactions (LMPI) over detected pulmonary nodules (i.e., a baseline term, nodule main effects and pairwise interaction terms), which is achieved by n‑Shapley Values (nSV, with n=2).
The authors leverage System-Embedded Diffusion Bridges (SDB) to perform interventions for counterfactual explanations: nodule removal (inpainting healthy tissue in a masked region) and nodule insertion (transplanting a nodule into a new scan). The fidelity of these SDB-based interventions is validated via blinded radiologist evaluations.
Using these tools, the authors introduce two complementary auditing procedures: SHNAP (explain-by-removing) to attribute Sybil’s prediction to individual nodules and their pairwise interactions, and SNAP (explain-by-inserting) to probe Sybil’s spatial sensitivity and reasoning by measuring prediction shifts induced by inserting nodules at chosen locations. Together, SHNAP and SNAP provide some insights into Sybil’s decision-making.

**Compliance With Llm Reviewing Policy:**

Affirmed.

**Final Justification:**

I appreciate the effort the authors have put into demonstrating the robustness of the framework. However, the ambiguity regarding whether these flaws are systematic to the "Sybil approach" or just specific to these training instances, combined with the lack of a demonstrated path to mitigation, leads me to maintain my current score of Weak Reject.

**Key Questions For Authors:**

1. I am concerned about the computational scalability of the proposed method:
    1) Removal-based explanation. Each scan needs to perform removal 2^N times; while it can be reduce the number of diffusion bridge generations to N (one per detected nodule), it appears that evaluating Sybil still requeis  2^N forward.
    2) Insertion-based explanation. The insertion-based probing introduces a nodule at a target coordinate in a different scan. Please clarify: (i) how are target coordinates  selected in practice and how do you ensure anatomical plausibility (e.g., avoiding airways/vessels/pleura)? (ii) If figures (e.g., fig 7) needs 5000 insertion, how to make it scalable?
2. Reliability / generalizability. Since the analysis is conducted on a single Sybil instance, how reliable are the conclusions? Would different Sybil models trained with different settings exhibit materially different behaviors and therefore different SHNAP/SNAP findings? Also, to what extent could the conclusions be affected by the SDB intervention mechanism itself (e.g., artifacts or distribution shift introduced by removal/insertion)?
3. Solutions. While the paper exposes critical reasoning misalignments, is there a reliable solution?

Minor:
Missing experimental details (Appendix C.2).

**Limitations:**

While the authors acknowledge generative artifacts, they should further discuss the computational bottlenecks, the anatomical plausibility of synthetic insertions, and the lack of actionable solutions for the identified model biases.

**Strengths And Weaknesses:**

Strengths:
1. The paper is technically sound with verified hypothesis, where the attribution method (SHAP) and generation method (diffusion bridge) are used appropriately.
2. The paper is clearly written and well structured.
3. The blinded radiologist study is a strong addition and provides convincing evidence of generation fidelity.
4. The paper provide critical reasoning misalignments of Sybil, including both the case by case analysis as well as some  results at the statistical level.

Weaknesses:
1. The framework faces significant overhead.
2. The audit is restricted to a single instance of Sybil, leaving it unclear if the identified biases (e.g., radial sensitivity) are consistent across different model seeds or settings.
3. While the framework is highly effective at exposing reasoning misalignments and spurious correlations, the paper does not propose or evaluate any actionable solutions to mitigate or fix these identified flaws.

---

> ### Author Rebuttal · Authors · 2026-03-30
>
> We thank the reviewer for their valuable feedback and hope these clarifications encourage a score increase.
>
> # W1. and Q1. Computational overhead and anatomical plausibility.
>
> Practically, the pulmonary nodule count (N) is typically low (1-4, Figs. 10/11), ensuring the overhead remains manageable. Comparing SHNAP to black-box methods (Kernel SHAP, Occlusion, Feature Ablation) reveals that it is 3x-9x faster in practice, even with large perturbations for these approaches (32³ for Occlusion/Ablation, 16³ for Kernel SHAP), by operating at a conceptual rather than volumetric level.
>
> | Attribution Method | Execution Time [mm:ss] |
> | :--- | :--- |
> | Kernel SHAP (2048 evaluations) | 70:20 |
> | Occlusion | 56:20 |
> | Feature Ablation | 26:05 |
> | **SHNAP** (8 nodules) | **08:08** |
>
> Conversely, SNAP requires many insertions (~1000/image) for high resolution. A single SNAP map takes ~100 minutes on one GPU (comparable to black-box methods) but scales linearly across devices, providing unique, actionable insights.
>
> For SNAP target coordinates, we restrict analysis to regions of interest using lung masks from TotalSegmentator (Section C.4) divided into uniform cubes. Anatomical plausibility (airway/vessel alignment, avoiding pleura) is maintained by the SDB model and boundary restrictions imposed by the lung segmentation mask, and confirmed via expert validation (Sec. 5.1), which also included such samples.
>
> # W2. and Q2. Reliability.
>
> Although presented as a single model, Sybil is an ensemble of 5 identical backbones (trained in leave-one-out fashion on NLST). Recomputing SNAP for each backbone reveals statistically distinct attribution distributions (Anderson-Darling p < 0.001; pairwise Kolmogorov-Smirnov confirms difference). This proves SNAP generalizes to capture global Sybil biases and distinct backbone patterns.
>
> SNAP distribution per backbone: https://www.dropbox.com/scl/fi/wp4ygexzmndtsjgp2hlmh/1a_snap_stability_over_members.png?rlkey=pnpwfk6rvmx3xtqa1kumes9sm&st=6l5c041d&dl=0
>
> SNAP per-backbone statistics: https://www.dropbox.com/scl/fi/iqt6goa1kt3br9rd0zyur/1b_snap_stability_over_members_stats.png?rlkey=380q23rjvl4x6i9uhpha6a6uq&st=kcddh8md&dl=0
>
> We must argue that focusing solely on Sybil is not a weakness given its great scientific importance (lines 48–51). To demonstrate broader applicability, we provide an independent audit of a chest-X-ray risk model (CXR-LC [1]) using simulated LDCT-to-X-ray conversion, successfully extending our framework.
>
> Fit: https://www.dropbox.com/scl/fi/lvkma7ti3xoib41kj6o38/1a_faitfhulness.png?rlkey=bddlogk7zss7naiqkmqw2y4ji&st=qj2cicbv&dl=0
>
> Fit by nodule count: https://www.dropbox.com/scl/fi/kby8rgqs1m5qiyrqetfz6/1b_faitfhulness_by_nodule_count.png?rlkey=me3t6ispzby7waql7ldqcgh7r&st=tjgewm1c&dl=0
>
> Fit statistics: https://www.dropbox.com/scl/fi/aepvf4i6pzhgnlyrkrsyq/1c_faitfhulness_table.png?rlkey=mq81tnxszemwi51y8rdfptsfl&st=adztfbth&dl=0
>
> The flaws are unlikely due to chance, and conclusions are not driven by generative artifacts or distribution shifts, as shown by previous and new result. Board-certified radiologists could not distinguish synthetic from original images. Extended quantitative comparisons (updating Table 1 below) show our approach minimizes distribution shift risks across multiple SDB checkpoints (34k and 36k iterations), maintaining stable performance without adversarial patterns.
>
> SHNAP stability over checkpoints: https://www.dropbox.com/scl/fi/earkrmc76vwtra1v58072/2_shnap_stability_over_different_checkpoints.png?rlkey=cq4f3g4omsqz1pf13w6gz0axn&st=lwyrrgy3&dl=0
>
> Performance across SDB checkpoints: https://www.dropbox.com/scl/fi/o0unp45q3r6u3382yc7i5/2_sdb_quantitative_evaluation.png?rlkey=32oql44xdyv5uzn4h2mezzcgp&st=p8uqmv1y&dl=0
>
> Furthermore, SNAP attributions are insensitive to initial random noise, as presented in the results below.
>
> Stability over seeds: https://www.dropbox.com/scl/fi/ghxsxb1jbtwpaqcqhnh0o/1a_snap_stability_across_random_seeds.png?rlkey=6cj4d55rf12pmbvgbo3kz1yio&st=i3tnywkl&dl=0
>
> Statistics over seeds: https://www.dropbox.com/scl/fi/rd0mdsexprrpegxbcmbh8/1b_snap_stability_across_random_seeds.png?rlkey=2cn0cwsw2zvyvx1lglng9sxbz&st=tpd6bpfz&dl=0
>
> # W3. and Q3. Mitigation.
>
> Primary mitigation strategies stem directly from our methods. Because synthetic inpaints remain in-distribution, they provide excellent data augmentations—e.g., inserting malignant nodules into samples showing spurious correlation. Alternatively, procedurally generated or expert-based SHNAP maps could be used to regularize the model during training.
>
> While we do not explicitly propose mitigation strategies, this should not be viewed as a weakness. Uncovering biases and rigorously evaluating our approach required over 20 pages; extending the scope to bias mitigation is simply infeasible for this submission.
>
> [1] Lu, M. T. et al., _CXR-LC: Deep learning using Chest Radiographs to Identify High-risk Smokers for Lung Cancer Screening_, _Annals of Internal Medicine_, 2020.

---

> > ### Author Rebuttal · Reviewer_yrMN · 2026-04-03
> >
> > Thank the authors for detailed rebuttal. While the authors have addressed some of my concerns, several critical issues remain:
> >
> > 1. Regarding the response to W2 & Q2, the authors provided new results showing that the 5 different backbones of Sybil exhibit "statistically distinct attribution distributions". If the auditing tool yields different findings for different seeds of the same model architecture, it suggests that the "reasoning misalignments" identified may be artifacts of training stochasticity rather than inherent, systematic flaws in the Sybil model itself.
> >
> > 2. Regarding the response to W3 & Q3, the authors argue that mitigation is out of scope.  However, for a paper focused on "auditing," the ultimate goal is to improve model safety and performance. Without a "proof-of-concept" for the solution, it remains unproven whether the identified biases are even fixable using the proposed generative interventions.
> >
> > Therefore, I will maintain my score.

---

> > > ### Author Response · Authors · 2026-04-03
> > >
> > > We thank the reviewer for their constructive and detailed feedback regarding our rebuttal. We provide further details below, hoping that they resolve any remaining ambiguities.
> > >
> > > # W2 & Q2
> > >
> > > Sybil is an ensemble of 5 identical architectures with distinct learned weights. Therefore, Sybil's output is always the average of all backbones, while individual backbones are never used in isolation for inference. The requested experiment simply shows that each backbone plays a different role in the final prediction, and their combined effect is what is mainly analyzed in our submission -- assuming the perturbations are _correct_ (for which we provided strong evidence in the submission and initial response), our audit reveals the flaws of the actual model used by practitioners and scientists.
> > >
> > > The experiment reveals an expected behavior -- training an ensemble in a leave-one-out fashion, where each backbone has a unique validation split, is performed to obtain _diverse_ models with _distinct_ behaviors that jointly 'vote' for the final prediction, offering an actual mitigation for the influence of training stochasticity. Statistically distinct attribution distributions are empirical proof that this diverse behavior is combined during Sybil's inference under the hood.
> > >
> > > Finally, while resolving exactly *why* Sybil possesses these "reasoning misalignments" is intractable, deep learning models notoriously learn spurious correlations. We do not claim to definitively prove whether these stem from an exactly specified phenomenon; our core contribution is diagnosing the flawed reasoning itself in a manner impossible with classical methods. However, numerous SHNAP examples and our *global* statistical analysis of SNAP (verified to be independent from individual patients and nodules) provide solid empirical proof that Sybil has indeed learned these spurious correlations.
> > >
> > > # W3 & Q3
> > >
> > > Once again, we thank the reviewer for highlighting the need for potential mitigation strategies. We have described them in our initial response and will include them in the final version of the paper.
> > >
> > > Regarding the scope, we completely agree that the ultimate goal of the field is to improve model safety and performance. However, rigorously identifying a model's vulnerabilities is the mandatory prerequisite to achieving that safety. Our work focuses exclusively on this foundational auditing step. We do not claim that the generative interventions can fix the identified biases, nor do we assert that these specific flaws are inherently fixable. Instead, we perform an audit—a systematic, independent evaluation of the model's effectiveness and vulnerabilities. Our primary contribution is meticulously demonstrating that Sybil does not align with expert clinical reasoning. Revealing these inefficiencies serves as a vital warning sign for practitioners, preventing unsafe deployment and establishing the necessary groundwork for future mitigation research.
> > >
> > > # Audits published in the past without mitigation strategies
> > >
> > > We also list several papers published at A* conferences that, just as our work, reveal flaws of very important models in a given domain by performing an audit. These papers do not focus on simultaneously fixing these issues, which shows that the value of such works is high within the scientific community.
> > >
> > > [Language Models Don't Always Say What They Think: Unfaithful Explanations in Chain-of-Thought Prompting (NeurIPS 2023)](https://arxiv.org/abs/2305.04388)
> > >
> > > [Do ImageNet Classifiers Generalize to ImageNet? (ICML 2019)](https://arxiv.org/abs/1902.10811)
> > >
> > > [Unmasking Clever Hans Predictors and Assessing What Machines Really Learn (Nature Communications 2019)](https://arxiv.org/abs/1902.10178)
> > >
> > > We would be happy to provide the results of any further experiments that the reviewer would propose, which could further address the reviewer's concerns. We believe that, at this point, we have provided multiple independent verifications of the validity of our audit (see also other responses) and that the paper's contributions are significant and broad enough.

---

### Official Review · Reviewer_x26A · 2026-03-16

**Soundness:** 3
**Presentation:** 3
**Significance:** 2
**Originality:** 2
**Overall Recommendation:** 4
**Confidence:** 3

**Summary:**

In this paper, the authors proposed a post-hoc explainability technique called S(H)NAP to provide nodule-level explanations for lung cancer risk models, specifically a model called Sybil. The proposed S(H)NAP consists of two main components: SHNAP and SNAP. Given a lung cancer risk model (e.g., Sybil) and a particular patient's CT scan, the SHNAP (SHapley Nodule Attribution Profiles) technique explains the model's risk prediction by computing the 2-Shapley values of detected nodules (where nodule removal is achieved by replacement with synthetic healthy tissues) and approximating the model prediction using a linear model with pairwise interactions (LMPI). The SNAP (Substitutive Nodule Attribution Probing) technique elucidates the model's spatial bias in risk prediction by inserting synthetic nodules into various lung locations and measuring the changes in the model-predicted risk. In experiments, the authors applied their S(H)NAP to explain lung cancer risk predictions from a model called Sybil, and uncovered some flaws in the model (e.g., benign nodules driving high risk predictions, ignoring malignant pleural nodules, using clinically unjustified artifacts, radial sensitivity bias).

**Compliance With Llm Reviewing Policy:**

Affirmed.

**Ethical Review Concerns:**

N/A.

**Final Justification:**

My concerns have been mostly addressed by the authors, and I have increased my score to 4: Weak accept.

**Key Questions For Authors:**

1. Why does the proposed S(H)NAP only focus on explaining the base hazard logit of the Sybil model?
2. Are there other lung cancer risk prediction models that can benefit from S(H)NAP?

**Limitations:**

Yes, the authors have adequately discussed the limitations and potential negative societal impact of their work.

**Strengths And Weaknesses:**

Strengths:
- The paper presents an interesting extension of using n-Shapley values to explain risk predictions made by a medical AI model, based on known medical features (e.g., nodules).
- The paper uses generative models to "erase" and insert nodules, to ensure that the perturbed images stay in-distribution.
- The paper presents a comprehensive analysis of a lung cancer risk prediction model (Sybil), and exposes many potential flaws of the model despite its state-of-the-art performance.

Weaknesses:
- The scope of the paper is quite narrow. The paper focuses almost exclusively on a single medical AI model, Sybil, and does not demonstrate the generality of their approach.
- There is limited methodological advance of the proposed method, since the method relies on n-Shapley values and existing generative AI techniques.
- There is a lack of quantitative evaluations on the explanations provided by the proposed S(H)NAP (e.g., deletion metric and insertion metric (Petsiuk et al., 2018)) against those provided by other post-hoc feature attribution methods (e.g., LIME, GradCAM, etc.).
- There is also a lack of qualitative comparisons with other post-hoc feature attribution methods.

Reference:
Petsiuk et al. (2018). RISE: Randomized Input Sampling for Explanation of Black-box Models.

---

> ### Author Rebuttal · Authors · 2026-03-30
>
> We thank the reviewer for their valuable feedback and hope these clarifications encourage a score increase.
>
> # W3. and W4.: Quantitative evaluation of S(H)NAP and comparison to other post-hoc attribution methods.
>
> We provide a comprehensive comparison of S(H)NAP with white-box gradient-based (Saliency, Guided Grad-CAM, Input X Gradient, Integrated Gradients) and black-box perturbation-based (Feature Ablation, KernelSHAP, Occlusion 3D) methods. We evaluate localization (relative regional attribution (RRA) using pulmonary nodules as ground truth, aligning with standard practice), faithfulness (insertion/deletion curves, area under the perturbation curve (AUC Pert)), complexity (Sparseness), and exactness (Completeness).
>
> Comparison: https://www.dropbox.com/scl/fi/1xsl9nfd2iqal4wlxb7ux/1_quantitative_evaluation.png?rlkey=rye21fq04llkwt0v2lcb2cl2x&st=0tsbnm9f&dl=0
>
> As shown, our method significantly outperforms standard approaches on typical explanation quality measures. However, these metrics function as mere proxies and often rely on unrealistic assumptions (e.g., OOD perturbations with black pixels). Ultimately, the *actual* correspondence to the model and the provided insights matter most. SHNAP is uniquely capable of an *analytically tractable* and *exact* faithfulness evaluation via the $R^2$ coefficient on *all* coalitions—a computation practically impossible for standard methods like KernelSHAP. Supported by quantitative and human evaluations, SHNAP provides *actionable insights* with direct *semantic* meaning, highlighting actual knowledge gained about Sybil while perfectly approximating it. Such clarity is impossible to deduce from standard post-hoc or counterfactual explanations, which remain inherently ambiguous.
>
> # W1. and W2.: Advancements and scope.
>
> We must argue that focusing on a single model is not a weakness. Within the medical imaging community, Sybil is an immensely popular model (>300 citations). It is often considered the AlexNet of lung cancer screening due to providing an end-to-end solution to a notoriously difficult, clinically vital problem without handcrafted features, requiring only a single LDCT. This is supported by the sheer amount of independent global scientific validations of Sybil (see lines 48--51 for 9 works), which to date have not detected the plethora of pitfalls presented in our paper.
>
> Moreover, the methodological advancements in our paper are significant:
> - To date, we propose the first approach for training a generative model capable of manipulating pulmonary nodules that does not require nodule labels during training (inference only), allowing for training at a scale of tens of thousands of 3D volumes.
> - Our method is the first to simultaneously remove nodules and insert new ones with _known_ malignancy, opening crucial research directions in synthetic data augmentation and fine-tuning.
> - We introduce the first medical imaging approach considering attributions at the level of modular semantic units (pulmonary nodules) vital to experts. This shift allows adopting $n$-Shapley Values as an _analytically tractable_ and _exact_ solution to the explanatory problem.
> - Using rigorously validated in-distribution perturbations, S(H)NAP reveals previously unknown and undetectable flaws in Sybil.
>
> In summary, our approach is not a simple mix of existing techniques; its innovative design _allows_ incorporating advanced tools that are, for the first time, exactly computable.
>
> # Questions
>
> Q1. As stated in lines 79–83, we focus on the base hazard logit because it almost entirely determines outputs for consecutive years (Fig. 9 correlation matrix). Remaining predictions differ by simple offsets that determine temporal risk gain, not the core hazard. While S(H)NAP can explain any output, the base hazard is the most important.
>
> Q2. As mentioned in the final section of our paper, multiple lung cancer risk prediction models exist that can benefit from a S(H)NAP-based analysis. For a practical example of an extension, we apply S(H)NAP to a Chest X-Ray Lung Cancer (CXR-LC) risk prediction model [1] by simulating chest x-rays from 3D LDCT. The highly skewed fit of SHNAP in the results below demonstrates that the explanations capture almost all of CXR-LC's decision-making, highlighting the applicability of S(H)NAP to other models.
>
> Fit: https://www.dropbox.com/scl/fi/lvkma7ti3xoib41kj6o38/1a_faitfhulness.png?rlkey=bddlogk7zss7naiqkmqw2y4ji&st=qj2cicbv&dl=0
>
> Fit by nodule count: https://www.dropbox.com/scl/fi/kby8rgqs1m5qiyrqetfz6/1b_faitfhulness_by_nodule_count.png?rlkey=me3t6ispzby7waql7ldqcgh7r&st=tjgewm1c&dl=0
>
> Fit statistics: https://www.dropbox.com/scl/fi/aepvf4i6pzhgnlyrkrsyq/1c_faitfhulness_table.png?rlkey=mq81tnxszemwi51y8rdfptsfl&st=adztfbth&dl=0
>
> [1] Lu, M. T. et al., _CXR-LC: Deep learning using Chest Radiographs to Identify High-risk Smokers for Lung Cancer Screening: Development and Validation of a Prediction Model_, _Annals of Internal Medicine_, 2020

---

> > ### Author Rebuttal · Reviewer_x26A · 2026-04-03
> >
> > My concerns have been mostly addressed by the authors, and I have increased my score to 4: Weak accept.

---

> > > ### Author Response · Authors · 2026-04-05
> > >
> > > We sincerely thank the Reviewer for their engagement during the rebuttal process and for their positive assessment of our work. Should there be any remaining concerns, we would welcome the opportunity to provide further clarification.

---

### Decision · Program_Chairs · 2026-04-30

**Decision:**

Accept (regular)

**Comment:**

This paper proposes a clinically guided explainability framework for auditing a foundation model for lung cancer risk prediction, approximating the model with a nodule-based linear surrogate including pairwise interaction terms and computing 2-Shapley values to expose reasoning misalignments. The work addresses a genuine and important gap: strong observational performance does not certify that a model's internal mechanism is appropriate for high-stakes clinical deployment. Notable strengths include the principled use of generative models to perform in-distribution nodule insertion and erasure, the convincing blinded radiologist study validating generation fidelity, and the practical relevance of the audit framework. The main limitations acknowledged by reviewers concern the modest methodological novelty - the approach builds on existing n-Shapley value theory and generative AI techniques - the restriction of the audit to a single Sybil instance, the absence of attribution map comparisons to classical methods, and the lack of proposed remedies for the identified biases. However, the area chair notes that application-oriented contributions of this quality are valuable to the community, that the author rebuttal addressed the raised concerns convincingly, and that the identified shortcomings are largely outside the paper's stated scope. Despite two reviewers maintaining their ratings after rebuttal - one of whom awarded top scores across all subcategories - the overall consensus of three weak accepts and one weak reject supports a recommendation of **accept**.